# Removal of phosphorus and nitrogen in sediments of the eutrophic Stockholm Archipelago, Baltic Sea

Niels A.G.M. van Helmond[1,2,3,a], Elizabeth K. Robertson[2,4a], Daniel J. Conley[2], Martijn Hermans[1], Christoph Humborg[5], L. Joëlle Kubeneck[1b], Wytze K. Lenstra[1] and Caroline P. Slomp[1]

[1]Department of Earth Sciences, Faculty of Geosciences, Utrecht University. Princetonlaan 8a, 3584 CB Utrecht, The Netherlands
[2]Department of Geology, Lund University, Sölvegatan 12, 223 62 Lund, Sweden.
[3]Department of Microbiology, Institute for Water and Wetland Research, Radboud University, Heyendaalseweg 135, 6525 AJ Nijmegen, The Netherlands.
[4]Department of Marine Sciences, University of Gothenburg, Box 461, 40530 Göteborg, Sweden;
[5]Baltic Sea Centre, Stockholm University, 106 91 Stockholm, Sweden;
[a] these authors contributed equally to this work
[b] now at: Institute of Biogeochemistry and Pollutant Dynamics, Department of Environmental Systems Science, ETH Zürich. Universitätstrasse 16, 8092 Zürich, Switzerland.

*Correspondence to*: Niels A.G.M. van Helmond (n.vanhelmond@uu.nl)

**Abstract.** Coastal systems can act as filters for anthropogenic nutrient input into marine environments. Here, we assess the processes controlling the removal of phosphorus (P) and nitrogen (N) for four sites in the eutrophic Stockholm Archipelago. Bottom water concentrations of oxygen ($O_2$) and P are inversely correlated. This is attributed to the seasonal release of P from iron (Fe)-oxide-bound P in surface sediments and from degrading organic matter. The abundant presence of sulfide in the pore water and its high upward flux towards the sediment surface (~4 to 8 mmol m$^{-2}$ d$^{-1}$), linked to prior deposition of organic-rich sediments in a low $O_2$ setting ("legacy of hypoxia"), hinders the formation of a larger Fe-oxide-bound P pool in winter. This is most pronounced at sites where water column mixing is naturally relatively low and where low bottom water O2 concentrations prevail in summer. Burial rates of P are high at all sites (0.03-0.3 mol m$^{-2}$ y$^{-1}$), a combined result of high sedimentation rates (0.5 to 3.5 cm yr$^{-1}$) and high sedimentary P at depth (~30 to 50 μmol g$^{-1}$). Sedimentary P is dominated by Fe-bound P and organic P at the sediment surface and by organic P, authigenic Ca-P and detrital P at depth. Apart from one site in the inner archipelago, where a vivianite-type Fe(II)-P mineral is likely present at depth, there is little evidence for sink-switching of organic or Fe-oxide bound P to authigenic P minerals. Denitrification is the major benthic nitrate-reducing process at all sites (0.09 to 1.7 mmol m$^{-2}$ d$^{-1}$) with rates decreasing seaward from the inner to outer archipelago. Our results explain how sediments in this eutrophic coastal system can remove P through burial at a relatively high rate, regardless of whether the bottom waters are oxic or (frequently) hypoxic. Our results suggest that benthic N processes undergo annual cycles of removal and recycling in response to hypoxic conditions. Further nutrient load reductions are expected to contribute to the recovery of the eutrophic Stockholm Archipelago from hypoxia. Based on the dominant pathways of P and N removal identified in this study, it is expected that the sediments will continue to remove part of the P and N loads.

## 1. Introduction

Anthropogenic activities are altering coastal marine ecosystems worldwide (Jackson et al., 2001; Halpern et al., 2008; Diaz and Rosenberg, 2008). Excessive inputs of the nutrients phosphorus (P) and nitrogen (N), primarily derived from agricultural activities and wastewater, have led to widespread eutrophication, particularly in coastal areas (e.g. Nixon, 1996; Smith, 2003; Rabalais et al., 2009). Besides increased marine primary productivity, often in the form of harmful algal blooms (Anderson et al., 2003), eutrophication results in depletion of bottom water oxygen ($O_2$), as a result of increased $O_2$ consumption upon degradation of organic material (Diaz and Rosenberg, 2008; Rabalais et al., 2010).

Restoration of coastal ecosystems requires a reduction in eutrophication (e.g. Boesch, 2002). However, simply decreasing nutrient loading often does not render the desired effect because of nonlinearities in the response of coastal ecosystems to changes in nutrient loading (Duarte et al., 2009; Kemp et al., 2009; Carstensen et al., 2011). Much of this behavior is due to recycling of nutrients from the seafloor and the $O_2$ demand created by the historical deposition of organic-rich sediments ("legacy of hypoxia"; Conley et al., 2002; Turner et al., 2008; Hermans et al., 2019a). In addition, cyanobacteria, which are frequently present in eutrophic systems, can biologically fix atmospheric N ($N_2$), thereby sustaining eutrophication (e.g. Paerl and Otten, 2013).

Coastal systems also act as temporary and permanent sinks for nutrients and as filters for adjacent marine environments (e.g. Cloern, 2001; McGlathery et al., 2007; Bouwman et al., 2013). In brief, coastal environments can lead to: (1) transformation of nutrients, changing their chemical form, e.g. from dissolved to particulate and from inorganic to organic, (2) the retention of nutrients, i.e. delaying the nutrient flow from terrestrial to marine environments by incorporating nutrients into biomass or abiotic particles, (3) the removal of nutrients, i.e. permanent direction of nutrients out of the ecosystem (Asmala et al., 2017). The filtering capacity of coastal systems depends on a variety of biological, physical and chemical parameters, such as the concentrations of nutrients and dissolved $O_2$ (McGlathery et al., 2007), the presence and type of flora and fauna (Norkko et al., 2012; Krause-Jensen and Duarte, 2016), the type of coastal system and underlying sediment (Asmala et al., 2017), wind, tides and the water residence time (Nixon et al. 1996; Josefson and Rasmussen 2000). This can lead to a wide variety of removal rates for P and N in different types of coastal environments (Asmala et al., 2017; Asmala et al., 2019).

Most removal of P in coastal systems takes place through burial in fine-grained sediments. The main P-burial phases are: (1) P associated with organic matter, (2) P bound to iron (Fe)-(oxyhydr)oxides (henceforth termed Fe-oxides) and (3) P in authigenic carbonate fluorapatite (Ruttenberg and Berner, 1993; Slomp et al., 1996). Recent work has shown that P may also be sequestered in the form of vivianite-type Fe(II)-phosphate minerals, particularly in low-salinity environments with high inputs of Fe-oxides (e.g. Egger et al., 2015). Burial of P is redox sensitive, with retention of P bound to Fe-oxides and in organic matter decreasing upon increased hypoxia and anoxia (e.g. Van Cappellen and Ingall, 1994). However, a more limited exposure to $O_2$ also enhances the preservation of organic matter, and may allow organic P to become the dominant form of P in the sediment (Lukkari et al., 2009; Mort et al., 2010; Slomp, 2011).

Fixed N can be removed via multiple pathways: (1) denitrification, (2) anaerobic ammonium ($NH_4^+$) oxidation (anammox) and (3) burial in sediments. Burial of N generally only represents a small fraction of the total N removed (e.g. Gustafsson et al., 2012; Almroth-Rosell et al., 2016). In coastal systems, benthic denitrification is generally the dominant pathway for N removal (e.g. Seitzinger, 1990; Dalsgaard et al., 2005). However, dissimilatory nitrate ($NO_3^-$) reduction to ammonium (DNRA), also competes for $NO_3^-$ in sediments and reduces

$NO_3^-$ to $NH_4^+$, thereby potentially contributing to internal N recycling (Thamdrup, 2012; Giblin et al., 2013).

Field, laboratory and modelling studies have indicated that DNRA may dominate over N removal when $NO_3^-$ is limited (e.g. Algar and Vallino, 2014; Kraft et al., 2014; Kessler et al., 2018), which frequently occurs during bottom water hypoxia (e.g. Christensen et al., 2000; Nizzoli et al., 2010; Jäntti and Hietanen, 2012). Thus, the partitioning between N removal (as $N_2$) from the ecosystem or transformation of organic-N to $NH_4^+$, which can be retained in the ecosystem, may be strongly influenced by eutrophic conditions.

Predictions of the response of coastal areas to decreased nutrient inputs and/or natural or artificial reoxygenation require insight in the processes responsible for P and N cycling and whether P and N are transformed, retained or removed. This is of particular relevance to the coastal zone of the Baltic Sea because of its highly eutrophic and frequently low $O_2$ state (Conley et al., 2011). Active nutrient reductions from the 1980s onward (Gustafsson et al., 2012) are now leading to the first signs of recovery in the region (Andersen et al., 2017), although not yet in the Baltic Proper (e.g. Hansson et al., 2019). A good example of a recovering coastal system within the Baltic Sea is the Stockholm Archipelago (Karlsson et al., 2010), where, based on modeling, recovery from hypoxia was suggested to be potentially associated with the build-up of a pool of Fe-oxide bound P in surface sediments driven by increased macrofaunal activity (Norkko et al., 2012). However, this mechanism would not lead to increased permanent P burial and hence, by itself, not lead to long term recovery of the system (> 5-10 years).

Based on coupled physical and biogeochemical models it was recently suggested that the Stockholm Archipelago was very efficient in removing P and N for the period 1990-2012, accounting for loss of 65 % of the land-derived P input and 75 % of the land-derived and atmospheric N input (Almroth-Rosell et al., 2016). The area-specific P and N retention was highest in the inner part of the Stockholm Archipelago. Based on the high $NO_3^-$ concentrations in the bottom water, and high organic carbon contents in the sediment in the Archipelago, benthic denitrification is expected to dominate N removal (Almroth-Rosell et al., 2016; Asmala et al., 2017). Recent mass balance modelling for the inner Archipelago suggests that sediments are a P sink in winter and a source in summer and autumn, with low annual net retention in the sediments (Walve et al., 2018). These apparently conflicting results between different modelling approaches emphasizes the need to better understand and quantify P removal, i.e. permanent burial of P in the sediment.

The objectives of this study are to identify and quantify the main P-burial phases and the processes controlling removal of N in sediments of the Stockholm Archipelago. We present geochemical depth profiles for a range of sediment components (P, Fe, organic carbon) and rate measurements of benthic N cycling processes for four sites along a gradient from the inner archipelago towards the open Baltic Sea. These sites capture a range of bottom water $O_2$ concentrations from seasonally hypoxic/occasionally euxinic to oxic. Our results highlight the key processes in sediments in eutrophic coastal systems that lead to removal of P and N and that may prevent their further transport to the marine environment.

## 2. Materials and methods

### 2.1 Study area

The Stockholm Archipelago covers ~5000 $km^2$, is formed by (post-)glacial processes and consists of approximately 30.000 mostly rocky islands that are surrounded by a network of basins and straits of different shapes, sizes and depths (Hill and Wallström, 2008). Based on the connections and rates of water exchange between the different basins and the open Baltic Sea the Stockholm Archipelago can be divided into an inner,

intermediate and outer archipelago (Almroth-Rosell et al., 2016). The Norrström river connects the Stockholm Archipelago to its main freshwater source, Lake Mälaren, which, on average, discharges about 160 m$^3$ s$^{-1}$ of freshwater into the most western part of the archipelago in central Stockholm (Lindh, 2013). As a consequence, surface waters in this part of the archipelago are nearly freshwater, whereas those in the outer archipelago have an average salinity of ~7 because of input of brackish water from the open Baltic Sea (Engqvist and Andrejev, 2003; Hill and Wallström, 2008). Particularly in the inshore parts of the archipelago, a (weak) halocline develops due to the differences in salinity between the (nearly) fresh surface water and the underlying more saline water. In the summer, water column stratification is more pronounced and widespread due to the development of a thermocline. However, in the more open parts of the archipelago, wind-driven mixing may interrupt stratification (Gidhagen, 1987).

The average annual nutrient input into the Stockholm Archipelago was 217 t P and 8288 t N for the period 1990-2012, of which approximately 174 t P and 5846 t N entered the inner archipelago via the Norrström river (Almroth-Rosell et al., 2016). This high nutrient load mostly originates from wastewater treatment facilities of Stockholm (Johansson and Wallström, 2001) and, in combination with (seasonal) stratification of the water column, led to widespread eutrophication in the past. As a result, large parts of the Stockholm Archipelago are or have been (seasonally) hypoxic to euxinic over the past century (Jonsson et al., 1990; Conley et al., 2011). Studies have shown decreases in dissolved inorganic P and total P due to reductions in nutrient inputs from sewage treatment plants (Walve et al., 2018) and indications of environmental recovery have been deduced from visual observations of sediment cores (Karlsson et al., 2010).

### 2.1.1 Study sites

For this study, sediments and bottom water from four different locations in the inner and intermediate part of the Stockholm Archipelago (cf. Almroth-Rosell et al., 2016; Fig. 1) were collected. The study sites are located in the basins Strömmen (central Stockholm), Baggensfjärden, Erstaviken and Ingaröfjärden and are characterized by a range of water depths and bottom water redox conditions (Fig. 2; Table 1; Sup. Fig. 1). Strömmen is located most proximate to the outlets of the sewage treatment plants close to the center of Stockholm, which presumably contributes to the relatively high total P concentrations in the bottom water (Fig. 3a). Baggensfjärden is the most restricted basin in this study (i.e. land-locked with narrow and relatively shallow connections to adjacent basins), leading to reduced water column mixing, culminating in low O$_2$ bottom waters annually in summer. Erstaviken and Ingaröfjärden have a more open connection with the Baltic Proper leading to better water column mixing. Ingaröfjärden is the least restricted basin in this study and subsequently the most consistently well-oxygenated basin throughout the year. Extensive water quality monitoring of the study area (obtained from the SHARK database at http://www.smhi.se/klimatdata/oceanografi/havsmiljodata/marina-miljooovervakningsdata; Swedish Meteorological and Hydrological Institute - SMHI, 2019), shows a clear inverse correlation between bottom water O$_2$ concentrations and P and a positive correlation between bottom water O$_2$ and N/P-ratios (Fig. 3a, b). Bottom water O$_2$ and nutrient concentrations follow a distinct annual pattern, with maximum O$_2$ and minimum nutrient concentrations in winter. After winter, O$_2$ gradually drops and nutrient concentrations gradually increase, reaching minimum and maximum values, respectively, at the end of summer and in autumn, followed by a reset of the system (Fig. 3c, d). The difference in amplitude of the

changes in and $O_2$ total P concentrations between sites is likely the combined effect of differences in water column mixing and in recycling of P from the sediment associated with changes in bottom water $O_2$.

**2.2 Sampling**

Sediment cores were collected with *R/V Electra* in March 2017. Prior to coring, a CTD (Sea-Bird 911plus), equipped with a circular Rosette of Niskin bottles (12 x 5 L), was deployed to determine key water column characteristics at the time of sampling, such as dissolved $O_2$ concentrations, temperature and salinity (Table 2) and to collect bottom water. At each site, ~20 Gemini cores (2 cores per cast; Ø = 8 cm; between 40 and 60 cm of sediment and >10 cm of overlying water) were collected for analysis of methane ($CH_4$), high-resolution micro-electrode depth profiling, (anoxic) sediment and pore water collection, $N_2$ slurry and incubation experiments and sieving for macrofauna (Table 2).

Samples for $CH_4$ analysis were taken directly after coring via pre-drilled holes (taped prior to coring) in the Gemini core-liner with a depth-spacing of 2.5 cm as described in Lenstra et al. (2018).

High-resolution (50 µm) depth profiles of dissolved $O_2$ were obtained from one core per site, using microelectrodes (Unisense A.S., Denmark), as described in Hermans et al. (2019b). The diffusive uptake of $O_2$ was determined by numerical modelling with PROFILE (Berg et al., 1998) using the high-resolution $O_2$ measurements.

For anoxic sediment and pore water collection, one core was sliced in a $N_2$-filled glove bag. Two bottom water samples were taken from the overlying water after which the core was sliced at a resolution of 0.5 cm (0 to 10 cm), 2 cm (10 to 20 cm), 4 cm (20 to 40 cm) and 5 cm until the bottom of the core. The sediment was centrifuged (in 50 mL tubes) at 3500 rpm for 20 minutes to extract pore water. The sediment remaining after centrifugation was stored in $N_2$-flushed gas-tight aluminum bags at -20 °C until further analysis. Bottom and pore water samples were filtered over a 0.45 µm filter in a $N_2$-filled glove bag. Subsamples were taken for (1) $H_2S$ analysis (0.5 mL was added to 2 mL 2 % zinc (Zn)-acetate); (2) analysis of dissolved Fe and P (1 mL was acidified with 10 µL 30 % suprapur HCl); (3) analysis of sulfate ($SO_4^{2-}$) (0.5 mL), and stored at 4°C. Subsamples for N-oxides ($NO_x = NO_3^- +$ nitrite ($NO_2^-$); 1 mL) and $NH_4^+$ (1 mL) were stored at -20 °C.

At Strömmen, one core was sliced at the same resolution as described above to determine porosity and $^{210}Pb$. Data for porosity and $^{210}Pb$ for the other three study sites were taken from van Helmond et al. (in review).

**2.3 Bottom and pore water analysis**

Concentrations of $CH_4$ were determined with a Thermo Finnigan Trace gas chromatograph equipped with a flame ionization detector as described by Lenstra et al. (2018). The average analytical uncertainty based on duplicates and triplicates was <5 %. Pore water $H_2S$ was determined spectrophotometrically using phenylenediamine and ferric chloride (Cline, 1969). Upward fluxes of $H_2S$ in the pore water towards the sediment surface were calculated as detailed in Hermans et al. (2019a). Dissolved Fe and P (assumed to be present as $Fe^{2+}$ and $HPO_4^{2-}$) were measured by Inductively Coupled Plasma-Optimal Emission Spectroscopy (ICP-OES; SPECTRO ARCOS). Nitrogen-oxides (Schnetger and Lehners, 2014) and $NO_2^-$ (Grasshoff et al. 1999) were determined colorimetrically. Concentrations of $NO_3^-$ were calculated from the difference between $NO_x$ and $NO_2^-$ concentrations. Ammonium was determined colorimetrically using indophenol-blue (Solorzano,

1969). Concentrations of $SO_4^{2-}$ were determined by ion chromatography. The average analytical uncertainty based on duplicates was <1 %.

**2.4 Solid phase analysis**

All sediment samples were freeze-dried, powdered and homogenized using an agate mortar and pestle in an argon-filled glovebox. Prior to analysis, samples were split into oxic and anoxic fractions (i.e. samples stored open to air and in a $N_2$ or argon atmosphere).

**2.4.1 Total elemental composition**

Approximately 125 mg of the oxic sediment split was digested in a mixture of strong acids as described by van Helmond et al. (2018). The residues were dissolved in 1 M $HNO_3$ and analysed for their elemental composition by ICP-OES. Average analytical uncertainty based on duplicates and triplicates was <5 % for calcium (Ca) and <3 % for P. The calcium carbonate content ($CaCO_3$ wt.%) was calculated based on the Ca content measured by ICP-OES, assuming that all Ca was in the form of $CaCO_3$.

**2.4.2 Organic carbon and nitrogen**

Between 200 and 300 mg of the oxic sediment split was decalcified using 1 M HCl as described by van Helmond et al. (2018) after which dried and re-powdered residues were analysed for their carbon and nitrogen content with a Fisons Instruments NA 1500 NCS analyzer. Average analytical uncertainty based on duplicates was <2 % for carbon and <3 % for nitrogen. Organic carbon ($C_{org}$) and nitrogen ($N_{org}$) contents were calculated after a correction for the weight loss upon decalcification and the salt content of the freeze-dried sediment. For Baggensfjärden, Erstaviken and Ingaröfjarden C and N contents were taken from van Helmond et al. (in review).

**2.4.3 Sequential extraction of iron**

Between 50 and 100 mg of the anoxic sediment split was subjected to a sequential extraction procedure based on a combination of the procedures by Poulton and Canfield (2005) and Claff et al. (2010) to determine the different phases of sedimentary Fe (Kraal et al., 2017). Briefly, under $O_2$-free conditions: (1) 10 mL 1 M HCl, pH 0 was added to extract (4 h) Fe(II) and Fe(III) minerals such as easily reducible Fe-oxides (e.g. ferrihydrite and lepidocrocite), Fe-carbonates and Fe-monosulfides; (2) 10 mL 0.35 M acetic acid/0.2 M $Na_3$-citrate/50 g $L^{-1}$ Na dithionite, pH 4.8 was added to extract (4 h) crystalline Fe oxide minerals such as goethite and hematite; (3) 10 mL 0.17 M ammonium oxalate/0.2 M oxalic acid, pH 3.2 was added to extract (6 h) recalcitrant oxide minerals such as magnetite; (4) 10 mL 65 % $HNO_3$ was added to extract (2 h) pyrite ($FeS_2$). For all extracts, Fe concentrations were determined colorimetrically with the phenanthroline method, adding hydroxylamine-hydrochloride as a reducing agent to convert all $Fe^{3+}$ into $Fe^{2+}$ (APHA, 2005). For the first step the absorbance before and after addition of the reducing agent was measured, in order to separate $Fe^{2+}$ and $Fe^{3+}$. The Fe concentrations of the $Fe^{3+}$ fraction of the first step and the second step were summed, and are henceforth referred to as Fe-oxides. Average analytical uncertainty based on duplicates and triplicates was <10 % for all fractions.

### 2.4.4 Sequential extraction of sulfur

Approximately 300 mg of the anoxic sediment split was subjected to a sequential extraction procedure (Burton et al., 2008) to determine sedimentary sulfur phases. Briefly, under $O_2$-free conditions: (1) 10 mL 6 M HCl and 2 mL 0.1 M ascorbic acid were added to dissolve acid-volatile sulfur (AVS, assumed to represent Fe-monosulfides - FeS) and the released $H_2S$ was trapped in a tube filled with 7 mL alkaline zinc acetate solution (24 h); (2) 10 mL acidic chromium(II)chloride was added to dissolve chromium-reducible sulfur (CRS, assumed to represent $FeS_2$) and the released $H_2S$ was trapped in a tube filled with 7 mL alkaline zinc acetate solution (48 h). For both fractions, the amount of sulfur in the zinc sulfide precipitates was determined by iodometric titration (APHA, 2005). Average analytical uncertainty, based on duplicates, was <7 % for both AVS and CRS.

### 2.4.5 Sequential extraction of phosphorus

Approximately 100 mg of the anoxic sediment split was subjected to a sequential extraction procedure following the procedure of Ruttenberg (1992), modified by Slomp et al. (1996), but including the exchangeable P step. Briefly, under $O_2$-free conditions: (1) 10 mL 1 M $MgCl_2$, pH 8 was added to extract (0.5 h) exchangeable P (Exch. P); (2) 10 mL 0.3 M $Na_3$-citrate/1 M $NaHCO_3$/25 g $L^{-1}$ Na dithionite (CDB), pH 7.6 was added after which 10 mL 1 M $MgCl_2$, pH 8 was added, together extracting (8 h and 0.5 h, respectively) P bound to Fe fraction, including Fe-oxide bound P and vivianite (Nembrini et al.,1983; Dijkstra et al., 2014) (Fe-bound P); (3) 10 ml 1 M Na-acetate buffered to pH 4 with acetic acid was added after which 10 mL 1 M $MgCl_2$, pH 8 was added, together extracting (6 h and 0.5 h respectively) authigenic Ca-P, including carbonate fluorapatite, hydroxyapatite and carbonate-bound P (Auth. P); (4) 10 ml 1 M HCl, pH 0 was added to extract (24 h) P in detrital minerals (Detr. P); (5) ashing of the residue at 550 °C (2 h) after which 10 ml 1 M HCl, pH 0 was added to extract (24 h) P in organic matter (Org. P). The P content in the citrate-dithionite-bicarbonate extract was analysed by ICP-OES. All other solutions were measured colorimetrically (Strickland and Parsons, 1972). Average analytical uncertainty, based on duplicates, was <7 % for all fractions. Total P derived from acid digestion and subsequent ICP-OES analyses was on average within 5 % of the summed P fractions derived from the sequential extraction.

### 2.5 Sediment nitrogen cycling

### 2.5.1 $^{15}N$ incubations

Rates of benthic $NO_3^-$-reducing pathways were determined using the whole-core isotope pairing technique (IPT) and parallel slurry incubations (Nielsen, 1992; Risgaard-Petersen et al., 2003). Bottom water from Niskin bottles collected at each site was used to fill the incubation chamber (approx. 30 L) and maintained at *in situ* $O_2$ concentrations using compressed air and nitrogen gas mixtures. Small core liners (Ø 2.5 cm) were used to take sub-cores from the Gemini cores and were immediately transferred to the incubation tank so that all cores were submerged and stoppers were removed. Sodium $^{15}N$-nitrate solution ($Na^{15}NO_3$, 98 atom % $^{15}N$, Sigma Aldrich, final concentration ~50 µmol $L^{-1}$) was added to the water of the incubation tank and cores were pre-incubated in the dark at *in situ* temperature for 2 to 5 h. Three replicate cores were sacrificed by slurrying the entire sediment volume at approximately 0, 2, 5 and 8 h following pre-incubation. Sediment was allowed to settle for 2 minutes

before samples for gas (12 mL exetainers, Labco, UK, killed with 250 µL zinc chloride solution, 50 % w/v) and nutrients (10 mL, killed with 250 µL zinc chloride solution, frozen) were taken.

Sediment slurries were carried out in parallel to whole-core incubations. Briefly, a glass bead (0.5 cm Ø) was added to each 12 mL exetainer, which was then filled with filtered (0.2 µm) helium-purged bottom water. Homogenised surface sediment (2 mL, 0-2 cm depth horizon) was added to each exetainer and vials were sealed. Exetainers were incubated on a shaker table in the dark at in situ temperature for 8 to 12 h ensuring consumption of background $NO_3^-$ and $O_2$ before addition of [15]N-substrates. Exetainers were divided into two treatments, amended with sodium [15]N-nitrate or with sodium [14]N-nitrite and [15]N-ammonium chloride (each 100 µmol $L^{-1}$ final concentration). Slurries were sacrificed at approximately 0, 5 and 10 h after substrate addition by injection of 250 µL zinc chloride solution through the septum of exetainers.

### 2.5.2 Analytical methods

Analysis of [15]N composition of $N_2$ (and any nitrous oxide: $N_2O$) was determined by gas-chromatography isotope ratio mass spectrometry (GC-IRMS). A helium head space was introduced to filled exetainers and gas samples were manually injected as described in Dalsgaard et al. (2013). Any [15]N-$N_2O$ was reduced in a reduction oven and measured as [15]N-$N_2$. Determination of [15]N in $NH_4^+$ was carried out by conversion of $NH_4^+$ to $N_2$ with alkaline hypobromite iodine solution (Risgaard-Petersen et al., 1995; Füssel et al., 2012). Ammonium was extracted from sediment in slurry and whole-core samples by shaking for 1 h with 2M KCl (1:1 sample:KCl) before any $NH_4^+$ analysis. The isotopic composition of the produced $N_2$ was determined using a GC-IRMS as above. Recovery efficiency of [15]$NH_4^+$ following the hypobromite conversion was >95 %.

Concentrations of $NO_x$ ($NO_3^-$ + $NO_2^-$) in incubations were determined colorimetrically as described for pore water. For determination of total $NH_4^+$, samples were extracted with KCl as above and $NH_4^+$ concentrations were analysed colorimetrically using the salicylate-hypochlorite method (Bower and Holm-Hansen, 1980).

### 2.5.3 Data calculations

Anammox and DNRA were detectable in slurry incubations, although both processes only played a minor role in $NO_3^-$ reduction at most sites. However, they may have interfered to a minor degree with the IPT calculations. Thus areal rates of benthic N cycling processes were calculated according to Song et al. (2016) at all sites. The relative contribution of anammox to $N_2$ production (*ra*) in slurries was calculated as in Song et al. (2013) using the average mole fraction of [15]$NH_4^+$ in the total $NH_4^+$ pool ($F_A$) as this was demonstrated to increase linearly over time.

Fluxes of $NO_3^-$ and $NH_4^+$ were calculated using gradients (~0-1 cm and ~0-5 cm, respectively) of sediment pore water depth profiles and Fick's first law of diffusion. Porosity values were taken from the average porosities of the integrated depth horizons and diffusion coefficients from Schulz (2006).

### 2.6 Sediment accumulation rates

Freeze-dried sediment samples for Strömmen were measured for [210]Pb by direct gamma counting using a high purity germanium detector (Ortec GEM-FX8530P4-RB) at Lund University. [210]Pb was measured by its emission at 46.5 keV. Self-absorption was measured directly and the detector efficiency was determined by counting a National Institute of Standards and Technology sediment standard. Excess [210]Pb was calculated as the difference

between the measured total $^{210}Pb$ and the estimate of the supported $^{210}Pb$ activity as given by $^{214}Pb$ ($^{210}Pb_{exc} = {}^{210}Pb_{total} - {}^{214}Pb$).

Sediment accumulation rates for the four study sites were estimated by fitting a reactive transport model (Soetaert and Herman, 2008) to the $^{210}Pb$ depth profiles accounting for depth dependent changes in porosity (Sup. Fig. 2).

## 3. Results

### 3.1 Pore water profiles

The $O_2$ penetration depth is deepest (18 mm) at Ingaröfjärden, while at the other three sites the $O_2$ penetration depth is relatively shallow (<4 mm; Table 2; Sup. Fig. 3). The diffusive uptake of $O_2$ is high at Strömmen and Baggensfjärden (~14 mmol $m^{-2}$ $d^{-1}$) and low at Ingaröfjärden (3 mmol $m^{-2}$ $d^{-1}$; Table 2). All four sites are characterized by a shallow sulfate methane transition zone (SMTZ), with near complete $SO_4^{2-}$ removal between 7 and 15 cm (Fig. 4). Concentrations of $CH_4$ increase with depth at all sites and are highest at Erstaviken (up to 8 mmol $L^{-1}$) and lowest at Ingaröfjärden (max. ~2 mmol $L^{-1}$). At Strömmen, Baggensfjärden and Erstaviken, $H_2S$ concentrations increase rapidly with depth below 2 cm, while at Ingaröfjärden this is observed below 10 cm. After a distinct maximum (of up to 1.3 mM in Ingaröfjärden), $H_2S$ concentrations decrease again with depth, and even reach values close to zero at Strömmen and Erstaviken (at approximately 20 and 40 cm, respectively). The flux of $H_2S$ towards the sediment surface is high at all sites (~4 to 8 mmol $m^{-2}$ $d^{-1}$).

Dissolved $Fe^{2+}$ concentrations show a maximum directly below the sediment-water interface at all sites, with the highest maximum values at Strömmen (~60 µmol $L^{-1}$), and a rapid decrease to values around zero in the upper centimeters of the sediment. At Strömmen and Erstaviken dissolved $Fe^{2+}$ concentrations increase again when $H_2S$ is depleted at depth. At all sites, concentrations of $HPO_4^{2-}$ and $NH_4^+$ are low near the sediment-water interface, and then increase with depth, first quickly then more gradually. Only at Strömmen $HPO_4^{2-}$ decreases below ~15 cm. Bottom water $NO_3^-$ concentrations decrease from the inner archipelago towards the outer archipelago, i.e. Strömmen > Baggensfjärden > Erstaviken > Ingaröfjärden. For the three most inshore sites $NO_3^-$ concentrations in the bottom water are higher than $NO_3^-$ concentrations in the sediments. In contrast, at Ingaröfjärden $NO_3^-$ concentrations in the surface sediments are almost four times higher than $NO_3^-$ concentrations in the bottom water.

### 3.2 Solid phase profiles

Sediment $C_{org}$ concentrations are relatively high at all four sites (Fig. 5), whereas $CaCO_3$ concentrations are low (< 3 wt. %; Table 4). Surface sediments are enriched in $C_{org}$ by 1-2 wt. % when compared to sediments at depth. Concentrations of $C_{org}$ are highest at Strömmen and decrease from the inner archipelago towards the outer archipelago (Table 4; Sup. Fig. 4). Sediment C/N ratios are somewhat lower in the top centimeters and become constant with depth. Overall C/N values decrease towards the outer archipelago. At all four sites, surface sediments are enriched in P. The thickness of this enriched surface layer ranges from 2 to 4 cm. At Strömmen, surface P concentrations are twice as high (ranging up to 165 µmol $g^{-1}$) as those observed at the other sites. Below this enriched surface layer, P concentrations are mostly rather constant at all sites (ranging from 30 to 40 µmol $g^{-1}$). Similar to the high concentrations in the surface layer at Strömmen, sedimentary P concentrations are also high at depth (40 to 50 µmol $g^{-1}$), and two additional enrichments in P are observed at depth.

As a result of the relatively large enrichment in P in the surface sediments, $C_{org}/P_{tot}$ is low in the surface sediment. At depth $C_{org}/P_{tot}$ values are around the Redfield-ratio (Table 4). With the exception of Strömmen, surface sediments are enriched in Fe-oxides. This enrichment is most pronounced at Ingaröfjärden. At depth, Fe-oxide concentrations are relatively constant and similar for all four sites. Just below the surface, between ~1 to 10 cm, a pronounced enrichment in FeS is observed. Only at Ingaröfjärden such a pronounced enrichment in FeS is not observed, and FeS is entirely absent above 2.5 cm. Pyrite concentrations are relatively low in the surface sediments and gradually increase with depth. At Ingaröfjärden, a peak in $FeS_2$ is observed between 5 and 10 cm, superimposed on the gradual increase in $FeS_2$.

At all sites, Fe-bound P dominates the P in the surface sediments (Fig. 6). At Strömmen, Fe-bound P remains an important fraction of solid phase P, also at depth, while for the other sites Fe-bound P only represents ~10-20 % of total P. Exchangeable P shows trends similar to those observed for Fe-bound P, but concentrations are low. Detrital P, Authigenic P and P in organic matter all show relatively constant concentrations with depth. Only the P in organic matter is slightly enriched in the surface sediments. Below the Fe-bound P-dominated surface sediments, P in organic matter is the largest fraction, representing between ~30 and 40 % of the total P and between ~30 and 50 % of reactive P (i.e., the sum of Fe-bound P, exchangeable P, P in organic matter and authigenic Ca-P). Authigenic Ca-P represents ~25 to 30 % and detrital P ~20 to 25 % of total P.

## 3.3 Benthic nitrogen cycling

Bottom water $NO_3^-$ concentrations decrease from Strömmen (17.8 µmol $L^{-1}$) toward Ingaröfjärden (5.6 µmol $L^{-1}$, Table 5). The flux of $NH_4^+$ out of the sediment also decreases seawards. The sediment acts as a weak source of $NO_3^-$ to the overlying water at Strömmen while it is a $NO_3^-$ sink at the other three sites (Table 5).

Denitrification is the major $NO_3^-$-reducing process at all sites (Fig. 7; Table 5). Denitrification rates (Fig. 7) are highest at Strömmen (~1700 µmol $m^{-2}$ $d^{-1}$) and decrease towards the outer archipelago with the lowest rates at Ingaröfjärden (~100 µmol $m^{-2}$ $d^{-1}$). Nitrous oxide is not an important end-product of denitrification in whole core incubations. Nitrification is the dominant source of $NO_3^-$ for denitrification in the sediments at all sites, accounting for 60-89 % of all $NO_3^-$ supply for denitrification in the sediments (Table 5; Fig. 8). The importance of nitrification as $NO_3^-$ source relative to water column $NO_3^-$ increased towards the outer archipelago. DNRA was measurable but is not a significant $NO_3^-$-reducing pathway at any of the sites investigated, accounting for less than 1.5 % of total $NO_3^-$ reduced. Anammox plays only a minor role in overall N removal (< 1 % $N_2$ produced) at the three inner archipelago sites but accounts for 33% of $N_2$ production at Ingaröfjärden (44.1 µmol $m^{-2}$ $d^{-1}$) where overall $N_2$ production is lowest and heterotrophic denitrification was most limited in organic C substrate. Rates of N removal by denitrification are positively correlated with bottom water $NO_3^-$ concentrations and with organic carbon content (Fig. 9).

## 4. Discussion

### 4.1 Phosphorus dynamics in a eutrophic coastal system

#### 4.1.1 Phosphorus recycling

At the end of autumn and during the winter dissolved $O_2$ concentrations in the Stockholm Archipelago peak, largely due to mixing of the water column and subsequent ventilation (Fig. 3c,d; Sup. Fig. 1). After winter, $O_2$

concentrations decrease during spring and summer, following enhanced $O_2$ consumption by degrading organic matter after the spring bloom and reaching minimum values at the end of summer and in autumn. The loss of $O_2$ from the bottom water is further enhanced by reduced ventilation of deeper waters following intensified water column stratification as a result of formation or strengthening of the thermocline (Gidhagen, 1987), which at many locations in the Stockholm Archipelago leads to hypoxia (Karlsson et al., 2010; Conley et al., 2011). In addition to nutrient availability, spring bloom intensity and water depth, hydrological restriction may contribute to low $O_2$ conditions. This is also reflected at our study sites, with Baggensfjärden being the most $O_2$ depleted and restricted basin (i.e. land-locked with narrow and relatively shallow connections to adjacent basins) and Ingaröfjärden being the least restricted and subsequently, the most consistently well-oxygenated basin throughout the year (Table 1; Figs. 1, 2 and Sup. Fig. 1).

High dissolved $O_2$ concentrations allow the formation and presence of Fe-oxides (Fig. 5) in the surface sediments that bind P (e.g. Slomp et al., 1996; Fig. 6). Low dissolved $O_2$ concentrations, however, lead to the dissolution of Fe-oxides in the surface sediments. The P associated with these Fe-oxides can then be released into the water column again. This mechanism leads to P recycling in basins with strong (seasonal) contrasts in bottom water redox conditions, such as Baggensfjärden, where the sediments are a sink for P in the winter and a source for P in the spring and the summer (Fig. 3c), as also described previously for other basins in the Stockholm Archipelago (Walve et al., 2018). Nevertheless, in year-round well-oxygenated basins, such as Ingaröfjärden, this seasonal P recycling is (nearly) absent (Fig. 3a). In such basins, deeper $O_2$ penetration, which might partly be related to the presence of macrofauna (Sup. Fig. 3), leads to a thicker Fe-oxide bearing layer (Fig. 5) and a larger and stable Fe-bound P pool (Fig. 6), hence a larger enrichment of P in the surface sediments (Fig. 10). Besides Fe-oxides, a major part of the surface sediment P pool consists of P in organic matter (Fig. 6), which, based on the C/N values close to the Redfield-ratio (Fig. 5), is predominantly of marine origin. Part of the organic matter (and the P associated with it) is lost with depth (Fig. 6), because the most labile organic matter is degraded in the upper centimeters of the sediment, releasing the P associated with it to the pore water. For our study sites in the Stockholm Archipelago we calculated that this surface sediment P pool, i.e. the P active in turn-over as earlier already suggested by Rydin et al. (2011), varies between 0.036 mol P $m^{-2}$ at Baggensfjärden and 0.172 mol P $m^{-2}$ at Ingaröfjärden (between ~1 and 5 g P $m^{-2}$, respectively; Fig. 10; Table 6). This is comparable to values found for previously studied sites in the Stockholm Archipelago (1 to 7 g P $m^{-2}$; Rydin et al., 2011; Rydin and Kumblad, 2019). The surface sediment P pool, could, however, have been much larger for Strömmen, Baggensfjärden and Erstaviken if all of the FeS in the surface sediments would seasonally transform to Fe-oxides. The lack of such a transformation is likely linked to the high upward flux of $H_2S$ to the surface sediment (4.2 to 7.6 mmol $m^{-2}$ $d^{-1}$; Table 3). Besides the $H_2S$ flux, there is a relatively large efflux of $NH_4^+$ from the sediments into the bottom water (up to 1.4 mmol $m^{-2}$ $d^{-1}$; Table 5). Both the $H_2S$ and the $NH_4^+$ flux originate from decomposing organic rich sediments at depth (Fig. 4). Upon aerobic oxidation, two moles of $O_2$ are consumed per mole of $H_2S$ or $NH_4^+$ (e.g. Reed et al., 2011). Thus, the $O_2$ demand resulting from these $H_2S$ and $NH_4^+$ fluxes is very high when compared to the diffusive flux of $O_2$ into the sediment (3.1 – 13.8 mmol $m^{-2}$ $d^{-1}$; Table 2). As a consequence of the presence of $H_2S$ in the surface sediments and its high upward flux, in combination with reduced water column mixing and/or seasonally low $O_2$ bottom water conditions, FeS is formed and/or preserved (Fig. 5), and formation of a large(r) pool of Fe-oxides and Fe-bound P pool is hindered at Strömmen, Baggensfjärden and Erstaviken. At Ingaröfjärden, the well-mixed water column and year-round

well-oxygenated bottom water allow a deeper $O_2$ penetration (Sup. Fig. 3), preventing the presence of $H_2S$ in the surface sediment despite its high upward flux (Table 3) and leading to a thicker Fe-oxide bearing layer (Fig. 5) and a larger Fe-bound P pool (Fig. 6).

### 4.1.2 Phosphorus burial

Absolute P concentrations in the sediments in the Stockholm Archipelago (Figs. 6 and 10 in this study and in Rydin et al., 2011) are high (~30 to 50 $\mu$mol $g^{-1}$) in comparison with most other studied sites in the coastal zone of the Baltic Sea (generally <30 $\mu$mol $g^{-1}$; Jensen et al., 1995; Carman et al., 1996; Lenstra et al., 2018). The relatively low $C_{org}/P_{tot}$ values in the top ~2 cm, which are around the Redfield-ratio (Fig. 5), show that the seasonal $O_2$ depletion of bottom waters in our study area is not severe or long enough to cause substantial

preferential regeneration of P relative to C (Algeo and Ingall, 2007; Sulu-Gambari et al., 2018). The combination of high absolute P concentrations and relatively high sedimentation rates leads to relatively high rates of P burial (Table 6; Sup. Fig. 5). Further research of P burial rates at additional locations in the Stockholm Archipelago, including the impact of anthropogenic activities on sedimentation rates (e.g. near-shore construction and dredging) and of redeposition of sediments that have already undergone one or multiple

diagenetic cycles (after resuspension due to, for example, land uplift; Jonsson et al., 1990; Bryhn and Håkanson, 2011) is required before these results can be extrapolated to the scale of the entire system. Furthermore, it remains unclear what parts of the Stockholm Archipelago represent areas of net sediment accumulation (Karlsson et al., 2019; Asmala et al., 2019) and how much P (and in what form) is buried in euxinic parts of the Stockholm Archipelago. Hence, our results cannot be directly used to resolve the apparent discrepancy between

the model results of Almroth-Rosell et al. (2016) and Walve et al. (2018).

The constant concentrations of most P forms in the sediment below the clearly "enriched" surface sediments, suggest there generally is little to no sink-switching of sediment P (i.e. the transformation of relatively labile P reservoirs such as Fe-oxide bound P and organic P to authigenic P minerals such as for example vivianite) forms in the Stockholm Archipelago. The curved shape of the pore water $HPO_4^{2-}$ profiles indicate, however, that there

is still some release of P to the pore water at depth and we attribute this to slow degradation of organic matter. Both the detrital and authigenic (Ca-P) fractions are likely buried in the form in which they reached the sediment-water interface. The general dominance of P in organic matter and apatite (authigenic and detrital P; Fig. 6) at depth (representing permanent P burial, since the release of P from organic P is only minor), agrees with previous findings for organic rich sediments in the Baltic Sea (e.g. Jensen et al., 1995; Carman et al., 1996;

Mort et al., 2010; Rydin et al., 2011). By contrast, in the Bothnian Sea, Fe-bound P is a much more important P pool at depth (e.g. Egger et al., 2015; Lenstra et al., 2018). Evidence for potential sink-switching is only found at Strömmen, which is characterized by a larger Fe-bound P pool at depth (Fig. 6). This larger Fe-bound P pool at depth contributes to the high P burial rate at Strömmen (Table 6; Sup. Fig. 5). Coastal sediments with a shallow SMTZ, relatively high inputs of Fe-oxides and organic matter and high sediment accumulation rates are

prime locations for formation of vivianite-type minerals (e.g. Slomp et al., 2013; Egger et al., 2015). The presence of dissolved $Fe^{2+}$ and decreasing dissolved $HPO_4^{2-}$ concentrations at depth at Strömmen (Fig. 4) in combination with elevated Fe-bound P in the lower part of the record (Fig. 6), hence may result from the formation of a vivianite-type mineral.

### 4.2  Nitrogen cycling in the Stockholm Archipelago

### 4.2.1 Spatial differences in benthic N dynamics

Denitrification is by far the dominant pathway of $NO_3^-$ reduction at our study sites, accounting for ~80 to 99 % of total dissimilatory $NO_3^-$ reduction (DNRA + anammox + (2 x denitrification)) at the time of sampling (Table 4).

The dominant role of denitrification in removing N and the gradient from inner to outer archipelago agrees well with regional models based on long-term monitoring data, which show the highest N-removal capacity in the inner archipelago region (Almroth-Rosell et al., 2016; Edman et al., 2018). In the model of Almroth-Rosell et al. (2016), the inner archipelago, where Strömmen is located, annually removes approximately 3-5 times more N (~8-12 t N $km^{-2}$ $yr^{-1}$) than the intermediate and outer archipelago sites (~1-3 t N $km^{-2}$ $yr^{-1}$). Denitrification rates of both Baggensfjärden and Erstaviken are within this range (~2.5 and ~3 times lower than at Strömmen, respectively). However, despite Ingaröfjärden being located in a basin adjacent to Erstaviken (Fig. 1) and modelled as having an almost identical area-specific N retention capacity (Almroth-Rosell et al., 2016), denitrification rates were almost 20 times lower than those at Strömmen and ~8 to 6 times lower than at Baggensfjärden and Erstaviken, respectively. As such, N removal rates between adjacent basins may be more variable than assumed by models. The differences in rates are likely related to lower organic matter inputs and subsequent lower sediment respiration rates as indicated by deeper $O_2$ penetration at Ingaröfjärden (Table 2; Sup. Fig. 3). Suspended particulate organic matter may also be removed more quickly from Ingaröfjärden due to its more direct connection to the open Baltic Sea (Fig. 1) permitting more rapid water exchange and transport of particulate organic matter out of the basin than at Baggensfjärden and Erstaviken (Engqvist and Andrejev, 2003).

### 4.2.2 Controls on benthic $NO_3^-$ reduction

Given the minor contributions of anammox and DNRA in these sediments at the time of sampling, we focus predominantly on the control of heterotrophic denitrification in Stockholm Archipelago sediments at the time of sampling. Heterotrophic denitrification in sediments is limited by both the availability of $NO_3^-$ and $C_{org}$.

In sediments, $NO_3^-$ is supplied from overlying water and/or from nitrification in the surface layers (coupled nitrification-denitrification; Seitzinger, 1988; Seitzinger et al., 2006). The relative importance of the two $NO_3^-$ sources to denitrification in coastal systems can be highly variable between locations and seasons (e.g. Seitzinger et al., 2006; Jäntti et al., 2011; Bonaglia et al., 2014). We observed a distinct positive correlation between rates of denitrification and bottom water $NO_3^-$ concentration (Fig. 9) indicating a high capacity of the sediments to reduce riverine $NO_3^-$ loads along the seaward gradient, as shown for other coastal systems of the Baltic Sea (Asmala et al., 2017). We additionally demonstrate that benthic nitrification provided the major proportion (~55-90 %) of $NO_3^-$ which was reduced in the sediments at all four sites (Table 5; Fig. 8), as has been demonstrated in previous studies and syntheses on coastal systems (e.g. see Seitzinger et al., 2006) and studies within the Baltic Sea (e.g. Silvennoinen et al., 2007; Bonaglia et al., 2014; Bonaglia et al., 2017; Hellemann et al., 2017). One of the highest contributions of nitrification to $NO_3^-$ production for denitrification (~85 %) was measured at Ingaröfjärden. At this site, the lowest overall denitrification rates and bottom water $NO_3^-$ concentrations were measured, despite the deep (18 mm) $O_2$ penetration providing a large sediment volume for nitrification to occur (Table 2). This high $O_2$ penetration may in part be due to less $C_{org}$ inputs and thus a lower $C_{org}$ content (Table 2), discussed in section 4.2.1 and further below.

Inputs of $C_{org}$ provide both a C-source for heterotrophic processes (e.g. denitrification) as well as a source of $NH_4^+$ (from remineralisation processes) for nitrification and subsequent $NO_3^-$ production. In coastal sediments $C_{org}$ is not thought to limit denitrification. However, in complex basin systems such as the Stockholm Archipelago, and the Baltic Sea coastal zone in general, differences in ventilation and retention times between basins (implying differences in vertical and lateral exchange of water and $O_2$, and hence, variations in bottom water $O_2$) may mean that $C_{org}$ inputs are more variable than assumed (see section 4.2.1). Available $C_{org}$ in Ingaröfjärden (Table 2) may be less labile than at other sites due to such variations in hydrology and bottom water $O_2$, with the deep (18 mm) $O_2$ penetration indicating a lower organic matter reactivity and sediment respiration compared to the other sites. Lower labile $C_{org}$ availability will limit heterotrophic denitrification and may explain why anammox, an autotrophic process, is more dominant at this site (Table 5; Fig. 7). The presence of the invasive polychaete *Marenzelleria* (Table 2) may also reduce N removal at Ingaröfjärden and enhance the efflux and transport of $NH_4^+$ from sediments (e.g. Hietanen et al., 2007; Bonaglia et al., 2013), although it should be noted that the impacts of *in fauna* on N cycling are notoriously complex (Robertson et al., 2019).

### 4.2.3 Seasonal cycles of N processes

Sampling and experiments in this study were carried out in late winter (March), a period in the Baltic Sea when the water column is well mixed, with cold and well oxygenated bottom waters and with persistently low organic inputs to sediments. However, conditions are of course not static throughout the annual cycle. Seasonal warming, stratification, phytoplankton blooms and consumption and release of nutrients as seen in year-round monitoring data (Fig. 3d; Sup. Fig. 1) will have marked effects on sediment nutrient cycling. Year-round bottom water monitoring data collected at Bäggensfjärden show that $NO_3^-$ accumulates annually in bottom waters during the autumn and winter months before being consumed during spring and summer by phytoplankton blooms (Fig. 3d). Hypoxic bottom waters develop over summer following bloom collapse and subsequent enhanced deposition of fresh organic matter and enhanced benthic respiration during summer and early autumn. Bottom water total N concentrations increase during summer in connection with the hypoxic events (Fig. 3d) due to enhanced benthic remineralization and subsequent $NH_4^+$ efflux from sediments.

Increased organic inputs following the spring bloom are likely to lead to increases in denitrification as the season progresses, as is commonly observed in coastal sediments (e.g. Piña-Ochoa and Álvarez-Cobelas, 2006; Jäntti et al., 2011; Bonaglia et al., 2014). Thus, a similar scenario would be assumed for the Stockholm Archipelago as for other estuaries, leading to higher rates of denitrification during spring and early summer and a reduction in autumn and winter as organic inputs subside (e.g. Bonaglia et al., 2014). Depending on the bloom intensity and organic matter inputs during spring, increased benthic respiration may lead to more reduced conditions in surface sediments as bottom water $O_2$ is depleted. The availability of $NO_3^-$ also declines under hypoxic/anoxic conditions due to $NO_3^-$ consumption in the water column, lower $O_2$ penetration and thus a reduced volume of surface sediment where nitrification can occur and from the reduced efficiency of nitrification under low $O_2$ conditions. The resulting high C/N conditions may cause process dominance to shift from N removal by denitrification (or anammox) to retention by DNRA (e.g. An and Gardner, 2002; Burgin and Hamilton, 2007; Giblin et al., 2013; Algar and Vallino, 2014; Kraft et al., 2014), as has been repeatedly demonstrated in field, laboratory and model studies (An and Gardner, 2002; Algar and Vallino, 2014; Kraft et al., 2014; van den Berg et al., 2016; Kessler et al., 2018). Thus, under hypoxic conditions in summer/autumn,

DNRA may become the dominant $NO_3^-$-reducing process, altering the role of sediments from a $NO_3^-$ sink through $N_2$ production, to a source via increased $NH_4^+$ release by DNRA.

While we have not assessed $NO_3^-$-reducing process over different seasons at these four sites, we have demonstrated the microbial metabolic potential for DNRA is present through the detection of DNRA activity in incubations at all four sites (Table 5). We suggest that it is highly likely that DNRA contributes to $NH_4^+$ efflux at sites during sporadic bottom water hypoxia. Thus, the capacity for N removal by denitrification may be reduced during bottom water hypoxia while the likelihood of N recycling by DNRA increases as shown in previous Baltic Sea studies (e.g. Jäntti et al., 2011; Jäntti and Hietanen, 2012; Bonaglia et al., 2014).

**4.3 Implications for future water quality in the Stockholm Archipelago**

Continued decreases in nutrient inputs to the Baltic Sea (Gustafsson et al., 2012; Andersen et al., 2017) and the Stockholm Archipelago (Karlsson et al., 2010) are likely to reduce phytoplankton growth, lead to reduced organic matter input into the sediments and, eventually, to higher $O_2$ concentrations in bottom waters.

Our results indicate that increases in bottom water $O_2$ would likely impede the observed present-day P recycling pattern at the seasonally hypoxic sites (Fig. 3c), allowing thicker Fe-oxide bearing layers and a larger Fe-bound P pool in the surface sediments (e.g. Slomp et al., 1996), hence a larger (semi-permanent) surface sedimentary P pool. This process will, however, be delayed due to the prior deposition of organic rich sediments which results in a high upward flux of $H_2S$ (Table 3), i.e. legacy of hypoxia hindering the formation of Fe-oxides that can bind P. Because of this legacy effect, we expect that artificial reoxygenation of bottom waters (e.g. Stigebrandt et al., 2015), if applied in the Stockholm Archipelago, is unlikely to be a long-term effective measure towards improving the water quality since it does not stimulate permanent P burial in these sediments and a large impact on the Fe-P pool is hindered by the high upward $H_2S$ flux. Further nutrient reduction for the Stockholm Archipelago is expected to eventually lead to a reversal from export of P to the open Baltic Sea to import of P from the open Baltic Sea (Savchuk, 2005; Almroth-Rosell et al., 2016). This shows that improvement of the water quality in the Stockholm Archipelago is to a great extent coupled to nutrient management strategies for the entire Baltic Sea.

Our results indicate that in the Stockholm Archipelago, N likely goes through cycles of retention and removal throughout the year in relation to bottom water hypoxia. N is removed by denitrification during colder months when $NO_3^-$ availability is high, while DNRA is likely to increase during hypoxic, $NO_3^-$-depleted months. Reductions in the frequency of hypoxic bottom waters will thus reduce the amount of time that sediments potentially recycle bioavailable N via DNRA and sediments may be more likely to act as a net sink for N through denitrification on an annual basis.

Continued recovery of the Stockholm Archipelago is also likely to lead to (re-)colonisation by bioturbating macrofaunal populations that have been driven out by hypoxic bottom waters (Diaz and Rosenberg, 2008; Voss et al., 2011). This may enhance temporary P burial and denitrification by sediment reworking and oxygenation (e.g. Pelegri and Blackburn, 1995; Laverock et al., 2011; Norkko et al., 2012). While we still lack the predictive capabilities required to allow us to assess how fauna may influence sediment biogeochemistry (Griffiths et al., 2017; Robertson et al., 2019), reductions in nutrient inputs and phytoplankton bloom intensities, and eventual recolonization by fauna at inner archipelago sites will likely sustain active P and N removal processes. Thus,

these coastal sediments are likely to continue to contribute to removal of P and N as long as we continue to actively reduce nutrient inputs.

### 5. Conclusion

Seasonally hypoxic sites in the Stockholm Archipelago are characterized by active sedimentary P recycling, because low bottom water $O_2$ concentrations seasonally destabilize Fe-oxides that bind P in the surface sediments. A high upward flux of $H_2S$, due to prior deposition of organic rich sediments in a low $O_2$ setting, leads to the formation and preservation of $FeS_x$ instead of burial of Fe-oxides at these sites. At the site where bottom waters are well-oxygenated year round, the surface sedimentary P pool is mainly characterized by P

bound to Fe-oxides and organic matter, in a pool that is 5 times larger than that at the most hypoxic site (~0.172 versus ~0.036 mol P m$^{-2}$). At depth, sedimentary P is dominated by P in organic matter and apatite. Only for the site in the inner Archipelago (Strömmen), there is an indication for sink-switching, i.e. authigenic formation of a vivianite-type Fe(II)-P mineral, at depth. Burial rates of P at our sites in the Stockholm Archipelago are high (0.03-0.3 mol m$^{-2}$ y$^{-1}$) because of the combined effect of high sediment accumulation rates and high sedimentary

concentrations of P. Benthic denitrification is the primary $NO_3^-$-reducing pathway in the Stockholm Archipelago leading to remediation of $NO_3^-$ introduced from the water column and from benthic nitrification. Decreases in denitrification rates follow the gradient of bottom water $NO_3^-$ and sedimentary $C_{org}$ content from the inner archipelago towards the open Baltic Sea from ~1700 to ~100 μmol N m$^{-2}$ d$^{-1}$. Combining our process measurements with available monitoring data, it is likely that N in the Stockholm Archipelago undergoes

seasonal cycles of removal through denitrification/anammox and recycling by DNRA. Further reductions in P and N inputs are expected to reduce the frequency of hypoxic events. Our results show that the permanent burial of P is largely independent from bottom water redox conditions. Increased bottom water oxygen is expected to allow benthic denitrification to be sustained. Hence, we expect that the sediments in the Stockholm Archipelago will continue to remove part of the P and N loads upon reduction of such loads.

**Code and data availability**

Monitoring data can be extracted from the SHARK database at http://www.smhi.se/klimatdata/oceanografi/havsmiljodata/marina-miljoovervakningsdata. All other data, if not directly available from the tables and supplement, will be made available in the PANGAEA database. In the meantime data is available upon request to the authors.

**Supplement**

The supplement related to this article is available online at:

**Author contribution**

NvH, ER, DC, and CS designed the research. NvH, ER, MH, CH, WL and CS carried out the fieldwork. NvH, ER, MH, JK and WL performed the analyses. All authors interpreted the data. NvH, ER and CS wrote the paper

with comments provided by DC, MH, CH, JK and WL.

**Competing interests**

The authors declare that they have no conflict of interest.

**Acknowledgements**

We thank the captain and crew of the *R/V Electra*, Laurine Burdorf and Katharina Theopold for their help during the research cruise in the Stockholm Archipelago in March 2017. Arnold van Dijk, Coen Mulder, Thom Claessen, Floor Wille, Alexander Dorgelo and Joyce Maine (Utrecht University) and Rosine Cartier (Lund University) are thanked for analytical assistance. We thank Volker Brüchert (Stockholm University) for lending the whole-core incubation equipment, Morten Larsen and Bo Thamdrup (University of Southern Denmark) for allowing us to borrow the gas mixer and the use of the GC-IRMS for $^{15}$N isotope analyses. This study was funded by the Swedish Agency for Marine and Water Management (Havs- och vattenmyndigheten, DNR 1960-2018) and by the BONUS COCOA project (grant #2112932-1), funded jointly by the European Union and Swedish Research Council for Environment, Agricultural Sciences and Spatial Planning (FORMAS), grant #2013-2056. This research was also supported by the Netherlands Organisation for Scientific Research (NWO) Vici grant #865.13.005 and the European Research Council under the European Community's Seventh Framework Programme (FP7/2007–2013)/ERC Starting Grant #278364, both awarded to C.P. Slomp. This work was carried out under the program of the Netherlands Earth System Science Centre (NESSC), financially supported by the Ministry of Education, Culture and Science (OCW).

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

**Tables**

**Table 1.** General study site characteristics.

| | Strömmen | Baggensfjärden | Erstaviken | Ingaröfjärden |
|---|---|---|---|---|
| Coordinates (DD°MM'SS") | 59°19'09"N 18°07'09"E | 59°18'36"N 18°19'24"E | 59°13'06"N 18°23'42"E | 59°13'20"N 18°27'01"E |
| Water depth (m) | 30 | 40 | 68 | 37 |
| Bottom water redox conditions* | Seasonally Hypoxic | Seasonally hypoxic Sometimes euxinic | Sporadically hypoxic | Oxic |
| Location in the Archipelago** | Inner | Intermediate | Intermediate | Intermediate |

* Bottom water redox conditions based on monitoring data (SMHI, 2019).

** Following the classification by Almroth-Rosell et al. (2016).

**Table 2.** Key site characteristics at time of sampling (March 2017).

| | Strömmen | Baggensfjärden | Erstaviken | Ingaröfjärden |
|---|---|---|---|---|
| Bottom water $O_2$ (mL $L^{-1}$) | 7.6 | 7 | 6.7 | 8.5 |
| $O_2$ penetration depth* (mm) | 2.1 | 1.9 | 3.6 | 18 |
| Diffusive uptake of $O_2$ (mmol $m^{-2}$ $d^{-1}$) | 13.4 | 13.8 | 7.3 | 3.1 |
| Bottom water salinity | 5.2 | 6.2 | 6.4 | 6.2 |
| Bottom water temperature (°C) | 1.5 | 2.4 | 2.2 | 1.3 |
| | Mud | Mud | Mud | Bioturbated |

| | | | | |
|---|---|---|---|---|
| Sediment type | | | | mud |
| Suboxic zone* (mm) | 4 | - | 15 | 25 |
| Macrofauna | None | None | None | *Marenzelleria* |

* Derived from high-resolution micro-electrode profiling (Sup. Fig. 3)**Table 3.** Diffusive fluxes of pore water $H_2S$.

| | Strömmen | Baggensfjärden | Erstaviken | Ingaröfjarden |
|---|---|---|---|---|
| Sediment top (cm) | 0.75 | 0.75 | 1.75 | 8.25 |
| Sediment bottom (cm) | 2.25 | 7.25 | 8.25 | 15 |
| $H_2S$ top ($\mu$mol L$^{-1}$) | 2 | 3 | 7 | 36 |
| $H_2S$ bottom ($\mu$mol L$^{-1}$) | 385 | 899 | 1111 | 1340 |
| Diffusive flux (mmol m$^{-2}$ d$^{-1}$) | 7.6 | 4.2 | 5.2 | 6.5 |

**Table 4.** Sedimentary concentrations of organic carbon ($C_{org}$), nitrogen (N), phosphorus (P) and calcium carbonate ($CaCO_3$) and C/N and $C_{org}/P_{tot}$ ratios for the different study sites.

| | Depth interval (cm) | Strömmen | Baggensfjärden* | Erstaviken* | Ingaröfjärden* |
|---|---|---|---|---|---|
| $C_{org}$ avg. (wt. %) | 0-2 | 7.9 | 6.3 | 6.0 | 5.1 |
| $C_{org}$ avg. (wt. %) | 10-40 | 6.3 | 4.5 | 4.5 | 3.8 |
| $CaCO_3$ avg. (wt. %) | Entire core | 2.5 | 2.3 | 2.4 | 2.9 |
| N avg. (wt. %) | 0-2 | 0.99 | 0.83 | 0.78 | 0.69 |
| N avg. (wt. %) | 10-40 | 0.59 | 0.54 | 0.54 | 0.48 |
| P avg. (wt. %) | 0-2 | 0.36 | 0.17 | 0.19 | 0.25 |
| P avg. (wt. %) | 10-40 | 0.14 | 0.10 | 0.11 | 0.11 |
| C/N avg. (mol$^{-1}$ mol$^{-1}$) | 0-2 | 9.4 | 8.9 | 9.0 | 8.7 |
| C/N avg. (mol$^{-1}$ mol$^{-1}$) | 10-40 | 12.4 | 9.6 | 9.8 | 9.1 |
| $C_{org}/P_{tot}$ avg. (mol$^{-1}$ mol$^{-1}$) | 0-2 | 69 | 96 | 95 | 53 |
| $C_{org}/P_{tot}$ avg. (mol$^{-1}$ mol$^{-1}$) | 10-40 | 116 | 116 | 108 | 88 |

*Organic carbon and nitrogen concentrations for Baggensfjärden, Erstaviken and Ingaröfjärden are derived from

van Helmond et al. (in review).

**Table 5.** Areal rates of benthic $NO_3^-$-reducing processes, including standard error (SE). 'Nitrification-denitrification' indicates the proportion of denitrification supported by $NO_3^-$ from nitrification (as opposed to water column $NO_3^-$). Bottom water $NO_3^-$ concentrations and $NH_4^+$ and $NO_3^-$ fluxes from the surface sediments into the water column (calculated from pore water profiles), including standard error (SE).

| | Strömmen | Baggensfjärden | Erstaviken | Ingaröfjärden |
|---|---|---|---|---|
| Denitrification - ($\mu$mol N m$^{-2}$ d$^{-1}$) | 1723 | 685 | 564 | 90 |
| (SE) | (774) | (58) | (86) | (38) |
| DNRA - ($\mu$mol N m$^{-2}$ d$^{-1}$) | 11.1 | 6.1 | 3.6 | 2.8 |
| (SE) | (8.1) | (1.7) | (3.3) | (0.4) |
| Nitrification-denitrification - ($\mu$mol N m$^{-2}$ d$^{-1}$) (SE) | 1027 | 500 | 500 | 76 |
| | (461) | (42) | (76) | (32) |
| Nitrification-denitrification (%) | 59.6 | 73 | 88.6 | 84.4 |
| Anammox - ($\mu$mol N m$^{-2}$ d$^{-1}$) | 0.27 | 0.76 | 3.11 | 44.12 |
| (SE) | (0.1) | (0.1) | (0.5) | (18.4) |
| $N_2$ anammox (%) | 0.02 | 0.11 | 0.55 | 32.93 |
| Bottom water $NO_3^-$ ($\mu$mol L$^{-1}$) | 17.8 | 12.1 | 9.0 | 5.6 |
| $NH_4^+$ flux - ($\mu$mol N m$^{-2}$ d$^{-1}$) | 1399 | 629 | 600 | 0 |
| (SE) | (122.4) | (88.8) | (76.8) | (0) |
| $NO_3^-$ flux - ($\mu$mol N m$^{-2}$ d$^{-1}$) (SE) | 4.1 | -1.44 | -7.68 | -85.7 |
| | (0.05) | (1.0) | (0.24) | (35.0) |

**Table 6.** Burial rates of total and reactive P.

| | Unit | Strömmen | Baggens-Fjärden | Erstaviken | Ingarö-fjärden |
|---|---|---|---|---|---|
| Total P burial rates | (mol m$^{-2}$ yr$^{-1}$) | 0.28 | 0.03 | 0.09 | 0.05 |
| | (g m$^{-2}$ yr$^{-1}$) | 8.74 | 0.87 | 2.89 | 1.53 |
| Reactive P* burial rates | (mol m$^{-2}$ yr$^{-1}$) | 0.24 | 0.02 | 0.07 | 0.03 |
| | (g m$^{-2}$ yr$^{-1}$) | 7.47 | 0.70 | 2.22 | 1.03 |
| Thickness enriched top layer** | (mm) | 30 | 20 | 20 | 40 |
| Total P burial in enriched top layer | (mol m$^{-2}$) | 0.29 | 0.08 | 0.10 | 0.38 |
| | (g m$^{-2}$) | 9.12 | 2.50 | 3.19 | 11.85 |
| Total P burial in enriched top layer - background | (mol m$^{-2}$) | 0.160 | 0.036 | 0.047 | 0.172 |
| | (g m$^{-2}$) | 4.96 | 1.11 | 1.47 | 5.33 |
| Reactive P burial in enriched top layer | (mol m$^{-2}$) | 0.24 | 0.067 | 0.081 | 0.32 |
| | (g m$^{-2}$) | 7.49 | 2.07 | 2.51 | 10.05 |
| React. P burial in enriched top layer- background | (mol m$^{-2}$) | 0.127 | 0.031 | 0.039 | 0.200 |
| | (g m$^{-2}$) | 3.94 | 0.94 | 1.20 | 6.21 |
| Sediment accumulation rate*** | (mm yr$^{-1}$) | 35 | 5 | 15 | 5 |
| | (g DW m$^{-2}$ yr$^{-1}$) | 6300 | 865 | 2588 | 1353 |

*Reactive P is the sum of Fe-bound P, Exch. P, Org. P and Auth. P

**See Fig. 9 for definition of top layer (red) and background (dashed line)

*** Sediment accumulation rates for Baggensfjärden, Erstaviken and Ingaröfjärden are based on $^{210}$Pb data from van Helmond et al. (in review), see Sup. Fig. 2. DW = dry weight sediment.

**Figures**

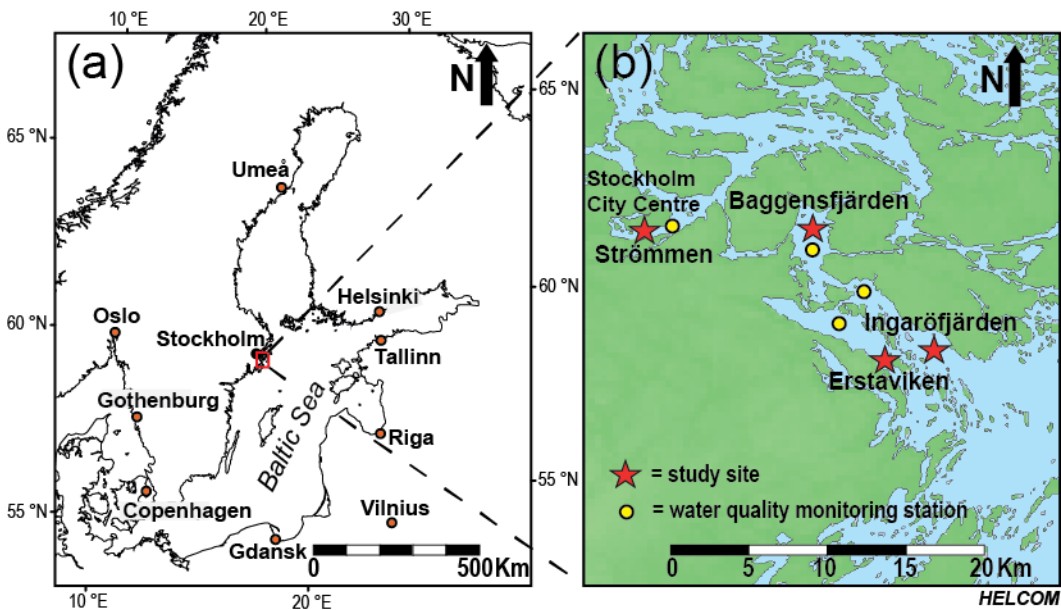

**Figure 1.** The Baltic Sea (Ning et al., 2016), with the study area in the Stockholm Archipelago indicated by the
red box (a). Detailed map of the southwestern part of the inner and intermediate Stockholm Archipelago (cf.
Almroth-Rosell et al., 2016). Red stars indicate the locations of the study sites: Strömmen, Baggensfjärden,
Erstaviken and Ingaröfjärden. Yellow dots indicate the locations of the water quality monitoring stations
(SMHI, 2019) most proximate to the sites in this study (b).

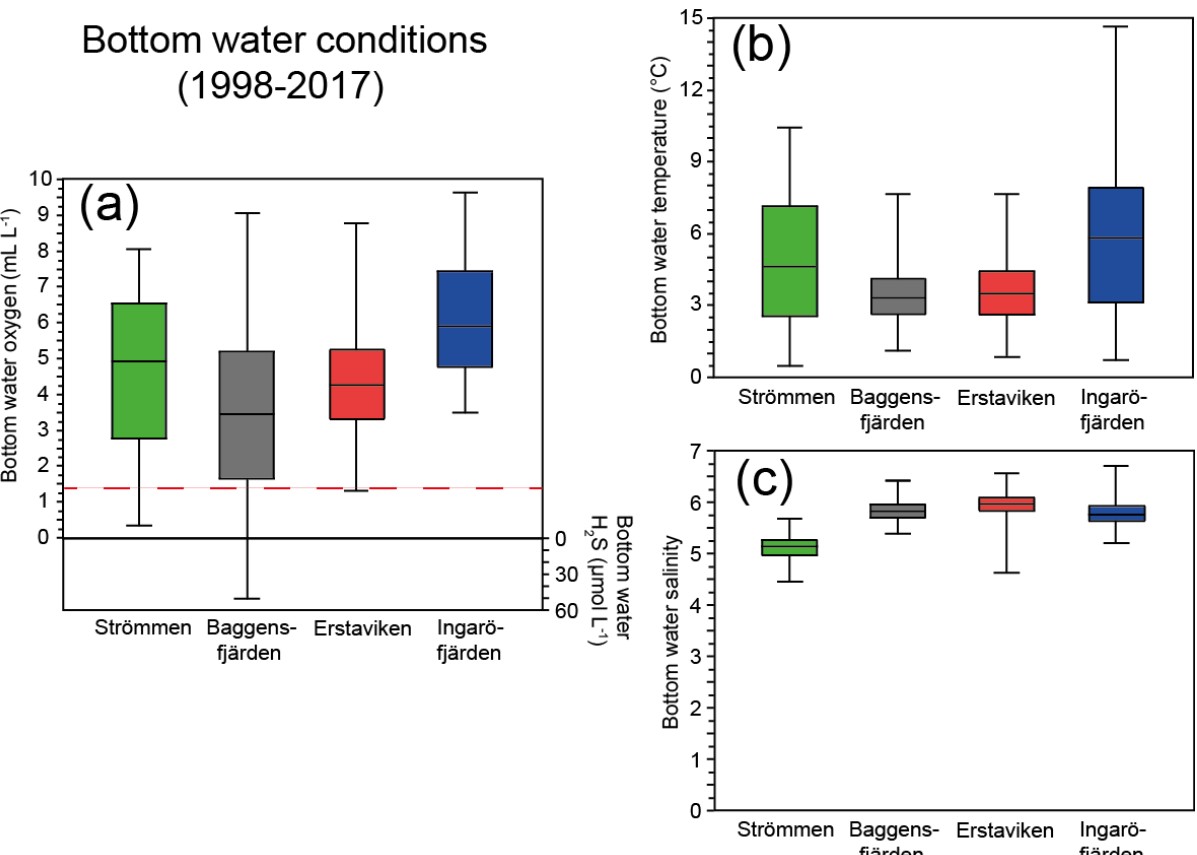

**Figure 2.** Ranges in bottom water oxygen and sulfide (a), temperature (b) and salinity (c) over the last 20 years (1998-2017) water quality monitoring stations (SMHI, 2019) (Fig. 1) most proximate to the study sites. The solid line between the boxes is the median, whereas the boxes represent the second and third quartiles. The error bars indicate the minimum and maximum value recorded for the displayed period. The red dashed line (located at 1.4 mL L$^{-1}$; Fig. 2a) indicates the hypoxic boundary.

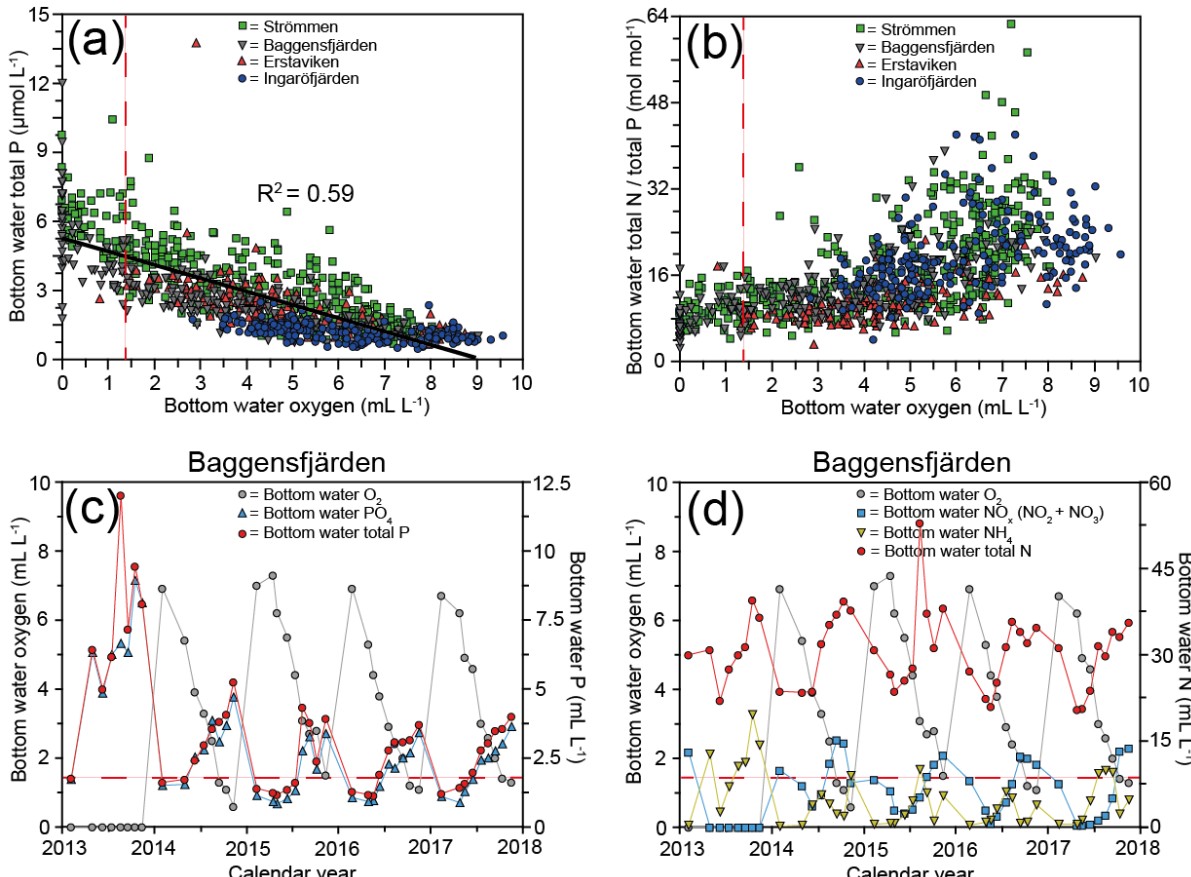

**Figure 3.** Bottom water dissolved oxygen plotted against total P, with the black line depicting the inverse linear relationship between the two (a) and total N/total P (b) for the water quality monitoring stations (SMHI, 2019) (Fig. 1) most proximate to the study sites. Bottom water dissolved oxygen and bottom water P (c) and N (d) for Baggensfjärden from 2013 until 2017. The red dashed line (located at 1.4 ml L$^{-1}$) indicates the hypoxic boundary in all panels.

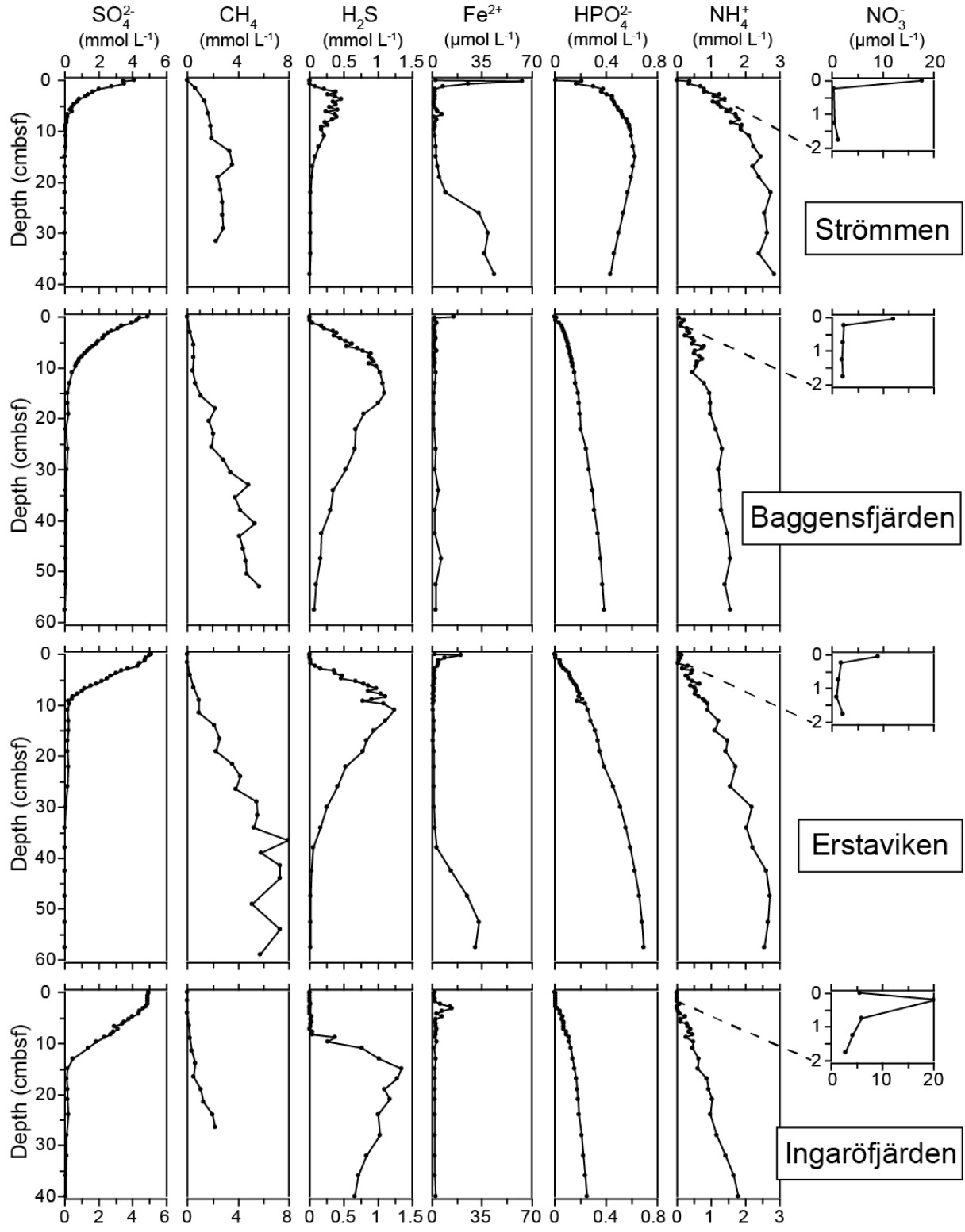

**Figure 4.** Pore water depth profiles of $SO_4^{2-}$, $CH_4$, $H_2S$, $Fe^{2+}$, $HPO_4^{2-}$, $NH_4^+$ and $NO_3^-$ at the sites in the Stockholm Archipelago: Strömmen, Baggensfjärden, Erstaviken and Ingaröfjärden.

1035

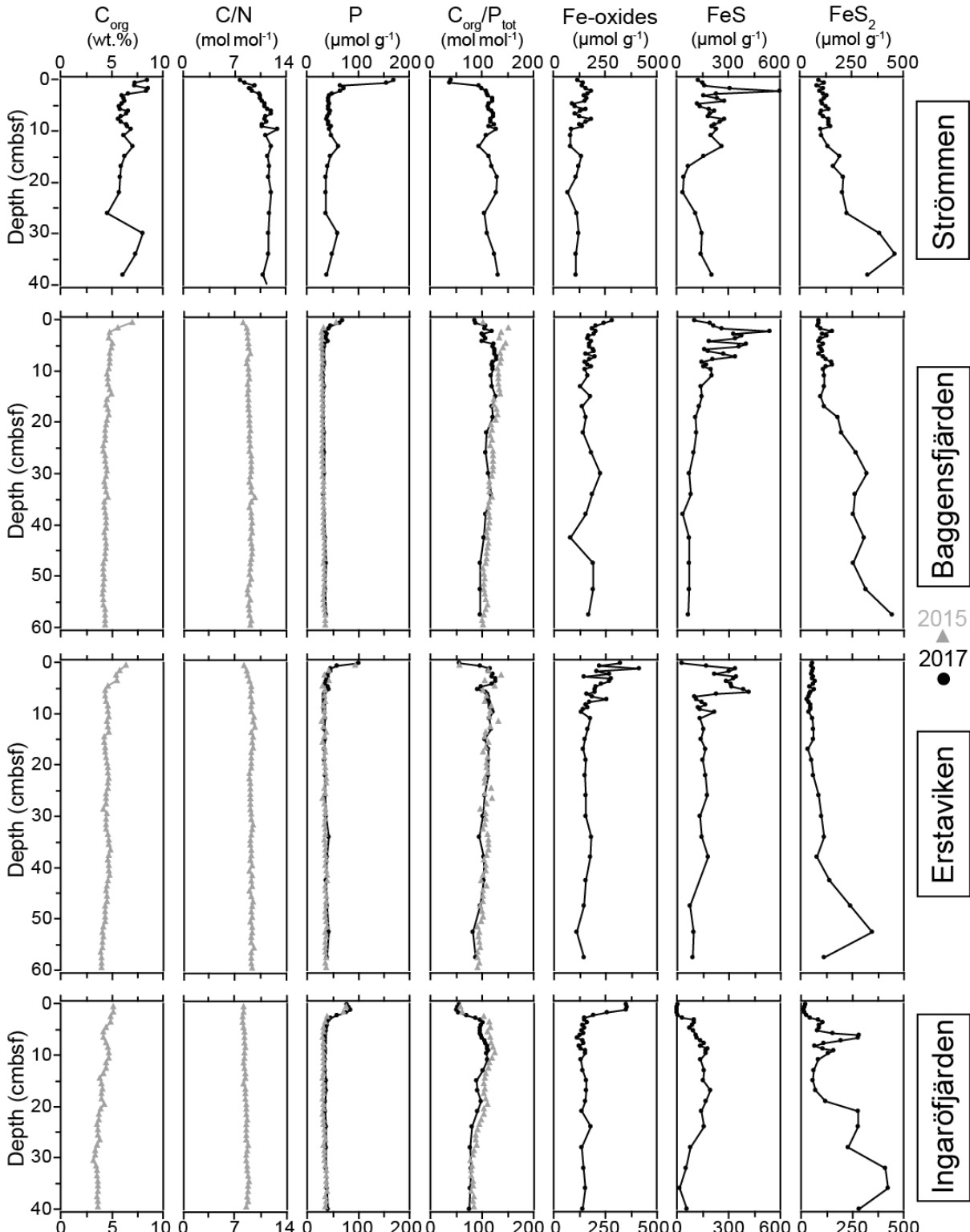

**Figure 5.** Solid phase depth profiles of $C_{org}$, C/N, P, $C_{org}/P_{tot}$, Fe-oxides, FeS (AVS-derived) and $FeS_2$ (CRS-derived) for the study sites in the Stockholm Archipelago: Strömmen, Baggensfjärden, Erstaviken and Ingaröfjärden. Grey triangles are data from van Helmond et al. (in review).

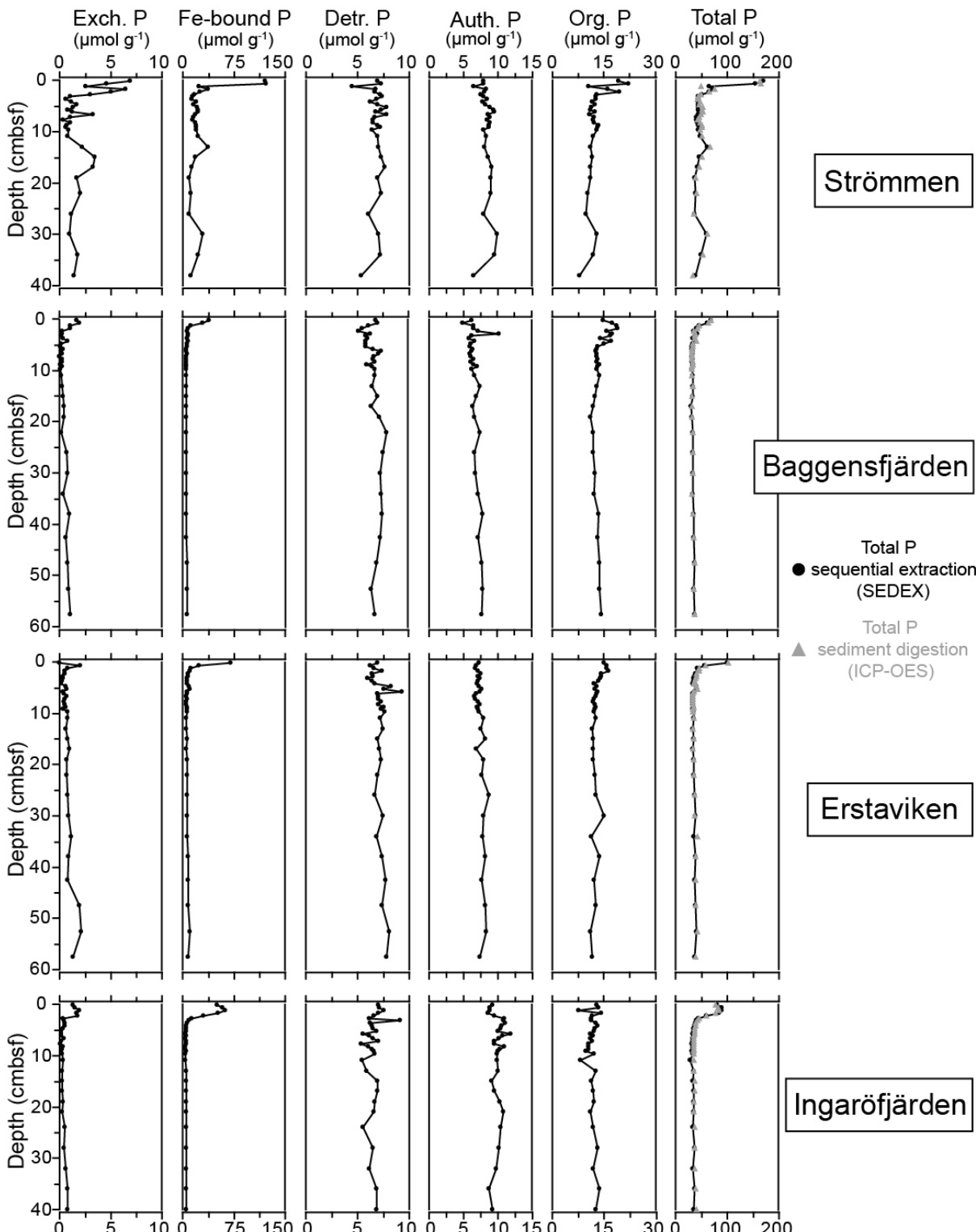

**Figure 6.** Depth profiles of the different fractions of solid phase P for the study sites in the Stockholm Archipelago: Strömmen, Baggensfjärden, Erstaviken and Ingaröfjärden. Total P is the sum of the different sequentially extracted P phases (SEDEX; black dots) and the P content derived from acid digested sediment aliquots and subsequent ICP-OES analysis for the sediment samples taken in March 2017 (grey triangles).

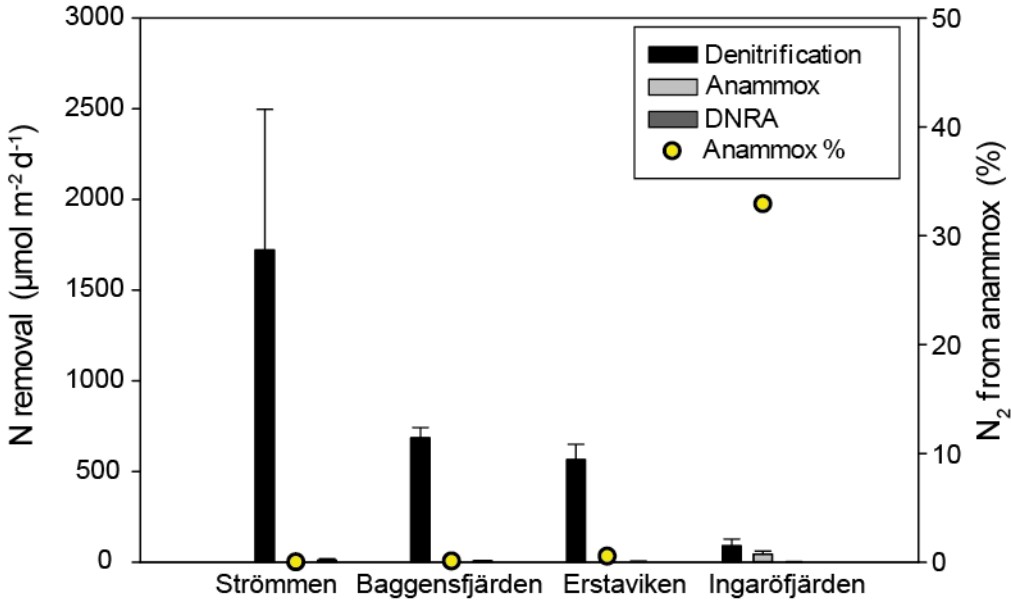

**Figure 7.** Bar diagram showing the areal rates of benthic $NO_3^-$-reducing processes, including error bars. Relative contribution of anammox is indicated by the yellow dots.

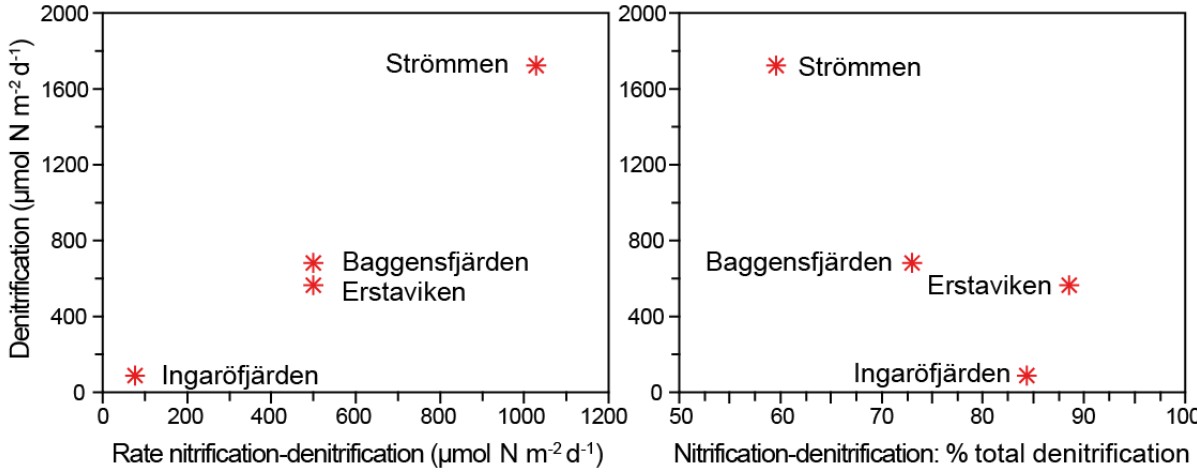

**Figure 8**. Relationship between total denitrification rates and denitrification driven by $NO_3^-$ from nitrification (nitrification-denitrification) as process rates (left) and as a percentage of total denitrification (right).

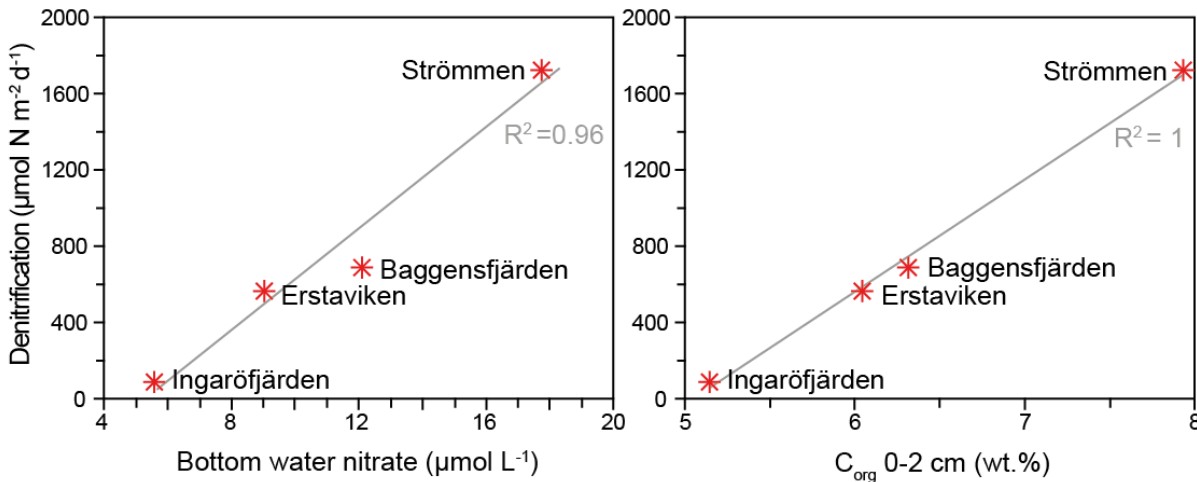

**Figure 9.** Relationship between denitrification and bottom water $NO_3^-$ concentrations, and upper sediment $C_{org}$ content for the study sites in the Stockholm Archipelago.

1060

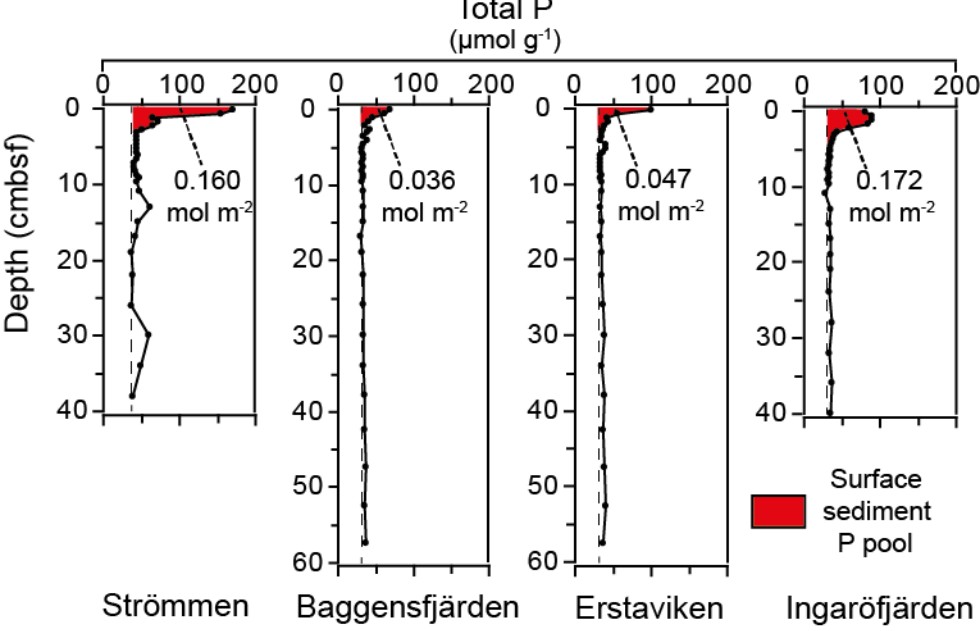

**Figure 10.** Surface sedimentary P pools for the study sites in the Stockholm Archipelago. The red color indicates the enriched surface sediment layer, or "top layer" (Table 6). Dashed lines indicate "background" sedimentary P.

1065