# Peer review of "Removal of phosphorus and nitrogen in sediments of the eutrophic Stockholm Archipelago, Baltic Sea"

_Biogeosciences, 2019_

## Referee Comment (RC1) · Emil Rydin (Referee) · 2 Dec 2019

Efficient removal of phosphorus and nitrogen in sediments of the eutrophic Stockholm Archipelago, Baltic Sea

By Niels A.G.M. van Helmond1,2a,b, Elizabeth K. Robertson2,3a, Daniel J. Conley2, Martijn Hermans1, Christoph Humborg4, L. Joëlle Kubeneck1c, Wytze K. Lenstra1 and Caroline P. Slomp1

The manuscript addresses the critical process of sediment P burial for the development of eutrophication. It also covers nitrogen turnover on which I am not an expert. My

review will therefore focus on the P.

The burial of P in accumulation bottom areas in four sites is presented with high-quality data, both regarding the accumulation rate of matter and the P content and P-forms in these layers. I find the manuscript well written in terms of language and easy accessible. I do, however get the impression that the main scope with the P investigation partly was something more than the burial as it is presented; a "sink-switch" process to e.g vivianite-formation, as evident from the many references included covering that possible process.

Main concerns

The outcome of the study regarding the P burial ends up rather basic by summing up total-P concentrations in deeper sediment layer with the sediment accumulation rate at the specific site. With all the supporting data presented, perhaps could new insights be developed?

I suggest a more in-depth analysis of the P burial both in a spatial (quantitative) scale, as well as in a qualitative (formation of refractory P that forms during diagenesis and resists it) perspective. I offer my reflections on the subject as a platform to develop the discussion:

Does the lack of concentration changes in most P forms with sediment depth (Fig. 6) actually reflect mainly inert P forms settling out on the sediment surface, resistant to sediment diagenesis? Even the "authigenic P" (Ca-P) seems thus to have been formed elsewhere than in the present sediment profile, since it already in the top sediment layers is present at a concentration it will remain throughout the sediment profile. The "sink-switch" process seems to be virtually absent (except perhaps for the Strömmen station).

Indeed, some share of the organic P settling out on the sediment surface is mineralized, as evident from the decline in org-P concentration in the top ca 5 cm. This is

well presented in Fig 9 where the "background" concentration is indicated. Perhaps the "top-layer" (indicated as red labelled "surface sediment sink") actually represents the P active in turn-over, as suggested in Rydin 2011. All the P indicated as background concentration would then largely be inert P forms, not relevant for the eutrophication process. A key question would then be to what extent autochthonous organic P (e.g. plankton) contributes to the supply of organic sediment P resistant enough to get permanently buried. Is the only main sink-switch of importance in this region the transformation of dissolved P in the water column to organic P (plankton), to a larger or lesser extent permanently buried in the sediment?

Line 391: Only "near shore construction and dredging (line 391) is presented as alternative sources for the P accumulated than land-derived. Another explanation for the high burial rate could be that the sediment to a large extent consists of old sediment (old clays) that already have undergone sediment P diagenesis processes one or even several times during the Baltic Sea life span, exposed to resuspension due to land up-lift (ca 0,5 cm/yr), and a secondary (or a third) settling out on accumulation bottom areas (Karlsson et al 2019 and references therein). But also P from the catchment (e.g. Lake Mälaren) of course contributes to this pool of P.

A back of the envelope calculation regarding the Lake Mälaren impact on the P burial in the region might look something like this: The Stockholm county archipelago covers a water area of 3100 km2. Assuming that only 25% of this area represents accumulation bottom conditions; 780 km2, and an average burial rate of 3 g/m2 yr, it ends up to 2340 ton P/yr. This is a high flux compared to external inputs, and corresponds to as much as ca 15% of the total P input to the Baltic proper. Thus, a high share of the P found in accumulation bottom sediment seems to be recycled.

To summarize my thoughts: the archipelago sediment seems effective in permanently trapping already inert, particulate P. But less efficient in transforming the "surface sediment sink" (Fig 9) into P forms that will be permanently buried.

The Discussion can be expanded with comparisons to other studies regarding accumulation of matter and phosphorus in the region; see Karlsson et al 2019 for suggestions on references for a more developed Discussion. Both the "surface sediment sink" (Figure 9), and the sediment accumulation rate, in Baggensfjärden and Erstaviken, are close to those found in a recent study in the adjacent Björnöfjärden (Rydin & Kumblad, 2019).

Specific comments

Figure 10 is of limited value since it (only) shows that the P concentration at depth is rather constant, and the burial rate is largely dependent on sediment accumulation rate (at the investigated sites).

The references need a check. Are all the references present needed? Line 802: Rydin et al 2011 reference is missing.

Line 900: It would be useful to present sediment accumulation rates as g DW/m2 yr besides cm/yr to compensate for the compacting of the surface sediment.

References Karlsson, M., Bryhn, A., Håkanson, L., Hållén, J., Jonsson, P., Malmaeus M, Rydin E. 2019. On the role of sedimentological processes controlling phosphorus burial in the coastal zone of the Baltic Sea. Limnology and Oceanography 9999:1-4.

Rydin, E. Kumblad, L. 2019. Capturing past eutrophication in coastal sediments – Towards water-quality goals. Estaurine, Coastal and Shelf Science. DOI: 10.1016/j.ecss.2019.02.046

Rydin, E., Malmaeus, M., Karlsson, M., Jonsson, P. (2011) Phosphorus Release From Coastal Baltic Sea Sediments As Estimated From Sediment Profiles. Estaurine, Coastal and Shelf Science. 92:111-117.

---

## Referee Comment (RC2) · Anonymous Referee #2 · 30 Dec 2019

**Review of "Efficient removal of phosphorus and nitrogen in sediments of the eutrophic Stockholm Archipelago, Baltic Sea"**
**by Helmond et al.**
**MS# bg-2019-376**

General comments/requests (in random order):

1) The authors present bottom water nitrate as one controlling factor of benthic denitrification. What about nitrate formed during benthic nitrification as one controlling factor? Very little is mentioned about it. Please do, and discuss nitrification as a control of benthic denitrification in these sediments.

2) Lines 421-436 and elsewhere: Denitrification rates decreased going seaward, and the authors explained this by lower bottom water nitrate concentrations and lower organic C content of sediment along the transect going seaward. However, it is generally assumed that in coastal and shelf sediments availability of nitrate controls denitrification rates since there often is no shortage of organic C in such sediments – at least not to limit benthic dentrification. Also, in coastal and shelf sediments, the nitrate consumed in denitrification is mostly produced during nitrification in the sediments rather than being nitrate from the bottom water (cf. e.g. papers by Seitzinger (et al.)). The 18 mm oxygen penetration depth in sediment of Ingaröfjärden should allow active nitrification so that the lower bottom water nitrate concentration there should not lead to a decreased denitrification rate. Could the authors please discuss in their paper this apparent inconsistency between what has been generally found and what was found in the present study?

3) The measurements were made in March, which is late winter. This period should be among the most oxygenated of the year; the vertical stratification is weak and mixing/ventilation of bottom waters should be facilitated. In addition, bottom water temperature is at its minimum, the spring bloom has not started yet, and there should be very little fresh organic matter in sediments. Can the authors please include a discussion in their paper on this and especially on to what extent the results presented are representative on an annual scale? In my opinion, the results represent a late winter situation, and fixed N removal and retention of P in sediments most likely are very different than in e.g. summer-fall when bottom water oxygen levels and quantity of fresh organic matter in sediments can be completely different. The authors should make this very clear in their paper.

4) Section 4.3 Implications: Although this section includes some interesting discussions, I cannot see that it is relevant in this paper. This section consists of discussions and speculations far beyond what can be found warranted based on the original results of this paper. This paper is not a review paper. Please focus the discussion, and the presentation of implications, on the results obtained in _this_ study (carried out in March 2017).

5) Section 4.3 Implications: "…artificial reoxygenation of bottom waters (e.g. Stigebrandt and Gustafsson, 2007) will not be a long-term effective measure towards improving the water quality of the (coastal) Baltic Sea." Please explain how the

results of _this_ paper justify this statement. If the authors cannot convincingly do this, this statement should be removed.

6) Lines 536-538: "Further reductions in P and N inputs are necessary to ensure a reduction in the frequency of hypoxic events. Eventually this will lead to a larger surface sedimentary P sink and will be key to maintaining the efficient N filter and avoiding additional P and N recycling."  I agree that this is one important measure to improve the environmental status of coastal systems and that it should be done, BUT please explain to what extent the results of _this_ study justify this final conclusion.

Other comments by line number:
Line 24-25: What other form(s) of P make up the remaining 50-70% (i.e. the major fractions) of P burial? Please make this clear already in Abstract.

Line 31-32: Regardless whether this statement in general is correct, what evidence does this paper provide that this statement is correct? What level of removal or retention of N and P would have occurred in the studied area if bottom waters had been better oxygenated (natural or manmade)? I do not think this statement is warranted based on the results this paper presents, so I suggest it is deleted unless the authors convincingly can argue that it is warranted. See also above.

Line 390: Remove "rate" in the beginning of this line.

Lines 412-413: "…and the increasing role of sediments as a $NO_3^-$ sink along the estuarine gradient". Please clarify this text. In which direction are you meaning the estuarine gradient goes? Landward or seaward?

Lines 414-415: "…reduction in organic matter quality as shown by a concomitant reduction in surface sediment N and organic C contents".  Does organic matter _quality_ necessarily go down when _contents_ of N and organic C go down? Please explain.

Lines 440-451 (and later in Discussion): What did Bonaglia et al. (2017; BG) report on DNRA and its importance as a nitrate reducing process in Baltic sediments?

Lines 534-535 (and elsewhere): "Combining our process measurements with available monitoring data, it is likely that N in the Stockholm Archipelago undergoes seasonal cycles of removal and retention." What do you mean with "retention" of N here? Did you intend to write _recycling_ (e.g. through DNRA)?

Table 4 head. "DN is nitrate supplied from nitrification". DN does not exist in the Table, so why is it mentioned in the Table head?

Table 4: Annamox is incorrectly spelled. Should be anammox.

---

## Referee Comment (RC3) · Anders Stigebrandt (Referee) · 9 Jan 2020

This manuscript presents vertically highly resolved observations in bottom sediments of the Stockholm Archipelago with the aim to quantify removal of phosphorus and nitrogen by sediment processes. The net fluxes of P and N through the sediment-water interface determines the removals of P and N (the sediment sinks). These are very important at the system level and determine together with the nutrient supplies from land-based and ocean-based sources concentrations in the water column. However, in the manuscript the estimated sediment sinks are not upscaled (i.e. horizontally integrated) to the system level which makes it impossible to verify that the removal of

P and N in the sediments is efficient as postulated in the title of the manuscript. The manuscript needs thorough revision as discussed below.

Line 29-31. This is not shown in the manuscript, see comments on Line 514-515 below.

Line 31-32. This is not shown in the manuscript and moreover it is wrong as claimed in this interactive comment, see comments on Line 479-481 and Line 481-483 below.

Line 85. There are also strong signs of increasing eutrophication in the Baltic Sea with large and increasing volume of anoxic water and corresponding large and increasing area of anoxic bottoms (Hansson et al. 2019). See also the comment on Line 475 below.

Line 96. How is P removal defined? Is P removal = P burial – P reflux? P reflux is not quantified in the manuscript. Therefore, P removal is not determined. Please clarify.

Line 121. This is the so-called land-based supply. But there is also a sea-based supply by inflowing surface water from the open Baltic Sea. How large is the sea-based supply? This is Important for the calculation of the filter effect mentioned on Line 17, see also comments regarding Line 514-515 below.

Line 358. What is meant by hydrological restrictions? Do you mean restrictions in the water exchange due to vertical density stratification and topographical restrictions like sills? Please clarify.

Line 368-369. The small annual amplitude of P and $O_2$ in the bottom water of Ingaröfjärden is said to be due to a nearly absent seasonal P recycling. However, it is more likely due to an efficient water exchange (flushing) throughout the year. Please explain why you discard the effect of efficient flushing throughout the year. By the way, it would be fine if the sill depths for the four basins considered could be mentioned in the manuscript. Knowing these helps to interpret the flushing of the deepwater of the basins.

Line 390. Is P burial = P removal? If this is the case, P reflux=0. Please explain. See

also comments to Line 96 above.

Line 432. Is the deeper O2 penetration at Ingaröfjärden due to the action of Marenzel-leria?

Line 475 Continued decrease of the land-based P input to the Baltic proper has not led to reduced horizontally integrated P concentration c in the surface layer in winter. On the contrary c has increased by at least 25% since the 1980s although the land-based supply has been approximately halved (e.g. Stigebrandt, 2018). The input of organic matter into the sediments has thus rather increased. The area of anoxic bottoms increased by a factor of about 6 from the period before 1999 to the period after 1999 and attained its highest value in 2018 (Hansson et al., 2019). This should be discussed in the manuscript.

Line 479 – 485. The response of water column concentrations above the sediments to the sediment processes are not quantified in the present manuscript. However, there is an exception to this. This is the statement that artificial reoxygenation of bottom waters will not be a long-term effective measure towards improving the water quality of the (coastal) Baltic Sea. There is no analysis in the manuscript that supports this statement. As shown below, the statement is wrong and should be removed from the manuscript.

Citation from Line 479 – 481. "Increases in bottom water O2 would likely impede the observed present-day P recycling pattern in the seasonally hypoxic sites (Fig. 3c), allowing thicker Fe-oxide bearing layers and a larger Fe-bound P pool in the surface sediments (e.g. Slomp et al., 1996), hence a larger (semi-permanent) surface sedimentary P sink."

The thickness of the Fe-oxide bearing layers is determined by the oxygen penetration depth L. Cai and Sayles (1996) presented the following relationship between oxygen penetration depth L, benthic oxygen flux FO2 across the sediment-water interface and bottom water oxygen concentration $[O_2]_{bw}$:

[Figure]

L=2θDs [O2]bw)/FO2 (Equation 1)

Here $\theta$ and Ds are the porosity and diffusivity of O2 in sediment, respectively.

Equation (1) shows that the thickness L of the oxidized layer on top of the sediment varies with [O2]bw and, allowing for some inertia, the minimum thickness L=Lmin should occur approximately when [O2]bw attains its minimum. This means that Lmin can be increased by increasing the minimum bottom water oxygen concentration [O2]bw which is in accordance with the statement on Line 479-481.

However, the following statement (on Line 481-483) is presented without any proof of its validity for the Baltic Sea.

Citation from Line 481-483. "This process will, however, be delayed due to the prior deposition of organic rich sediments which results in a high upward flux of H2S (i.e. legacy of hypoxia) hindering the formation of Fe-oxides."

This statement is maybe true for highly eutrophic lakes, but it is not true for the deepwater sediments in the much less eutrophic Baltic Sea, as discussed on p. 41 in Stigebrandt (2018). Using Sediment Profile Imagery (SPI) it was observed that the sediment surface was oxygenated within a couple of months during a natural oxygenation event due to a Major Baltic Inflow (Rosenberg et al., 2016). This means that the upward flux of H2S in the Baltic Sea deepwater sediments is not large enough to hinder the formation of an oxic layer (containing Fe-oxides) on top of the sediment when the bottom water is oxygenated. Therefore Equation (1) is applicable to the deep sediments of Baltic Sea. The oxygen penetration depth L can thus be increased by increasing [O2]bw by artificial reoxygenation of the bottom waters of the Baltic Sea.

The major effect of oxygenation of anoxic bottom sediments is that it stops the outflow of P from the sediment. This was discussed in Stigebrandt et al. (2014), see also Almroth-Rosell et al. (2015) who show that the phosphorus release rate from the sediment drastically decreased and even became negative as a result of Major Baltic Inflows. As shown in Stigebrandt (2018), artificial reoxygenation of bottom waters should be a rapid and long-term effective measure towards reducing the eutrophication and improving the water quality of the open Baltic Sea and coastal areas with good water exchange with the open sea so that local effects of local land-based nutrient supplies are small. This disproves the following statement (on Line 483-485) in the manuscript.

Citation from Line 483 - 485 "This also explains why artificial reoxygenation of bottom waters (e.g. Stigebrandt and Gustafsson, 2007) will not be a long-term effective measure towards improving the water quality of the (coastal) Baltic Sea."

Line 516. What is meant by "control" in the sentence "continue to actively reduce and control nutrient inputs"

Line 514-515. In the manuscript it is postulated but not shown that the sediments are efficient filter. This would require that estimates of the N and P sinks (tonnes year-1) for the whole area were related to the total supply of nutrients (tonnes year-1), i.e. the supply from both land-based and sea-based sources.

References

Almroth-Rosell, E., Eilola, K., Kuznetzov, I., Hall, P.O.J., and Meier, H.E., 2015. A new approach to model oxygen dependent benthic phosphate fluxes in the Baltic Sea. Journal of Marine Systems, 144, 127-141.

Cai, W.J., and Sayles, F.L., 1996. Oxygen penetration depths and fluxes in marine sediments. Mar. Chem. 52, 123-131.

Hansson, M., Viktorsson, L., and Andersson, L., 2019. Oxygen survey in the Baltic Sea 2018 – Extent of anoxia and hypoxia, 1960 – 2018. SMHI, Report Oceanography No 65, 11 pp + 2 Appendices.

Rosenberg, R., Magnusson, M., Stigebrandt, A., 2016: Rapid re-oxygenation of Baltic Sea sediments following a large inflow. AMBIO, 45, 130-132.

Stigebrandt, A., 2018: On the response of the Baltic proper to changes of the total phosphorus supply. Ambio, 47:31-44.

Stigebrandt, A., Rahm, L., Viktorsson, L., Ödalen, M., Hall, P.O.J., Liljebladh, B., 2014: A new phosphorus paradigm for the Baltic proper. AMBIO, 43:634-643.

---

## Author Comment (AC1) · 29 Jan 2020

The manuscript addresses the critical process of sediment P burial for the development of eutrophication. It also covers nitrogen turnover on which I am not an expert. My review will therefore focus on the P.

Reply: We thank the reviewer for taking the time to critically assess this work. We reply to all points raised below.

The burial of P in accumulation bottom areas in four sites is presented with high-quality data, both regarding the accumulation rate of matter and the P content and P-forms in

[Figure]

these layers. I find the manuscript well written in terms of language and easy accessible. I do, however get the impression that the main scope with the P investigation partly was something more than the burial as it is presented; a "sink-switch" process to e.g vivianite-formation, as evident from the many references included covering that possible process.

Reply: In our study, we assess the processes controlling the removal of phosphorus (P) and nitrogen (N) in the eutrophic Stockholm Archipelago, as indicated in both the abstract and introduction. Since P is predominantly removed by burial we have looked into its different sedimentary forms. Given that sink-switching to vivianite occurs in sediments of other areas in the Baltic Sea (e.g. Egger et al., 2015), it was indeed one of our aims to study whether this is the case in the Stockholm Archipelago. Our data suggest a vivianite-type mineral might be present at depth at Strömmen. We agree with the reviewer that vivianite formation received too much attention in the original manuscript and we will remove some of this text, including many of the references.

Main concerns

The outcome of the study regarding the P burial ends up rather basic by summing up total-P concentrations in deeper sediment layer with the sediment accumulation rate at the specific site. With all the supporting data presented, perhaps could new insights be developed? I suggest a more in-depth analysis of the P burial both in a spatial (quantitative) scale, as well as in a qualitative (formation of refractory P that forms during diagenesis and resists it) perspective. I offer my reflections on the subject as a platform to develop the discussion:

Reply: We respond point-by-point to these comments and the other suggestions of the reviewer below.

Does the lack of concentration changes in most P forms with sediment depth (Fig. 6) actually reflect mainly inert P forms settling out on the sediment surface, resistant to sediment diagenesis? Even the "authigenic P" (Ca-P) seems thus to have been formed

elsewhere than in the present sediment profile, since it already in the top sediment layers is present at a concentration it will remain throughout the sediment profile. The "sink-switch" process seems to be virtually absent (except perhaps for the Strömmen station).

Reply: We will add a few lines to the discussion (section 4.1.2) to emphasize this: "The constant concentrations of most P forms in the sediment below the clearly "enriched" surface sediments, suggest there generally is little to no sink-switching of sediment P forms in the Stockholm Archipelago. The curved shape of the porewater HPO42- profiles indicate, however, that there is still some release of P to the porewater at depth and we attribute this to slow degradation of organic matter. Both the detrital and authigenic (Ca-P) fractions are likely buried in the form in which they reached the sediment-water interface."

Indeed, some share of the organic P settling out on the sediment surface is mineralized, as evident from the decline in org-P concentration in the top ca 5 cm. This is well presented in Fig 9 where the "background" concentration is indicated. Perhaps the "top-layer" (indicated as red labelled "surface sediment sink") actually represents the P active in turn-over, as suggested in Rydin 2011. All the P indicated as background concentration would then largely be inert P forms, not relevant for the eutrophication process. A key question would then be to what extent autochthonous organic P (e.g. plankton) contributes to the supply of organic sediment P resistant enough to get permanently buried. Is the only main sink-switch of importance in this region the transformation of dissolved P in the water column to organic P (plankton), to a larger or lesser extent permanently buried in the sediment?

Reply: We agree that the red-labeled "top-layer" in Fig. 9 likely represents the P active in turn-over. We will change the term "surface sediment P sink" to "surface sediment P pool" (both in Fig. 9, which will become Fig. 10 in the revised manuscript, and throughout the text), to clarify that this part of the P is not permanently buried and is the P active in turn-over. In addition to this adaption we will add a sentence to acknowledge that this P pool represents the P active, as previously suggested by Rydin et al. (2011) – discussion, section 4.1.1: "For our study sites in the Stockholm Archipelago we calculated that this surface sediment P pool, i.e. the P active in turn-over as earlier already suggested by Rydin et al. (2011),..."

The "background-P", may indeed largely represent inert forms of P, with the exception of the Strömmen station, as already indicated by the reviewer in his previous comment. Sedimentary C/N ratios for the studied sites are, however, close to the Redfield-ratio, indicating that most of the organic matter has a "marine" i.e. planktonic origin, suggesting that most of the organic P is of autochthonous origin. We will clarify this in the revised discussion, section 4.1.1: "Besides Fe-oxides, a major part of the surface sediment P pool consists of P in organic matter (Fig. 6), which, based on the C/N values close to the Redfield-ratio (Fig. 5), is predominantly of marine origin. Part of the organic matter (and the P associated with it) is lost with depth (Fig. 6), because the most labile organic matter is degraded in the upper centimeters of the sediment, releasing the P associated with it to the pore water."

Line 391: Only "near shore construction and dredging (line 391) is presented as alternative sources for the P accumulated than land-derived. Another explanation for the high burial rate could be that the sediment to a large extent consists of old sediment (old clays) that already have undergone sediment P diagenesis processes one or even several times during the Baltic Sea life span, exposed to resuspension due to land up-lift (ca 0,5 cm/yr), and a secondary (or a third) settling out on accumulation bottom areas (Karlsson et al 2019 and references therein). But also P from the catchment (e.g. Lake Mälaren) of course contributes to this pool of P.

Reply: We agree with the reviewer that there are other alternative sources of sediment and associated P that may influence sediment and P accumulation rates. We will adapt our discussion (section 4.1.2) to clarify this: "Further research of P burial rates at additional locations in the Stockholm Archipelago, including the impact of anthropogenic activities on sedimentation rates (e.g. near-shore construction and dredging) and of

redeposition of sediments that have already undergone one or multiple diagenetic cycles (after resuspension due to, for example, land uplift; Jonsson et al., 1990; Bryhn and Håkanson, 2011) is required before these results can be extrapolated to the scale of the entire system."

A back of the envelope calculation regarding the Lake Mälaren impact on the P burial in the region might look something like this: The Stockholm county archipelago covers a water area of 3100 km2. Assuming that only 25% of this area represents accumulation bottom conditions; 780 km2, and an average burial rate of 3 g/m2 yr, it ends up to 2340 ton P/yr. This is a high flux compared to external inputs, and corresponds to as much as ca 15% of the total P input to the Baltic proper. Thus, a high share of the P found in accumulation bottom sediment seems to be recycled.

Reply: It would indeed be interesting to analyze P burial on a spatial and quantitative scale, as suggested by the reviewer. Our study, however, is specifically aiming to assess the processes controlling the removal of P and N. Our dataset is therefore not suited for (detailed) spatial and quantitative analysis of P burial. Large outstanding questions that need to be answered before reliable budget calculations can be made, are for example: What part of the Stockholm Archipelago represents accumulation areas? How much P (and in what forms) is buried in euxinic areas? We will clarify this in the revised discussion (4.1.2): "Furthermore, it remains unclear what parts of the Stockholm Archipelago represent areas of net sediment accumulation (Karlsson et al., 2019; Asmala et al., 2019) and how much P (and in what form) is buried in euxinic parts of the Stockholm Archipelago. Hence, our results cannot be directly used to resolve the apparent discrepancy between the model results of Almroth-Rosell et al. (2016) and Walve et al. (2018)."

To summarize my thoughts: the archipelago sediment seems effective in permanently trapping already inert, particulate P. But less efficient in transforming the "surface sediment sink" (Fig 9) into P forms that will be permanently buried.

Reply: The reviewer's observation that the P in the "surface sediment sink" is not efficiently permanently buried is correct. Organic P is not inert, however. With the previously suggested adaptions this should now be clarified.

The Discussion can be expanded with comparisons to other studies regarding accumulation of matter and phosphorus in the region; see Karlsson et al 2019 for suggestions on references for a more developed Discussion. Both the "surface sediment sink" (Figure 9), and the sediment accumulation rate, in Baggensfjärden and Erstaviken, are close to those found in a recent study in the adjacent Björnöfjärden (Rydin & Kumblad, 2019).

Reply: We will expand the discussion based on the reviewer's comments and suggestions, also including the studies the reviewer refers to in the discussion sections 4.1.1 and 4.1.2.

Specific comments:

Figure 10 is of limited value since it (only) shows that the P concentration at depth is rather constant, and the burial rate is largely dependent on sediment accumulation rate (at the investigated sites).

Reply: Figure 10 is not key to our study. We will therefore remove the sentence in the discussion introducing Fig. 10 (lines 389-390 of the original manuscript): "...hence our study sites plot above the linear relationship between rates of sediment accumulation rate and P burial (Fig. 10; Table 5) in the coastal zone of the Baltic Sea (Asmala et al., 2017)." To keep the figure available for interested readers we will move it to the supplementary material, where it will become Supplementary Figure 5.

The references need a check. Are all the references present needed? Line 802: Rydin et al 2011 reference is missing.

Reply: We thank the reviewer for pointing out this mistake. We will recheck our references and correct them where necessary.

Line 900: It would be useful to present sediment accumulation rates as g DW/m2 yr besides cm/yr to compensate for the compacting of the surface sediment.

Reply: We will add the sediment accumulation rates in g DW m-2 yr-1 to Table 5 as suggested by the reviewer.

References:

Karlsson, M., Bryhn, A., Håkanson, L., Hållén, J., Jonsson, P., Malmaeus M, Rydin E. 2019. On the role of sedimentological processes controlling phosphorus burial in the coastal zone of the Baltic Sea. Limnology and Oceanography 9999:1-4.

Rydin, E. Kumblad, L. 2019. Capturing past eutrophication in coastal sediments – Towards water-quality goals. Estuarine, Coastal and Shelf Science. DOI: 10.1016/j.ecss.2019.02.046.

Rydin, E., Malmaeus, M., Karlsson, M., Jonsson, P. (2011) Phosphorus Release From Coastal Baltic Sea Sediments As Estimated From Sediment Profiles. Estuarine, Coastal and Shelf Science. 92:111-117.

Reply: References:

Almroth-Rosell, E., Edman, M., Eilola, K., Meier, H. E. M., and Sahlberg, J.: Modelling nutrient retention in the coastal zone of an eutrophic sea. Biogeosciences, 13, 5753–5769, https://doi.org/10.5194/bg-13-5753-2016, 2016.

Asmala, E., Carstensen, J., Conley, D. J., Slomp, C. P., Stadmark, J., and Voss, M.: A reply to the comment by Karlsson et al., Limnol. Oceanogr. 64, 1832-1833, https://doi.org/10.1002/lno.11195, 2019.

Bryhn, A. C., and Håkanson, L.: Land uplift effects on the phosphorus cycle of the Baltic Sea, Environ. Earth Sci., 62, 1761–1770, https://doi.org/10.1007/s12665-010-0656-6, 2011.

Egger, M., Jilbert, T., Behrends, T., Rivard, C., and Slomp, C. P.: Vivianite is a major sink for phosphorus in methanogenic coastal surface sediments, Geochim. Cosmochim. Acta 169, 217–235, https://doi.org/10.1016/j.gca.2015.09.012, 2015.

Jonsson, P., Carman, R., and Wulff, F.: Laminated Sediments in the Baltic: A Tool for Evaluating Nutrient Mass Balances, Ambio, 19(3), 152–158, 1990.

Karlsson, O. M., Bryhn, A. C., Håkanson, L., Hållén, J., Jonsson, P., Malmaeus, J. M, and Rydin, E.: On the role of sedimentological processes controlling phosphorus burial in the coastal zone of the Baltic Sea, Limnol. Oceanogr., 2019.

Rydin, E., and Kumblad, L.: Capturing past eutrophication in coastal sediments– Towards water-quality goals, Estuar. Coast. Shelf Sci., 221, 184-188, https://doi.org/10.1016/j.ecss.2019.02.046, 2019.

Rydin, E., Malmaeus, M., Karlsson, M., and Jonsson, P.: Phosphorus release from coastal Baltic Sea sediments as estimated from sediment profiles, Estuar. Coast Shelf Sci., 92, 111-117, https://doi.org/10.1016/j.ecss.2010.12.020, 2011.

Walve, J., Sandberg, M., Larsson, U., and Lännergren, C.: A Baltic Sea estuary as a phosphorus source and sink after drastic load reduction: seasonal and long-term mass balances for the Stockholm inner archipelago for 1968–2015, Biogeosciences, 15(9), 3003-3025, https://doi.org/10.5194/bg-15-3003-2018, 2018.

---

## Author Comment (AC2) · 29 Jan 2020

General comments/requests (in random order):

1) The authors present bottom water nitrate as one controlling factor of benthic denitrification. What about nitrate formed during benthic nitrification as one controlling factor? Very little is mentioned about it. Please do, and discuss nitrification as a control of benthic denitrification in these sediments.

Reply: We thank the reviewer for taking the time to critically assess this work. We reply to all points raised below. In the original manuscript we presented the percent

denitrification supported by nitrification in Table 4 (see 'nitrification-denitrification %') and in the text of the results section (3.3). We then discussed the potential effects of nutrient reductions on nitrification-dentification in the discussion (final paragraph of section 4.3). Our results show that nitrification is indeed an important in controlling denitrification. We agree with the reviewer that this can be discussed in more detail. We have now completely rearranged section 4.2.2 and added additional references on the importance of nitrification-denitrification.

2) Lines 421-436 and elsewhere: Denitrification rates decreased going seaward, and the authors explained this by lower bottom water nitrate concentrations and lower organic C content of sediment along the transect going seaward. However, it is generally assumed that in coastal and shelf sediments availability of nitrate controls denitrification rates since there often is no shortage of organic C in such sediments – at least not to limit benthic dentrification. Also, in coastal and shelf sediments, the nitrate consumed in denitrification is mostly produced during nitrification in the sediments rather than being nitrate from the bottom water (cf. e.g. papers by Seitzinger (et al.)). The 18 mm oxygen penetration depth in sediment of Ingaröfjärden should allow active nitrification so that the lower bottom water nitrate concentration there should not lead to a decreased denitrification rate. Could the authors please discuss in their paper this apparent inconsistency between what has been generally found and what was found in the present study?

Reply: See our reply to the comment above. We have completely rewritten this section and provide references illustrating that our findings are supported by the published literature on the role of organic C availability.

3) The measurements were made in March, which is late winter. This period should be among the most oxygenated of the year; the vertical stratification is weak and mixing/ventilation of bottom waters should be facilitated. In addition, bottom water temperature is at its minimum, the spring bloom has not started yet, and there should be very little fresh organic matter in sediments. Can the authors please include a discussion in

their paper on this and especially on to what extent the results presented are representative on an annual scale? In my opinion, the results represent a late winter situation, and fixed N removal and retention of P in sediments most likely are very different than in e.g. summer-fall when bottom water oxygen levels and quantity of fresh organic matter in sediments can be completely different. The authors should make this very clear in their paper.

Reply: In order to highlight and more clearly discuss this aspect we now added a new sub section discussing the seasonality and the representativeness of our data on an annual basis (4.2.3 - Seasonal cycles of N processes).

4) Section 4.3 Implications: Although this section includes some interesting discussions, I cannot see that it is relevant in this paper. This section consists of discussions and speculations far beyond what can be found warranted based on the original results of this paper. This paper is not a review paper. Please focus the discussion, and the presentation of implications, on the results obtained in this study (carried out in March 2017).

Reply: In this section, we primarily wish to summarize what our findings imply for future expected developments in nutrient dynamics in the Stockholm Archipelago. We will modify this section to clarify this (including explicit references to our results).

5) Section 4.3 Implications: "...artificial reoxygenation of bottom waters (e.g. Stigebrandt and Gustafsson, 2007) will not be a long-term effective measure towards improving the water quality of the (coastal) Baltic Sea." Please explain how the results of this paper justify this statement. If the authors cannot convincingly do this, this statement should be removed.

Reply: See our reply to point 4. We will modify this sentence so that the focus lies on the Stockholm Archipelago. Our results show that better oxygenation leads to a larger surface sedimentary P pool. At the well-oxygenated site Ingaröfjärden, this pool is ∼5 times larger than at the site with the most reducing conditions (Baggensfjärden)

– see Figure 9 of our initial submission (now Fig. 10). At depth, however, sedimentary P distributions and concentrations are rather similar at all stations (with exception of the enrichments in Fe-P at Strömmen), presumably also because of the presence of relatively high concentrations of sulfide in the pore waters at all sites. This suggests that there is relatively little control of bottom water oxygen concentrations on permanent P burial and thus removal. Therefore we think this sentence is important and warranted.

6) Lines 536-538: "Further reductions in P and N inputs are necessary to ensure a reduction in the frequency of hypoxic events. Eventually this will lead to a larger surface sedimentary P sink and will be key to maintaining the efficient N filter and avoiding additional P and N recycling." I agree that this is one important measure to improve the environmental status of coastal systems and that it should be done, BUT please explain to what extent the results of this study justify this final conclusion.

Reply: Please see above our reasoning concerning P burial and the associated changes in the text (i.e. focus on the Stockholm Archipelago).

In terms of N cycling, it is possible that denitrification may increase with more oxygen in bottom waters (e.g. by increasing the oxygen penetration depth and sediment volume for nitrification) although oxygen levels at the time of sampling were most likely the highest these sites experience year-round. It is challenging to predict how N cycling processes will respond to changing oxygen conditions – particularly when sediments are exposed for longer-term (weeks-months) in nature as opposed to short term (days) in laboratory experiments. It is likely (as discussed in the manuscript – and also now amended in the abstract) that denitrification will initially increase due to fresh organic matter inputs but then decrease in favor of recycling processes (i.e. DNRA) as $NO_3^-$ is consumed and oxygen decreases as C/N ratios increase. Thus the sediments act as a source rather than a sink of N during summers (as shown in monitoring data).

We will rephrase the sentences brought forward by the reviewer so that it is (more) focused on the results of this study.

Other comments by line number:

Line 24-25: What other form(s) of P make up the remaining 50-70% (i.e. the major fractions) of P burial? Please make this clear already in Abstract.

Reply: We will modify this sentence in the abstract to: "Sedimentary P is dominated by Fe-bound P and organic P in the surface and by organic P, authigenic Ca-P and detrital P at depth."

Line 31-32: Regardless whether this statement in general is correct, what evidence does this paper provide that this statement is correct? What level of removal or retention of N and P would have occurred in the studied area if bottom waters had been better oxygenated (natural or manmade)? I do not think this statement is warranted based on the results this paper presents, so I suggest it is deleted unless the authors convincingly can argue that it is warranted. See also above.

Reply: Please see our reply to point 5 of the reviewer above. We have revised this sentence and no longer mention "artificial reoxygenation" in the abstract. The line now reads: "We emphasize the importance of nutrient load reductions as a critical management strategy for N and P removal and for the recovery of eutrophic Baltic Sea coastal zones."

Line 390: Remove "rate" in the beginning of this line.

Reply: We will remove the entire sentence in response to Reviewer 1.

Lines 412-413: "...and the increasing role of sediments as a NO3- sink along the estuarine gradient". Please clarify this text. In which direction are you meaning the estuarine gradient goes? Landward or seaward?

Reply: The trend that we were describing here is seaward. Based on comments 1, 2 and 3 this part of the discussion will be modified. We will make sure to clarify the direction of the trends in the revised manuscript.

[Figure]

Lines 414-415: "...reduction in organic matter quality as shown by a concomitant reduction in surface sediment N and organic C contents". Does organic matter quality necessarily go down when contents of N and organic C go down? Please explain.

Reply: This sentence will be removed due to rearrangements/changes to the N cycling sections.

Lines 440-451 (and later in Discussion): What did Bonaglia et al. (2017; BG) report on DNRA and its importance as a nitrate reducing process in Baltic sediments?

Reply: Text will be added describing the co-occurrence of denitrification, anammox and DNRA in sediments of the Bothnian Bay in the discussion (section 4.2.3) and the reference to Bonaglia et al. (2017) will be added to other relevant sections in the discussion.

Lines 534-535 (and elsewhere): "Combining our process measurements with available monitoring data, it is likely that N in the Stockholm Archipelago undergoes seasonal cycles of removal and retention." What do you mean with "retention" of N here? Did you intend to write recycling (e.g. through DNRA)?

Reply: This should indeed have been "recycling" and will be amended in the revised manuscript.

Table 4 head. "DN is nitrate supplied from nitrification". DN does not exist in the Table, so why is it mentioned in the Table head?

Reply: "DN" was part of an earlier version of the manuscript, which we decided to remove in the writing process. We, however, forgot to adapt the heading of Table 4. We will adapt the table heading in the revised manuscript.

Table 4: Annamox is incorrectly spelled. Should be anammox.

Reply: This will be corrected in the revised manuscript.

Reply: References:

Bonaglia, S., Hylén, A., Rattray, J.E, Kononets, M. Y., Ekeroth, N., Roos, P., Thamdrup, B., Brüchert, V, and Hall, P. O. J.: The fate of fixed nitrogen in marine sediments with low organic loading: and in situ study. Biogeosciences, 14, 285-300, https://doi.org/10.5194/bg-14-285-2017, 2017.

---

## Author Comment (AC3) · 29 Jan 2020

This manuscript presents vertically highly resolved observations in bottom sediments of the Stockholm Archipelago with the aim to quantify removal of phosphorus and nitrogen by sediment processes. The net fluxes of P and N through the sediment-water interface determines the removals of P and N (the sediment sinks). These are very important at the system level and determine together with the nutrient supplies from land-based and ocean-based sources concentrations in the water column. However, in the manuscript the estimated sediment sinks are not upscaled (i.e. horizontally integrated) to the system level which makes it impossible to verify that the removal of

P and N in the sediments is efficient as postulated in the title of the manuscript. The manuscript needs thorough revision as discussed below.

Reply: We thank the reviewer for taking the time to critically assess this work. We reply to all points raised below. Regarding the title, we note that our results show that the sediments that we have studied act as effective sinks for P and N. Data for four sites are not sufficient to upscale to the system level, as will be mentioned explicitly in the revised text. To avoid confusion, we will change our title to "Removal of phosphorus and nitrogen in sediments of the eutrophic Stockholm Archipelago".

Line 29-31. This is not shown in the manuscript, see comments on Line 514-515 below.

Reply: We agree with the reviewer that for N this statement is not entirely correct. Our study suggests that benthic N processes undergo annual cycles of removal and recycling in response to changes in bottom water redox conditions. We will therefore remove N from this sentence and add an extra sentence to more clearly describe this. For P, however, our data does support this statement. Our results show that at sites with bottom waters with year-round well-oxygenated conditions, such as Ingaröfjärden, a larger surface sedimentary P pool can develop. At depth, however, sedimentary P distributions and concentrations are rather similar for all study sites (with the exception of the enrichments in Fe-P at Strömmen). This is directly related to the high sulfide concentrations in the pore waters at depth at all sites. Hence, we find only little effect of bottom water oxygen concentrations on permanent P burial. We will include a calculation of the upward flux of sulfide in the porewater and additional context to clarify this point in the revised manuscript.

Line 31-32. This is not shown in the manuscript and moreover it is wrong as claimed in this interactive comment, see comments on Line 479-481 and Line 481-483 below.

Reply: Our results indicate that bottom water redox conditions have no long-lasting effect on P burial in the Stockholm Archipelago (please also see our reply to the previous comment). Artificial (or natural) re-oxygenation will likely influence the size of
the surface sediment P pool, but long-lasting ecosystem improvement resulting from re-oxygenation will only be reached if the nutrient input into the system is reduced. We will rephrase this sentence as follows: "We emphasize the importance of nutrient load reductions as a critical management strategy for N and P removal and for the recovery of eutrophic Baltic Sea coastal zones."

Line 85. There are also strong signs of increasing eutrophication in the Baltic Sea with large and increasing volume of anoxic water and corresponding large and increasing area of anoxic bottoms (Hansson et al. 2019). See also the comment on Line 475 below.

Reply: The areal extent and volume of hypoxic and anoxic waters in the Baltic Sea remains undoubtedly large, as is also clear from the study by Hansson et al. (2019). The report by Hansson et al. (2019), however, does not provide any new data, or analysis of data on eutrophication. The study by Andersson et al. (2017) does. Therefore we would like to keep this sentence unchanged.

Line 96. How is P removal defined? Is P removal = P burial – P reflux? P reflux is not quantified in the manuscript. Therefore, P removal is not determined. Please clarify.

Reply: Phosphorus removal is permanent burial, i.e. P stored in the sediments for time scales longer than those relevant from an anthropogenic perspective. For our study sites this permanently buried P is the sedimentary P below the active surface layer as indicated in Figure 9 (Figure 10 in the revised manuscript). We will modify the sentence the reviewer was referring to so that it is clear that with P removal we are referring to permanent burial of P.

Line 121. This is the so-called land-based supply. But there is also a sea-based supply by inflowing surface water from the open Baltic Sea. How large is the sea based supply? This is important for the calculation of the filter effect mentioned on Line 17, see also comments regarding Line 514-515 below.

Reply: We do not have a number for the sea-based supply of N. For P the sea-based supply was calculated using two different models by Walve et al. (2018), to be ∼ 100-200 t P per year. Apart from this number, our dataset is not suited, nor intended to calculate the filter effect. The aim of our study was and is to assess the processes controlling the removal of P and N.

Line 358. What is meant by hydrological restrictions? Do you mean restrictions in the water exchange due to vertical density stratification and topographical restrictions like sills? Please clarify.

Reply: With hydrological restriction we indeed mean restrictions in the water exchange due to the geographical configuration of the basins. We will clarify this in the revised manuscript.

Line 368-369. The small annual amplitude of P and O2 in the bottom water of Ingaröfjärden is said to be due to a nearly absent seasonal P recycling. However, it is more likely due to an efficient water exchange (flushing) throughout the year. Please explain why you discard the effect of efficient flushing throughout the year. By the way, it would be fine if the sill depths for the four basins considered could be mentioned in the manuscript. Knowing these helps to interpret the flushing of the deepwater of the basins.

Reply: It is the other way around, i.e. the small annual amplitude of O2 at Ingaröfjärden, with minimum O2 concentrations always well above the hypoxic threshold (Fig. 3a), leads to the near absence of seasonal P recycling as observed in seasonally hypoxic basins such as Baggensfjärden (Fig. 3a,c). The reason for the absence or presence of seasonal hypoxia in turn is indeed partly related to "flushing" or water exchange of the deep waters in the different basins. It is, however, also related to other factors, such as net primary productivity and water depth (which both influence the amount of OM reaching the bottom waters). We prefer to keep the text as it is since the link between oxygen and P recycling is our key focus here. As detailed in our reply to the previous

comment, we will clarify what we mean by hydrological restriction. We do not have access to information on the exact sill depths.

Line 390. Is P burial = P removal? If this is the case, P reflux=0. Please explain. See also comments to Line 96 above.

Reply: Permanent P burial = P removal. Please also see our reply to the comment on line 96 by Reviewer 3.

Line 432. Is the deeper O2 penetration at Ingaröfjärden due to the action of Marenzelleria?

Reply: The deeper O2 penetration at Ingaröfjärden might indeed be partly related to activity by Marenzelleria. We will indicate the potential effect of the presence of macrofauna on deeper O2 penetration at the first instance where O2 penetration is discussed (section 4.1.1)

Line 475. Continued decrease of the land-based P input to the Baltic proper has not led to reduced horizontally integrated P concentration c in the surface layer in winter. On the contrary c has increased by at least 25% since the 1980s although the landbased supply has been approximately halved (e.g. Stigebrandt, 2018). The input of organic matter into the sediments has thus rather increased. The area of anoxic bottoms increased by a factor of about 6 from the period before 1999 to the period after 1999 and attained its highest value in 2018 (Hansson et al., 2019). This should be discussed in the manuscript.

Reply: We will revise this section to clarify that our focus lies on the Stockholm Archipelago. Given that the Stockholm Archipelago is affected by nutrient cycling in the Baltic Proper (as discussed later in the section), it is important to mention the expected long term response of processes in the Baltic Sea to reduced nutrient inputs here. Given the long residence time of P in the Baltic Sea and the various feedbacks, it is not a surprise that there is not yet a decline in the anoxic area. A discussion of the

issues relevant to the Baltic Proper as detailed by Stigebrandt (2018) and Hansson et al. (2019) lies outside the scope of this paper. We note that the work by Karlsson et al. (2010), which we cite in this sentence, provides evidence for improved conditions in the Stockholm Archipelago linked to active nutrient reduction.

Line 479 – 485. The response of water column concentrations above the sediments to the sediment processes are not quantified in the present manuscript. However, there is an exception to this. This is the statement that artificial reoxygenation of bottom waters will not be a long-term effective measure towards improving the water quality of the (coastal) Baltic Sea. There is no analysis in the manuscript that supports this statement. As shown below, the statement is wrong and should be removed from the manuscript. Citation from

Reply: As detailed above, our study focuses on understanding and quantifying removal of N and P in sediments of the Stockholm Archipelago. Artificial reoxygenation aims to increase P burial. We show that, in the Stockholm Archipelago, this will not increase permanent burial of P. In the revised version of our manuscript we will quantify the upward flux of H2S that hinders formation of a larger pool of Fe bound P.

Line 479 – 481. "Increases in bottom water O2 would likely impede the observed present-day P recycling pattern in the seasonally hypoxic sites (Fig. 3c), allowing thicker Fe-oxide bearing layers and a larger Fe-bound P pool in the surface sediments (e.g. Slomp et al., 1996), hence a larger (semi-permanent) surface sedimentary P sink."

The thickness of the Fe-oxide bearing layers is determined by the oxygen penetration depth L. Cai and Sayles (1996) presented the following relationship between oxygen penetration depth L, benthic oxygen flux FO2 across the sediment-water interface and bottom water oxygen concentration [O2]bw: L=2$\theta$Ds [O2]bw)/FO2 (Equation 1)

Here $\theta$ and Ds are the porosity and diffusivity of O2 in sediment, respectively.

Equation (1) shows that the thickness L of the oxidized layer on top of the sediment varies with [O2]bw and, allowing for some inertia, the minimum thickness L=Lmin should occur approximately when [O2]bw attains its minimum. This means that Lmin can be increased by increasing the minimum bottom water oxygen concentration [O2]bw which is in accordance with the statement on Line 479-481.

However, the following statement (on Line 481-483) is presented without any proof of its validity for the Baltic Sea. Citation from Line 481-483. "This process will, however, be delayed due to the prior deposition of organic rich sediments which results in a high upward flux of H2S (i.e. legacy of hypoxia) hindering the formation of Fe-oxides."

Reply: Please see our replies to the comments on Lines 29-31 and Lines 31-32 and to the comment directly preceding this one.

This statement is maybe true for highly eutrophic lakes, but it is not true for the deep-water sediments in the much less eutrophic Baltic Sea, as discussed on p. 41 in Stige-brandt (2018). Using Sediment Profile Imagery (SPI) it was observed that the sediment surface was oxygenated within a couple of months during a natural oxygenation event due to a Major Baltic Inflow (Rosenberg et al., 2016). This means that the upward flux of H2S in the Baltic Sea deepwater sediments is not large enough to hinder the formation of an oxic layer (containing Fe-oxides) on top of the sediment when the bottom water is oxygenated. Therefore Equation (1) is applicable to the deep sediments of Baltic Sea. The oxygen penetration depth L can thus be increased by increasing [O2]bw by artificial reoxygenation of the bottom waters of the Baltic Sea.

Reply: In the revised text, we will specifically limit our discussion to artificial reoxygenation of the Stockholm Archipelago. We note, however, that a similar legacy effect due to the upward flux of hydrogen sulfide has been reported previously for the Gotland Deep following the most recent Major Baltic Inflow by Hermans et al. (2019). Instead of visual observations and conclusions on the presence or absence of Fe oxides based on sediment imagery, Hermans et al. (2019) quantified the sediment content of Fe ox-

ides and associated P in Gotland Deep sediments. The results revealed only limited Fe oxide formation and very little sequestration of P. This finding was independently corroborated by water column studies of P dynamics in the Gotland Deep showing that most P was displaced to other parts of the Baltic Sea. The lack of Fe oxide formation was attributed to the high flux of reductants, such as sulfide from the deeper sediments which allowing the presence and preservation of FeS (or FeS2) and restricted the penetration of O2 into the sediment. We show here that sediments in the Stockholm Archipelago have quite similar characteristics to those in the central Baltic Sea, i.e. high contents of organic matter, high pore water sulfide concentrations and high sedimentary concentrations of FeS and FeS2.

The major effect of oxygenation of anoxic bottom sediments is that it stops the outflow of P from the sediment. This was discussed in Stigebrandt et al. (2014), see also Almroth-Rosell et al. (2015) who show that the phosphorus release rate from the sediment drastically decreased and even became negative as a result of Major Baltic Inflows. As shown in Stigebrandt (2018), artificial reoxygenation of bottom waters should be a rapid and long-term effective measure towards reducing the eutrophication and improving the water quality of the open Baltic Sea and coastal areas with good water exchange with the open sea so that local effects of local land-based nutrient supplies are small. This disproves the following statement (on Line 483-485) in the manuscript.

Citation from Line 483 - 485 "This also explains why artificial reoxygenation of bottom waters (e.g. Stigebrandt and Gustafsson, 2007) will not be a long-term effective measure towards improving the water quality of the (coastal) Baltic Sea."

Reply: We agree that the immediate response of oxygenation (artificial or natural) of bottom waters decreases the release of P from the sediment. Our results also show that less or non-reducing bottom waters, i.e. as observed for the year-round well-oxygenated Ingaröfjärden, leads to a larger surface sedimentary P pool. At depth, however, sedimentary P distributions and concentrations are rather similar for all study sites (with the exception of the enrichments in Fe-P at Strömmen). This implies that

bottom water oxygen concentrations have little to no effect on permanent P burial, i.e. permanent long-term removal. Therefore nutrient load reductions are necessary to improve the ecological status of the Stockholm Archipelago. We will revise the text to clarify this point.

Line 516. What is meant by "control" in the sentence "continue to actively reduce and control nutrient inputs"

Reply: Here, "control" refers to "managing" the nutrients inputs, which can be done through all kinds of regulations, incl. installation of sewage treatment plants etc.

Line 514-515. In the manuscript it is postulated but not shown that the sediments are efficient filter. This would require that estimates of the N and P sinks (tonnes year-1) for the whole area were related to the total supply of nutrients (tonnes year-1), i.e. the supply from both land-based and sea-based sources.

Reply: The aim of our study is to assess the processes leading to the removal of P and N in the sediments of the Stockholm Archipelago. We never had the intention, nor claim that we would calculate the filter efficiency of the system. Please, also see our reply to the comment to Line 121 and the general comment of the reviewer.

References

Almroth-Rosell, E., Eilola, K., Kuznetzov, I., Hall, P.O.J., and Meier, H.E., 2015. A new approach to model oxygen dependent benthic phosphate fluxes in the Baltic Sea. Journal of Marine Systems, 144, 127-141.

Cai, W.J., and Sayles, F.L., 1996. Oxygen penetration depths and fluxes in marine sediments. Mar. Chem. 52, 123-131.

Hansson, M., Viktorsson, L., and Andersson, L., 2019. Oxygen survey in the Baltic Sea 2018 – Extent of anoxia and hypoxia, 1960 – 2018. SMHI, Report Oceanography No 65, 11 pp + 2 Appendices.

Rosenberg, R., Magnusson, M., Stigebrandt, A., 2016: Rapid re-oxygenation of Baltic Sea sediments following a large inflow. AMBIO, 45, 130-132.

Stigebrandt, A., 2018: On the response of the Baltic proper to changes of the total phosphorus supply. Ambio, 47:31-44.

Stigebrandt, A., Rahm, L., Viktorsson, L., Ödalen, M., Hall, P.O.J., Liljebladh, B., 2014: A new phosphorus paradigm for the Baltic proper. AMBIO, 43:634-643.

Reply: References:

Andersen, J. H., Carstensen, J., Conley, D. J., Dromph, K., Fleming‐Lehtinen, V., Gustafsson, B. G., Josefson, A. B., Norkko, A., Villnäs, A., and Murray, C.: Long‐term temporal and spatial trends in eutrophication status of the Baltic Sea, Biol. Rev., 92(1), 135-149, https://doi.org/10.1111/brv.12221, 2017.

Hansson, M., Viktorsson, L., and Andersson, L., 2019. Oxygen survey in the Baltic Sea 2018 – Extent of anoxia and hypoxia, 1960 – 2018. SMHI, Report Oceanography No 65, 11 pp + 2 Appendices.

Hermans, M., Lenstra, W. K., van Helmond, N. A. G. M., Behrends, T., Egger, M., Séguret, M. J., Gustafsson, E., Gustafsson, B. G., and Slomp, C. P.: Impact of natural re-oxygenation on the sediment dynamics of manganese, iron and phosphorus in a euxinic Baltic Sea basin, Geochim. Cosmochim. Acta, 246, 174-196, https://doi.org/10.1016/j.gca.2018.11.033, 2019.

Karlsson, O. M., Jonsson, P. O., Lindgren, D., Malmaeus, J. M., and Stehn, A.: Indications of recovery from hypoxia in the inner Stockholm archipelago, Ambio, 39(7), 486-495, https://doi.org/10.1007/s13280-010-0079-3, 2010.

Walve, J., Sandberg, M., Larsson, U., and Lännergren, C.: A Baltic Sea estuary as a phosphorus source and sink after drastic load reduction: seasonal and long-term mass balances for the Stockholm inner archipelago for 1968–2015, Biogeosciences, 15(9), 3003-3025, https://doi.org/10.5194/bg-15-3003-2018, 2018.

---

## Author Response (AR1)

**Reply to Reviewer1**

The manuscript addresses the critical process of sediment P burial for the development of eutrophication. It also covers nitrogen turnover on which I am not an expert. My review will therefore focus on the P.

Reply: We thank the reviewer for taking the time to critically assess this work. We reply to all points raised below.

The burial of P in accumulation bottom areas in four sites is presented with high-quality data, both regarding the accumulation rate of matter and the P content and P-forms in these layers. I find the manuscript well written in terms of language and easy accessible. I do, however get the impression that the main scope with the P investigation partly was something more than the burial as it is presented; a "sink-switch" process to e.g vivianite-formation, as evident from the many references included covering that possible process.

Reply: In our study, we assess the processes controlling the removal of phosphorus (P) and nitrogen (N) in the eutrophic Stockholm Archipelago, as indicated in both the abstract and introduction. Since P is predominantly removed by burial we have looked into its different sedimentary forms. Given that sink-switching to vivianite occurs in sediments of other areas in the Baltic Sea (e.g. Egger et al., 2015), it was indeed one of our aims to study whether this is the case in the Stockholm Archipelago. Our data suggest a vivianite-type mineral might be present at depth at Strommen. We agree with the reviewer that vivianite formation received too much attention in the original manuscript. We have now removed most of this text, including many of the references.

Main concerns

The outcome of the study regarding the P burial ends up rather basic by summing up total-P concentrations in deeper sediment layer with the sediment accumulation rate at the specific site. With all the supporting data presented, perhaps could new insights be developed? I suggest a more in-depth analysis of the P burial both in a spatial (quantitative) scale, as well as in a qualitative (formation of refractory P that forms during diagenesis and resists it) perspective. I offer my reflections on the subject as a platform to develop the discussion:

Reply: We respond point-by-point to these comments and the other suggestions of the reviewer below. We expect that these changes now clarify that our data provide key new insights in the controls on P burial in this coastal system.

Does the lack of concentration changes in most P forms with sediment depth (Fig. 6) actually reflect mainly inert P forms settling out on the sediment surface, resistant to sediment diagenesis? Even the "authigenic P" (Ca-P) seems thus to have been formed elsewhere than in the present sediment profile, since it already in the top sediment layers is present at a concentration it will remain throughout the sediment profile. The "sink-switch" process seems to be virtually absent (except perhaps for the Strömmen station).

Reply: We have added a few lines to the discussion to emphasize this (Lines 409-413): "The constant concentrations of most P forms in the sediment below the clearly "enriched" surface sediments, suggest there generally is little to no sink-switching of sediment P forms in the Stockholm Archipelago. The curved shape of the porewater $HPO_4^{2-}$ profiles indicate, however, that there is still some release of P to the porewater at depth and we attribute this to slow degradation of organic matter. Both the detrital and authigenic (Ca-P) fractions are likely buried in the form in which they reached the sediment-water interface."

Indeed, some share of the organic P settling out on the sediment surface is mineralized, as evident from the decline in org-P concentration in the top ca 5 cm. This is well presented in Fig 9 where the "background" concentration is indicated. Perhaps the "top-layer" (indicated as red labelled "surface sediment sink") actually represents the P active in turn-over, as suggested in Rydin 2011. All the P indicated as background concentration would then largely be inert P forms, not relevant for the eutrophication process. A key question would then be to what extent autochthonous organic P (e.g. plankton) contributes to the supply of organic sediment P resistant enough to get permanently buried. Is the only main sink-switch of importance in this region the transformation of dissolved P

in the water column to organic P (plankton), to a larger or lesser extent permanently buried in the sediment?

Reply: We agree that the red-labeled "top-layer" in Fig. 9 likely represents the P active in turn-over. We have changed the term "surface sediment P sink" to "surface sediment P pool" (both in Fig. 9, which has now become Fig. 10, and the rest of the text), to clarify that this part of the P is not permanently buried and is the P active in turn-over. In addition to this adaption we have now add a sentence to acknowledge that this P pool represents the P active, as previously suggested by Rydin et al. (2011) – discussion, (Lines 377-382): "For our study sites in the Stockholm Archipelago we calculated that this surface sediment P pool, i.e. the P active in turn-over as earlier already suggested by Rydin et al. (2011), varies between 0.036 mol P m$^{-2}$ at Baggensfjärden and 0.172 mol P m$^{-2}$ at Ingaröfjärden (between ~1 and 5 g P m$^{-2}$, respectively; Fig. 10; Table 6). This is comparable to values found for previously studied sites in the Stockholm Archipelago (1 to 7 g P m$^{-2}$; Rydin et al., 2011; Rydin and Kumblad, 2019)."

The "background-P", indeed represents P in forms that do not change much in concentration with depth, with the exception of the Strömmen station, as already indicated by the reviewer in his previous comment. Sedimentary C/N ratios for the studied sites are, however, close to the Redfield-ratio, indicating that most of the organic matter has a "marine" i.e. planktonic origin, suggesting that most of the organic P is of autochthonous origin. We have now clarified this (Lines 373-377): "Besides Fe-oxides, a major part of the surface sediment P pool consists of P in organic matter (Fig. 6), which, based on the C/N values close to the Redfield-ratio (Fig. 5), is predominantly of marine origin. Part of the organic matter (and the P associated with it) is lost with depth (Fig. 6), because the most labile organic matter is degraded in the upper centimeters of the sediment, releasing the P associated with it to the pore water."

Line 391: Only "near shore construction and dredging (line 391) is presented as alternative sources for the P accumulated than land-derived. Another explanation for the high burial rate could be that the sediment to a large extent consists of old sediment (old clays) that already have undergone sediment P diagenesis processes one or even several times during the Baltic Sea life span, exposed to resuspension due to land up-lift (ca 0,5 cm/yr), and a secondary (or a third) settling out on accumulation bottom areas (Karlsson et al 2019 and references therein). But also P from the catchment (e.g. Lake Mälaren) of course contributes to this pool of P.

Reply: We agree with the reviewer that there are other alternative sources of sediment and associated P that may influence sediment and P accumulation rates. We have adapted our discussion to clarify this (Lines 400-404): "Further research of P burial rates at additional locations in the Stockholm Archipelago, including the impact of anthropogenic activities on sedimentation rates (e.g. near-shore construction and dredging) and of redeposition of sediments that have already undergone one or multiple diagenetic cycles (after resuspension due to, for example, land uplift; Jonsson et al., 1990; Bryhn and Håkanson, 2011) is required before these results can be extrapolated to the scale of the entire system."

A back of the envelope calculation regarding the Lake Mälaren impact on the P burial in the region might look something like this: The Stockholm county archipelago covers a water area of 3100 km2. Assuming that only 25% of this area represents accumulation bottom conditions; 780 km2, and an average burial rate of 3 g/m2 yr, it ends up to 2340 ton P/yr. This is a high flux compared to external inputs, and corresponds to as much as ca 15% of the total P input to the Baltic proper. Thus, a high share of the P found in accumulation bottom sediment seems to be recycled.

Reply: It would indeed be interesting to analyze P burial on a spatial and quantitative scale, as suggested by the reviewer. Our study, however, is specifically aiming to assess the processes controlling the removal of P and N. Our dataset is therefore not suited for (detailed) spatial and quantitative analysis of P burial. Large outstanding questions that need to be answered before reliable budget calculations can be made, are for example: What part of the Stockholm Archipelago represents accumulation areas? How much P (and in what forms) is buried in euxinic areas? We have now indicated this (Lines 404-408): "Furthermore, it remains unclear what parts of the Stockholm Archipelago represent areas of net sediment accumulation (Karlsson et al., 2019; Asmala et al., 2019) and how much P (and in what form) is buried in euxinic parts of the Stockholm Archipelago. Hence, our results cannot be directly used to resolve the apparent discrepancy between the model results of Almroth-Rosell et al. (2016) and Walve et al. (2018)."

To summarize my thoughts: the archipelago sediment seems effective in permanently trapping already inert, particulate P. But less efficient in transforming the "surface sediment sink" (Fig 9) into P forms that will be permanently buried.

Reply: The reviewer's observation that the P in the "surface sediment sink" is not efficiently permanently buried is correct. Organic P is not inert, however. With the previously indicated adaptions this should now be clarified.

The Discussion can be expanded with comparisons to other studies regarding accumulation of matter and phosphorus in the region; see Karlsson et al 2019 for suggestions on references for a more developed Discussion. Both the "surface sediment sink" (Figure 9), and the sediment accumulation rate, in Baggensfjärden and Erstaviken, are close to those found in a recent study in the adjacent Björnöfjärden (Rydin & Kumblad, 2019).

Reply: We have expanded the discussion based on the reviewer's comments and suggestions, also including the studies the reviewer refers to in the discussion sections 4.1.1 and 4.1.2.

Specific comments:

Figure 10 is of limited value since it (only) shows that the P concentration at depth is rather constant, and the burial rate is largely dependent on sediment accumulation rate (at the investigated sites).

Reply: Figure 10 is not key to our study. We have therefore removed the sentence in the discussion introducing Fig. 10 (lines 389-390 of the original manuscript): "...hence our study sites plot above the linear relationship between rates of sediment accumulation rate and P burial (Fig. 10; Table 5) in the coastal zone of the Baltic Sea (Asmala et al., 2017)." To keep the figure available for interested readers we have moved it to the supplementary material, where it has become Supplementary Figure 5.

The references need a check. Are all the references present needed? Line 802: Rydin et al 2011 reference is missing.

Reply: We thank the reviewer for pointing out this mistake. We have rechecked our references and corrected them where necessary.

Line 900: It would be useful to present sediment accumulation rates as g DW/m2 yr besides cm/yr to compensate for the compacting of the surface sediment.

Reply: We have added the sediment accumulation rates in g DW m$^{-2}$ yr$^{-1}$ to Table 5 (now Table 6) as suggested by the reviewer.

Inputs of $C_{org}$ provide both a C-source for heterotrophic processes (e.g. denitrification) as well as a source of $NH_4^+$ (from remineralisation processes) for nitrification and subsequent $NO_3^-$ production. In coastal sediments $C_{org}$ is not thought to limit denitrification. However, in complex basin systems such as the Stockholm Archipelago, and the Baltic Sea coastal zone in general, differences in ventilation and retention times between basins may mean that $C_{org}$ inputs are more variable than assumed (see section 4.2.1). Available $C_{org}$ in Ingaröfjärden (Table 2) may be less labile than at other sites due to such hydrological variations, with the deep (18 mm) oxygen penetration indicating a lower organic matter reactivity and sediment respiration compared to the other sites. Lower labile $C_{org}$ availability will limit heterotrophic denitrification and may explain why anammox, an autotrophic process, is more dominant at this site (Table 5; Fig. 7). The presence of the invasive polychaete Marenzelleria (Table 2) may also reduce N removal at Ingaröfjärden and enhance the efflux and transport of $NH_4^+$ from sediments (e.g. Hietanen et al., 2007; Bonaglia et al., 2013), although it should be noted that the impacts of in fauna on N cycling are notoriously complex (Robertson et al., 2019)."

New figure (Figure 8):

[Figure]

**Figure 8.** Relationship between total denitrification rates and denitrification driven by $NO_3^-$ from nitrification (nitrification-denitrification) as process rates (left) and as a percentage of total denitrification (right).

2) Lines 421-436 and elsewhere: Denitrification rates decreased going seaward, and the authors explained this by lower bottom water nitrate concentrations and lower organic C content of sediment along the transect going seaward. However, it is generally assumed that in coastal and shelf sediments availability of nitrate controls denitrification rates since there often is no shortage of organic C in such sediments – at least not to limit benthic dentrification. Also, in coastal and shelf sediments, the nitrate consumed in denitrification is mostly produced during nitrification in the sediments rather than being nitrate from the bottom water (cf. e.g. papers by Seitzinger (et al.)). The 18 mm oxygen penetration depth in sediment of Ingaröfjärden should allow active nitrification so that the lower bottom water nitrate concentration there should not lead to a decreased denitrification rate. Could the authors please discuss in their paper this apparent inconsistency between what has
been generally found and what was found in the present study?

Reply: See our reply to the comment above. We have completely rewritten this section and provide references illustrating that our findings are supported by the published literature on the role of organic C availability.

3) The measurements were made in March, which is late winter. This period should be among the most oxygenated of the year; the vertical stratification is weak and mixing/ventilation of bottom waters should be facilitated. In addition, bottom water temperature is at its minimum, the spring bloom has not started yet, and there should be very little fresh organic matter in sediments. Can the authors please include a discussion in their paper on this and especially on to what extent the results presented are representative on an annual scale? In my opinion, the results represent a late winter situation, and fixed N removal and retention of P in sediments most likely are very different than in e.g. summer-fall when bottom water oxygen levels and quantity of fresh organic matter in sediments can be completely different. The authors should make this very clear in their paper.

Reply: In order to highlight and more clearly discuss this aspect we now added a new sub section discussing the seasonality and the representativeness of our data on an annual basis" "4.2.3 - Seasonal cycles of N processes" (Lines 476-509):

"**4.2.3 Seasonal cycles of N processes**
Sampling and experiments in this study were carried out in late winter (March), a period in the Baltic Sea when the water column is well mixed, with cold and well oxygenated bottom waters and with persistently low organic inputs to sediments. However, conditions are of course not static throughout the annual cycle. Seasonal warming, stratification, phytoplankton blooms and consumption and release of nutrients as seen in year-round monitoring data (Fig. 3d; Sup. Fig. 1) will have marked effects on sediment nutrient cycling. Year-round bottom water monitoring data collected at Bäggensfjärden show that $NO_3^-$ accumulates annually in bottom waters during the autumn and winter months before being consumed during spring and summer by phytoplankton blooms (Fig. 3d). Hypoxic bottom waters develop over summer following bloom collapse and subsequent enhanced deposition of fresh organic matter and enhanced benthic respiration during summer and early

autumn. Bottom water total N concentrations increase during summer in connection with the hypoxic events (Fig. 3d) due to enhanced benthic remineralization and subsequent $NH_4^+$ efflux from sediments.

Increased organic inputs following the spring bloom are likely to lead to increases in denitrification as the season progresses, as is commonly observed in coastal sediments (e.g. Piña-Ochoa and Álvarez-Cobelas, 2006; Jäntti et al., 2011; Bonaglia et al., 2014). Thus, a similar scenario would be assumed for the Stockholm Archipelago as for other estuaries, leading to higher rates of denitrification during spring and early summer and a reduction in autumn and winter as organic inputs subside (e.g. Bonaglia et al., 2014). Depending on the bloom intensity and organic matter inputs during spring, increased benthic respiration may lead to more reduced conditions in surface sediments as bottom water $O_2$ is depleted. The availability of $NO_3^-$ also declines under hypoxic/anoxic conditions due to $NO_3^-$ consumption in the water column, lower oxygen penetration and thus a reduced volume of surface sediment where nitrification can occur and from the reduced efficiency of nitrification under low oxygen conditions. The resulting high C/N conditions may cause process dominance to shift from N removal by denitrification (or anammox) to retention by DNRA (e.g. An and Gardner, 2002; Burgin and Hamilton, 2007; Giblin et al., 2013; Algar and Vallino, 2014; Kraft et al., 2014), as has been repeatedly demonstrated in field, laboratory and model studies (An and Gardner, 2002; Algar and Vallino, 2014; Kraft et al., 2014; van den Berg et al., 2016; Kessler et al., 2018). Thus, under hypoxic conditions in summer/autumn, DNRA may become the dominant $NO_3^-$-reducing process, altering the role of sediments from a $NO_3^-$ sink through $N_2$ production, to a source via increased $NH_4^+$ release by DNRA.

While we have not assessed $NO_3^-$-reducing process over different seasons at these four sites, we have demonstrated the microbial metabolic potential for DNRA is present through the detection of DNRA activity in incubations at all four sites (Table 5). We suggest that it is highly likely that DNRA contributes to $NH_4^+$ efflux at sites during sporadic bottom water hypoxia. Thus, the capacity for N removal by denitrification may be reduced during bottom water hypoxia while the likelihood of N recycling by DNRA increases as shown in previous Baltic Sea studies (e.g. Jäntti et al., 2011; Jäntti and Hietanen, 2012; Bonaglia et al., 2014)."

4) Section 4.3 Implications: Although this section includes some interesting discussions, I cannot see that it is relevant in this paper. This section consists of discussions and speculations far beyond what can be found warranted based on the original results of this paper. This paper is not a review paper. Please focus the discussion, and the presentation of implications, on the results obtained in this study (carried out in March 2017).

Reply: In this section, we primarily wish to summarize what our findings imply for future expected developments in nutrient dynamics in the Stockholm Archipelago. We have now modified this section (including its heading) to clarify this and included explicit references to our results (Lines 510-541):

**"4.3 Implications for future water quality in the Stockholm Archipelago**
Continued decreases in nutrient inputs to the Baltic Sea (Gustafsson et al., 2012; Andersen et al., 2017) and the Stockholm Archipelago (Karlsson et al., 2010) are likely to reduce phytoplankton growth, lead to reduced organic matter input into the sediments and, eventually, to higher $O_2$ concentrations in bottom waters.

**Our results** indicate that increases in bottom water $O_2$ would likely impede the observed present-day P recycling pattern at the seasonally hypoxic sites (Fig. 3c), allowing thicker Fe-oxide bearing layers and a larger Fe-bound P pool in the surface sediments (e.g. Slomp et al., 1996), hence a larger (semi-permanent) surface sedimentary P pool. This process will, however, be delayed due to the prior deposition of organic rich sediments which results in a high upward flux of $H_2S$ (Table 3), i.e. legacy of hypoxia hindering the formation of Fe-oxides that can bind P. Because of this legacy effect, we expect that artificial reoxygenation of bottom waters (e.g. Stigebrandt and Gustafsson, 2007), if applied in the Stockholm Archipelago, is unlikely to be a long-term effective measure towards improving the water quality since it does not stimulate permanent P burial in these sediments and a large impact on the Fe-P pool is hindered by the high upward $H_2S$ flux. Further nutrient reduction for the Stockholm Archipelago is expected to eventually lead to a reversal from export of P to the open Baltic Sea to import of P from the open Baltic Sea (Savchuk, 2005; Almroth-Rosell et al., 2016). This shows that improvement of the water quality in the Stockholm Archipelago is to a great extent coupled to nutrient management strategies for the entire Baltic Sea.

**Our results** indicate that in the Stockholm Archipelago, N likely goes through cycles of retention and removal throughout the year in relation to bottom water hypoxia. N is removed by denitrification during colder months when $NO_3^-$ availability is high, while DNRA is likely to increase during hypoxic, $NO_3^-$-depleted months. Reductions in the frequency of hypoxic bottom waters will thus reduce the

amount of time that sediments potentially recycle bioavailable N via DNRA and sediments may be more likely to act as a net sink for N through denitrification on an annual basis.
Continued recovery of the Stockholm Archipelago is also likely to lead to (re-)colonisation by bioturbating macrofaunal populations that have been driven out by hypoxic bottom waters (Diaz and Rosenberg, 2008; Voss et al., 2011). This may enhance P burial and denitrification by sediment reworking and oxygenation (e.g. Pelegri and Blackburn, 1995; Laverock et al., 2011; Norkko et al., 2012). While we still lack the predictive capabilities required to allow us to assess how fauna may influence sediment biogeochemistry (Griffiths et al., 2017; Robertson et al., 2019), reductions in nutrient inputs and phytoplankton bloom intensities, and eventual recolonization by fauna at inner archipelago sites is likely to maintain and reinforce active P and N removal processes. Thus, these coastal sediments are likely to continue to contribute to removal of P and N as long as we continue to actively reduce and control nutrient inputs."

5) Section 4.3 Implications: "…artificial reoxygenation of bottom waters (e.g. Stigebrandt and Gustafsson, 2007) will not be a long-term effective measure towards improving the water quality of the (coastal) Baltic Sea." Please explain how the results of this paper justify this statement. If the authors cannot convincingly do this, this statement should be removed.

Reply: See our reply to point 4. We have now modified this sentence so that the focus lies on the Stockholm Archipelago. Our results show that better oxygenation leads to a larger surface sedimentary P pool. At the well-oxygenated site Ingaröfjärden, this pool is ~5 times larger than at the site with the most reducing conditions (Baggensfjärden), see Figure 10. At depth, however, sedimentary P distributions and concentrations are rather similar at all stations (with exception of the enrichments in Fe-P at Strömmen), presumably also because of the presence of relatively high concentrations of sulfide in the pore waters at all sites. This suggests that there is relatively little control of bottom water oxygen concentrations on permanent P burial and thus removal. Therefore we think this sentence is important and warranted.

6) Lines 536-538: "Further reductions in P and N inputs are necessary to ensure a reduction in the frequency of hypoxic events. Eventually this will lead to a larger surface sedimentary P sink and will be key to maintaining the efficient N filter and avoiding additional P and N recycling." I agree that this is one important measure to improve the environmental status of coastal systems and that it should be done, BUT please explain to what extent the results of this study justify this final conclusion.

Reply: Please see above our reasoning concerning P burial and the associated changes in the text (i.e. focus on the Stockholm Archipelago).

In terms of N cycling, it is possible that denitrification may increase with more oxygen in bottom waters (e.g. by increasing the oxygen penetration depth and sediment volume for nitrification) although oxygen levels at the time of sampling were most likely the highest these sites experience year-round. It is challenging to predict how N cycling processes will respond to changing oxygen conditions – particularly when sediments are exposed for longer-term (weeks-months) in nature as opposed to short term (days) in laboratory experiments. It is likely (as discussed in the manuscript – and also now amended in the abstract) that denitrification will initially increase due to fresh organic matter inputs but then decrease in favor of recycling processes (i.e. DNRA) as $NO_3^-$ is consumed and oxygen decreases as C/N ratios increase. Thus the sediments act as a source rather than a sink of N during summers (as shown in monitoring data).

We have now rephrased the sentences brought forward by the reviewer so that it is (more) focused on the results of this study (Lines 557-559): "Further reductions in P and N inputs are expected to reduce the frequency of hypoxic events and to continue to support the Stockholm Archipelago's capacity to remove P and N loads."

Other comments by line number:
Line 24-25: What other form(s) of P make up the remaining 50-70% (i.e. the major fractions) of P burial? Please make this clear already in Abstract.

Reply: We have now modified this sentence (Lines 25-26): "Sedimentary P is dominated by Fe-bound P and organic P at the sediment surface and by organic P, authigenic Ca-P and detrital P at depth."

Line 31-32: Regardless whether this statement in general is correct, what evidence does this paper provide that this statement is correct? What level of removal or retention of N and P would have occurred in the studied area if bottom waters had been better oxygenated (natural or manmade)? I do not think this statement is warranted based on the results this paper presents, so I suggest it is deleted unless the authors convincingly can argue that it is warranted. See also above.

Reply: Please see our reply to point 5 of the reviewer above. We have revised this sentence and no longer mention "artificial reoxygenation" in the abstract. The line now reads (Lines 32-34): "We emphasize the importance of nutrient load reductions as a critical management strategy for P and N removal and the recovery of eutrophic Baltic Sea coastal zones."

Line 390: Remove "rate" in the beginning of this line.

Reply: We have removed the entire sentence in response to Reviewer 1.

Lines 412-413: "…and the increasing role of sediments as a $NO_3^-$ sink along the estuarine gradient". Please clarify this text. In which direction are you meaning the estuarine gradient goes? Landward or seaward?

Reply: The trend that we were describing here is seaward. Based on comments 1, 2 and 3 this part of the discussion has been modified. We have clarified the direction of the trends throughout the manuscript.

Lines 414-415: "…reduction in organic matter quality as shown by a concomitant reduction in surface sediment N and organic C contents". Does organic matter quality necessarily go down when contents of N and organic C go down? Please explain.

Reply: This sentence has been removed due to rearrangements/changes to the N cycling sections.

Lines 440-451 (and later in Discussion): What did Bonaglia et al. (2017; BG) report on DNRA and its importance as a nitrate reducing process in Baltic sediments?

Reply: Text has been added describing the co-occurrence of denitrification, anammox and DNRA in sediments of the Bothnian Bay in the discussion (section 4.2.3) and the reference to Bonaglia et al. (2017) has been added to other relevant sections in the discussion.

Lines 534-535 (and elsewhere): "Combining our process measurements with available monitoring data, it is likely that N in the Stockholm Archipelago undergoes seasonal cycles of removal and retention." What do you mean with "retention" of N here? Did you intend to write recycling (e.g. through DNRA)?

Reply: This should indeed have been "recycling" and has been amended.

Table 4 head. "DN is nitrate supplied from nitrification". DN does not exist in the Table, so why is it mentioned in the Table head?

Reply: "DN" was part of an earlier version of the manuscript, which we decided to remove in the writing process. We, however, forgot to adapt the heading of Table 4 (now 5). We have adapted the table heading now.

Table 4: Annamox is incorrectly spelled. Should be anammox.

Reply: This has now been corrected.

[revised manuscript text omitted]

**Reply to reviewer 3**

This manuscript presents vertically highly resolved observations in bottom sediments of the Stockholm Archipelago with the aim to quantify removal of phosphorus and nitrogen by sediment processes. The net fluxes of P and N through the sediment-water interface determines the removals of P and N (the sediment sinks). These are very important at the system level and determine together with the nutrient supplies from land-based and ocean-based sources concentrations in the water column. However, in the manuscript the estimated sediment sinks are not upscaled (i.e. horizontally integrated) to the system level which makes it impossible to verify that the removal of P and N in the sediments is efficient as postulated in the title of the manuscript. The manuscript needs thorough revision as discussed below.

Reply: We thank the reviewer for taking the time to critically assess this work. We reply to all points raised below. Regarding the title, we note that our results show that the sediments that we have studied act as effective sinks for P and N. Data for four sites are not sufficient to upscale to the system level, as we mention now in the revised text (Lines 399-403): "Further research of P burial rates at additional locations in the Stockholm Archipelago, including the impact of anthropogenic activities on sedimentation rates (e.g. near-shore construction and dredging) and of redeposition of sediments that have already undergone one or multiple diagenetic cycles (after resuspension due to, for example, land uplift; Jonsson et al., 1990; Bryhn and Håkanson, 2011) is required before these results can be extrapolated to the scale of the entire system."

We have also changed our title to "Removal of phosphorus and nitrogen in sediments of the eutrophic Stockholm Archipelago".

Line 29-31. This is not shown in the manuscript, see comments on Line 514-515 below.

Reply: We agree with the reviewer and we have revised the text accordingly. Our study suggests that benthic N processes undergo annual cycles of removal and recycling in response to changes in bottom water redox conditions. We have now removed N from this sentence and added an extra sentence to more clearly describe this (Lines 31-32): "Our results suggest that benthic N processes undergo annual cycles of removal and recycling in response to hypoxic conditions."

Our results show that at sites with bottom waters with year-round well-oxygenated conditions, such as Ingaröfjärden, a larger surface sedimentary P pool can develop. At depth, however, sedimentary P distributions and concentrations are similar for all study sites (with the exception of the enrichments in Fe-P at Strömmen). Revised text (lines 29-31):" Our results explain how sediments in this eutrophic coastal system can remove P through burial at a relatively high rate, regardless of whether the bottom waters are oxic or (frequently) hypoxic."

In the manuscript we now better explain why we do not see a strong effect of bottom water oxygen on permanent P burial. We show that this is directly related to the high sulfide concentrations in the pore waters at depth at all sites. We have included a calculation of the upward flux of sulfide in the porewater and the diffusive uptake of $O_2$ by the sediment, including additional context to clarify that deeper oxygen penetration in the sediment (and hence, a greater Fe-bound P pool) is hindered by the high oxygen demand of the sulfide (and ammonium) diffusing up from deeper layers. Changes made in the text:

Abstract:

Lines 21-22: "The abundant presence of sulfide in the porewater and its high upward flux towards the sediment surface (~4 to 8 mmol $m^{-2}$ $d^{-1}$), linked to prior deposition of organic-rich sediments in a low oxygen setting ("legacy of hypoxia"), hinders the formation of a larger Fe-oxide-bound P pool in winter."

Materials and Methods:

Lines 154-156: "The diffusive uptake of $O_2$ was determined by numerical modelling with PROFILE (Berg et al., 1998) using the high-resolution $O_2$ measurements."

Lines 172-173: "Upward fluxes of $H_2S$ in the porewater towards the sediment surface were calculated as detailed in Hermans et al. (2019a)."

Results:

Lines 288-290: "The $O_2$ penetration depth is deepest (18 mm) at Ingaröfjärden, while at the other three sites the $O_2$ penetration depth is relatively shallow (<4 mm; Table 2; Sup. Fig. 3). The diffusive uptake of $O_2$ is high at Strömmen and Baggensfjärden (~14 mmol m$^{-2}$ d$^{-1}$) and low at Ingaröfjärden (3 mmol m$^{-2}$ d$^{-1}$; Table 2)."

Lines 296-297: "The flux of $H_2S$ towards the sediment surface is high at all sites (~4 to 8 mmol m$^{-2}$ d$^{-1}$)."

Discussion:

Lines 382-391: "The surface sediment P pool, could, however, have been much larger for Strömmen, Baggensfjärden and Erstaviken if all of the FeS in the surface sediments would seasonally transform to Fe-oxides. The lack of such a transformation is likely linked to the high upward flux of $H_2S$ to the surface sediment (4.2 to 7.6 mmol m$^{-2}$ d$^{-1}$; Table 3). Besides the $H_2S$ flux, there is a relatively large efflux of $NH_4^+$ from the sediments into the bottom water (up to 1.4 mmol m$^{-2}$ d$^{-1}$; Table 5). Both the $H_2S$ and the $NH_4^+$ flux originate from decomposing organic rich sediments at depth (Fig. 4). Upon aerobic oxidation, two moles of $O_2$ are consumed per mole of $H_2S$ or $NH_4^+$ (e.g. Reed et al., 2011). Thus, the oxygen demand resulting from these $H_2S$ and $NH_4^+$ fluxes is very high when compared to the diffusive flux of $O_2$ into the sediment (3.1 – 13.8 mmol m$^{-2}$ d$^{-1}$; Table 2). As a consequence of the high $H_2S$ flux, FeS is formed and/or preserved (Fig. 5), and formation of a large(r) pool of Fe-oxides and Fe-bound P pool is hindered."

Tables:

Table 2:

|  | Strömmen | Baggensfjärden | Erstaviken | Ingaröfjärden |
| --- | --- | --- | --- | --- |
| Bottom water $O_2$ (mL L$^{-1}$) | 7.6 | 7 | 6.7 | 8.5 |
| $O_2$ penetration depth* (mm) | 2.1 | 1.9 | 3.6 | 18 |
| **Diffusive uptake of $O_2$ (mmol m$^{-2}$ d$^{-1}$)** | **13.4** | **13.8** | **7.3** | **3.1** |
| Bottom water salinity | 5.2 | 6.2 | 6.4 | 6.2 |
| Bottom water temperature (°C) | 1.5 | 2.4 | 2.2 | 1.3 |
| Sediment type | Mud | Mud | Mud | Bioturbated mud |
| Suboxic zone* (mm) | 4 | - | 15 | 25 |
| Macrofauna | None | None | None | *Marenzelleria* |

Table 3:

|  | Strömmen | Baggensfjärden | Erstaviken | Ingaröfjarden |
| --- | --- | --- | --- | --- |
| Sediment top (cm) | 0.75 | 0.75 | 1.75 | 8.25 |
| Sediment bottom (cm) | 2.25 | 7.25 | 8.25 | 15 |
| $H_2S$ top (µmol L$^{-1}$) | 2 | 3 | 7 | 36 |
| $H_2S$ bottom (µmol L$^{-1}$) | 385 | 899 | 1111 | 1340 |
| Diffusive flux (mmol m$^{-2}$ d$^{-1}$) | 7.6 | 4.2 | 5.2 | 6.5 |

Line 31-32. This is not shown in the manuscript and moreover it is wrong as claimed in this interactive comment, see comments on Line 479-481 and Line 481-483 below.

Reply: We have modified this sentence and no longer mention artificial reoxygenation. Our results indicate that bottom water redox conditions have no long-lasting effect on P burial in the Stockholm Archipelago (please also see our reply to the previous comment). The line now reads (Lines 32-34): "We emphasize the importance of nutrient load reductions as a critical management strategy for P and N removal and the recovery of eutrophic Baltic Sea coastal zones."

Line 85. There are also strong signs of increasing eutrophication in the Baltic Sea with large and increasing volume of anoxic water and corresponding large and increasing area of anoxic bottoms (Hansson et al. 2019). See also the comment on Line 475 below.

Reply: The areal extent and volume of hypoxic and anoxic waters in the Baltic Sea remains undoubtedly large, as is also clear from the study by Hansson et al. (2019). The report by Hansson et al. (2019), however, does not provide any new data, or analysis of data on eutrophication. The study by Andersen et al. (2017) does. Therefore we prefer to keep this sentence as it is.

Line 96. How is P removal defined? Is P removal = P burial – P reflux? P reflux is not quantified in the manuscript. Therefore, P removal is not determined. Please clarify.

Reply: Phosphorus removal is permanent burial, i.e. P stored in the sediments for time scales longer than those relevant from an anthropogenic perspective. For our study sites this permanently buried P is the sedimentary P below the active surface layer as indicated in Figure 9 (now Figure 10). We have modified this sentence (Lines 100-102): "These apparently conflicting results between different modelling approaches emphasizes the need to better understand and quantify P removal, i.e. permanent burial of P in the sediment."

Line 121. This is the so-called land-based supply. But there is also a sea-based supply by inflowing surface water from the open Baltic Sea. How large is the sea based supply? This is important for the calculation of the filter effect mentioned on Line 17, see also comments regarding Line 514-515 below.

Reply: We do not have a number for the sea-based supply of N. For P the sea-based supply was calculated using two different models by Walve et al. (2018), to be ~ 100-200 t P per year. Apart from this number, our dataset is not suited, nor intended to calculate the filter effect. The aim of our study was and is to assess the processes controlling the removal of P and N.

Line 358. What is meant by hydrological restrictions? Do you mean restrictions in the water exchange due to vertical density stratification and topographical restrictions like sills? Please clarify.

Reply: With hydrological restriction we indeed mean restrictions in the water exchange due to the geographical configuration of the basins. We have now clarified this (Lines 360-363): "This is also reflected at our study sites, with Baggensfjärden being the most $O_2$ depleted and restricted basin (i.e. land-locked with narrow and relatively shallow connections to adjacent basins) and Ingaröfjärden being the least restricted and subsequently, the most consistently well-oxygenated basin throughout the year (Table 1; Figs. 1, 2 and Sup. Fig. 1)."

Line 368-369. The small annual amplitude of P and O2 in the bottom water of Ingaröfjärden is said to be due to a nearly absent seasonal P recycling. However, it is more likely due to an efficient water exchange (flushing) throughout the year. Please explain why you discard the effect of efficient flushing throughout the year. By the way, it would be fine if the sill depths for the four basins considered could be mentioned in the manuscript. Knowing these helps to interpret the flushing of the deepwater of the basins.

Reply: It is the other way around, i.e. the small annual amplitude of $O_2$ at Ingaröfjärden, with minimum $O_2$ concentrations always well above the hypoxic threshold (Fig. 3a), leads to the near absence of seasonal P recycling as observed in seasonally hypoxic basins such as Baggensfjärden (Fig. 3a,c). The reason for the absence or presence of seasonal hypoxia in turn is indeed partly related to "flushing" or water exchange of the deep waters in the different basins. It is, however, also related to other factors, such as net primary productivity and water depth (which both influence the amount of OM reaching the bottom waters). We have kept the text as it is since the link between oxygen and P recycling is our key focus here. As detailed in our reply to the previous comment, we

have now clarified what we mean by hydrological restriction. We do not have access to information on the exact sill depths.

Line 390. Is P burial = P removal? If this is the case, P reflux=0. Please explain. See also comments to Line 96 above.

Reply: Permanent P burial = P removal. Please also see our reply to the comment on line 96 by Reviewer 3.

Line 432. Is the deeper O2 penetration at Ingaröfjärden due to the action of Marenzelleria?

Reply: The deeper $O_2$ penetration at Ingaröfjärden might indeed be partly related to activity by *Marenzelleria*. We have indicated the potential effect of the presence of macrofauna on deeper $O_2$ penetration at the first instance where $O_2$ penetration is discussed (Lines 371-372): "In such basins, deeper O2 penetration, which might partly be related to the presence of macrofauna (Sup. Fig. 3),"

Line 475. Continued decrease of the land-based P input to the Baltic proper has not led to reduced horizontally integrated P concentration c in the surface layer in winter. On the contrary c has increased by at least 25% since the 1980s although the landbased supply has been approximately halved (e.g. Stigebrandt, 2018). The input of organic matter into the sediments has thus rather increased. The area of anoxic bottoms increased by a factor of about 6 from the period before 1999 to the period after 1999 and attained its highest value in 2018 (Hansson et al., 2019). This should be discussed in the manuscript.

Reply: We have revised this section to clarify that our focus lies on the Stockholm Archipelago (please see our reply and adaptions in response to point 4 by Reviewer 2). Given that the Stockholm Archipelago is affected by nutrient cycling in the Baltic Proper (as discussed later in the section), it is important to mention the expected long term response of processes in the Baltic Sea to reduced nutrient inputs here. Given the long residence time of P in the Baltic Sea and the various feedbacks, it is not a surprise that there is not yet a decline in the anoxic area. A discussion of the issues relevant to the Baltic Proper as detailed by Stigebrandt (2018) and Hansson et al. (2019) lies outside the scope of this paper. We note that the work by Karlsson et al. (2010), which we cite in this sentence, provides evidence for improved conditions in the Stockholm Archipelago linked to active nutrient reduction.

Line 479 – 485. The response of water column concentrations above the sediments to the sediment processes are not quantified in the present manuscript. However, there is an exception to this. This is the statement that artificial reoxygenation of bottom waters will not be a long-term effective measure towards improving the water quality of the (coastal) Baltic Sea. There is no analysis in the manuscript that supports this statement. As shown below, the statement is wrong and should be removed from the manuscript. Citation from

Reply: As detailed above, our study focuses on understanding and quantifying removal of N and P in sediments of the Stockholm Archipelago. Artificial reoxygenation aims to increase P burial. We show that, in the Stockholm Archipelago, this is unlikely to increase permanent burial of P. We have now calculated the upward flux of $H_2S$ that hinders formation of a larger pool of Fe bound P. Please see our elaborate response to the comment on Line 29-31 by the reviewer. The revised text now reads as follows: "Because of this legacy effect, we expect that artificial reoxygenation of bottom waters (e.g. Stigebrandt and Gustafsson, 2007), if applied in the Stockholm Archipelago, is unlikely to be a long-term effective measure towards improving the water quality since it does not stimulate permanent P burial in these sediments and a large impact on the Fe-P pool is hindered by the high upward H2S flux."

Line 479 – 481. "Increases in bottom water O2 would likely impede the observed present-day P recycling pattern in the seasonally hypoxic sites (Fig. 3c), allowing thicker Fe-oxide bearing layers and a larger Fe-bound P pool in the surface sediments (e.g. Slomp et al., 1996), hence a larger (semi-permanent) surface sedimentary P sink."

The thickness of the Fe-oxide bearing layers is determined by the oxygen penetration depth L. Cai and Sayles (1996) presented the following relationship between oxygen penetration depth L, benthic oxygen flux FO2 across the sediment-water interface and bottom water oxygen concentration [O2]bw: $L=2\theta D_s [O2]bw/FO2$ (Equation 1)

Here θ and Ds are the porosity and diffusivity of O2 in sediment, respectively.

Equation (1) shows that the thickness L of the oxidized layer on top of the sediment varies with [O2]bw and, allowing for some inertia, the minimum thickness L=Lmin should occur approximately when [O2]bw attains its minimum. This means that Lmin can be increased by increasing the minimum bottom water oxygen concentration [O2]bw which is in accordance with the statement on Line 479-481.

However, the following statement (on Line 481-483) is presented without any proof of its validity for the Baltic Sea. Citation from Line 481-483. "This process will, however, be delayed due to the prior deposition of organic rich sediments which results in a high upward flux of H2S (i.e. legacy of hypoxia) hindering the formation of Fe-oxides."

Reply: Please see our replies to the comments on Lines 29-31 and Lines 31-32 and to the comment directly preceding this one and the associated changes in the text.

This statement is maybe true for highly eutrophic lakes, but it is not true for the deepwater sediments in the much less eutrophic Baltic Sea, as discussed on p. 41 in Stigebrandt (2018). Using Sediment Profile Imagery (SPI) it was observed that the sediment surface was oxygenated within a couple of months during a natural oxygenation event due to a Major Baltic Inflow (Rosenberg et al., 2016). This means that the upward flux of $H_2S$ in the Baltic Sea deepwater sediments is not large enough to hinder the formation of an oxic layer (containing Fe-oxides) on top of the sediment when the bottom water is oxygenated. Therefore Equation (1) is applicable to the deep sediments of Baltic Sea. The oxygen penetration depth L can thus be increased by increasing [O2]bw by artificial reoxygenation of the bottom waters of the Baltic Sea.

Reply: In the revised text, we have now specifically limited our discussion to artificial reoxygenation of the Stockholm Archipelago. We note, however, that a similar legacy effect due to the upward flux of hydrogen sulfide has been reported previously for the Gotland Deep following the most recent Major Baltic Inflow by Hermans et al. (2019). Instead of visual observations and conclusions on the presence or absence of Fe oxides based on sediment imagery, Hermans et al. (2019) quantified the sediment content of Fe oxides and associated P in Gotland Deep sediments. The results revealed only limited Fe oxide formation and very little sequestration of P. This finding was independently corroborated by water column studies of P dynamics in the Gotland Deep showing that most P was displaced to other parts of the Baltic Sea. The lack of Fe oxide formation was attributed to the high flux of reductants, such as sulfide from the deeper sediments which allowing the presence and preservation of FeS (or $FeS_2$) and restricted the penetration of $O_2$ into the sediment. We show here that sediments in the Stockholm Archipelago have quite similar characteristics to those in the central Baltic Sea, i.e. high contents of organic matter, high pore water sulfide concentrations and high sedimentary concentrations of FeS and $FeS_2$.

The major effect of oxygenation of anoxic bottom sediments is that it stops the outflow of P from the sediment. This was discussed in Stigebrandt et al. (2014), see also Almroth-Rosell et al. (2015) who show that the phosphorus release rate from the sediment drastically decreased and even became negative as a result of Major Baltic Inflows. As shown in Stigebrandt (2018), artificial reoxygenation of bottom waters should be a rapid and long-term effective measure towards reducing the eutrophication and improving the water quality of the open Baltic Sea and coastal areas with good water exchange with the open sea so that local effects of local land-based nutrient supplies are small. This disproves the following statement (on Line 483-485) in the manuscript.

Citation from Line 483 - 485 "This also explains why artificial reoxygenation of bottom waters (e.g. Stigebrandt and Gustafsson, 2007) will not be a long-term effective measure towards improving the water quality of the (coastal) Baltic Sea."

Reply: We agree that the immediate response of oxygenation (artificial or natural) of bottom waters decreases the release of P from the sediment. Our results also show that less or non-reducing bottom waters, i.e. as observed for the year-round well-oxygenated Ingaröfjärden, leads to a larger surface sedimentary P pool. At depth, however, sedimentary P distributions and concentrations are rather similar for all study sites (with the exception of the enrichments in Fe-P at Strömmen). This implies that bottom water oxygen concentrations have little to no effect on permanent P burial, i.e. permanent long-term removal. Therefore nutrient load reductions are necessary to improve the ecological status of the Stockholm Archipelago. We have revised the text to clarify this point (Lines 519-522): "Because of this legacy effect, we expect that artificial reoxygenation of bottom waters

(e.g. Stigebrandt and Gustafsson, 2007), if applied in the Stockholm Archipelago, is unlikely to be a long-term effective measure towards improving the water quality since it does not stimulate permanent P burial in these sediments and a large impact on the Fe-P pool is hindered by the high upward $H_2S$ flux."

Line 516. What is meant by "control" in the sentence "continue to actively reduce and control nutrient inputs"

Reply: Here, "control" refers to "managing" the nutrients inputs, which can be done through all kinds of regulations, incl. installation of sewage treatment plants etc. We have now removed "and control" from the text. This sentence now reads (540-541): "Thus, these coastal sediments are likely to continue to contribute to removal of P and N as long as we continue to actively reduce nutrient inputs."

Line 514-515. In the manuscript it is postulated but not shown that the sediments are efficient filter. This would require that estimates of the N and P sinks (tonnes year$^{-1}$) for the whole area were related to the total supply of nutrients (tonnes year$^{-1}$), i.e. the supply from both land-based and sea-based sources.

Reply: The aim of our study is to assess the processes leading to the removal of P and N in the sediments of the Stockholm Archipelago. We never had the intention, nor claim that we would calculate the filter efficiency of the system. Please, also see our reply to the comment to Line 121 and the general comment of the reviewer.

[revised manuscript text omitted]
$. 'Sediment top' and 'Sediment bottom' indicate the top and bottom depth of the interval for which the diffuse flux was determined.

| | Strömmen | Baggensfjärden | Erstaviken | Ingaröfjarden |
|---|---|---|---|---|
| Sediment top (cm) | 0.75 | 0.75 | 1.75 | 8.25 |
| Sediment bottom (cm) | 2.25 | 7.25 | 8.25 | 15 |
| $H_2S$ top ($\mu$mol $L^{-1}$) | 2 | 3 | 7 | 36 |
| $H_2S$ bottom ($\mu$mol $L^{-1}$) | 385 | 899 | 1111 | 1340 |
| Diffusive flux (mmol $m^{-2}$ $d^{-1}$) | 7.6 | 4.2 | 5.2 | 6.5 |

**Table 43.** Sedimentary concentrations of organic carbon ($C_{org}$), nitrogen (N), phosphorus (P) and calcium carbonate for the different study sites.

995

| | Depth interval (cm) | Strömmen | Baggensfjärden* | Erstaviken* | Ingaröfjärden* |
|---|---|---|---|---|---|
| $C_{org}$ avg. (wt. %) | 0-2 | 7.9 | 6.3 | 6.0 | 5.1 |
| $C_{org}$ avg. (wt. %) | 10-40 | 6.3 | 4.5 | 4.5 | 3.8 |
| $CaCO_3$ avg. (wt. %) | Entire core | 2.5 | 2.3 | 2.4 | 2.9 |
| N avg. (wt. %) | 0-2 | 0.99 | 0.83 | 0.78 | 0.69 |
| N avg. (wt. %) | 10-40 | 0.59 | 0.54 | 0.54 | 0.48 |
| P avg. (wt. %) | 0-2 | 0.36 | 0.17 | 0.19 | 0.25 |
| P avg. (wt. %) | 10-40 | 0.14 | 0.10 | 0.11 | 0.11 |
| C/N avg. ($mol^{-1}$ $mol^{-1}$) | 0-2 | 9.4 | 8.9 | 9.0 | 8.7 |
| C/N avg. ($mol^{-1}$ $mol^{-1}$) | 10-40 | 12.4 | 9.6 | 9.8 | 9.1 |
| $C_{org}/P_{tot}$ avg. ($mol^{-1}$ $mol^{-1}$) | 0-2 | 69 | 96 | 95 | 53 |
| $C_{org}/P_{tot}$ avg. ($mol^{-1}$ $mol^{-1}$) | 10-40 | 116 | 116 | 108 | 88 |

*Organic carbon and nitrogen concentrations for Baggensfjärden, Erstaviken and Ingaröfjärden are derived from van Helmond et al. (in review).

1000

Table 54. Areal rates of benthic nitrate-reducing processes, including standard error (SE). DN 'Nitrification-denitrification' indicates the proportion of denitrification supported by $NO_3^-$ from nitrification (as opposed to water column nitrate). Bottom water nitrate concentrations and ammonium and nitrate fluxes from the surface sediments into the water column (calculated from pore water profiles), including standard error (SE).

1005

| | Strömmen | Baggensfjärden | Erstaviken | Ingaröfjärden |
|---|---|---|---|---|
| Denitrification - ($\mu$mol N m$^{-2}$ d$^{-1}$) (SE) | 1723 (774) | 685 (58) | 564 (86) | 90 (38) |
| DNRA - ($\mu$mol N m$^{-2}$ d$^{-1}$) (SE) | 11.1 (8.1) | 6.1 (1.7) | 3.6 (3.3) | 2.8 (0.4) |
| Nitrification-denitrification - ($\mu$mol N m$^{-2}$ d$^{-1}$) (SE) | 1027 (461) | 500 (42) | 500 (76) | 76 (32) |
| Nitrification-denitrification (%) | 59.6 | 73 | 88.6 | 84.4 |
| Anammox - ($\mu$mol N m$^{-2}$ d$^{-1}$) (SE) | 0.27 (0.1) | 0.76 (0.1) | 3.11 (0.5) | 44.12 (18.4) |
| Nitrification-denitrification (%) | 59.6 | 73 | 88.6 | 84.4 |
| N$_2$ anammox (%) | 0.02 | 0.11 | 0.55 | 32.93 |
| Bottom water nitrate ($\mu$mol L$^{-1}$) | 17.8 | 12.1 | 9.0 | 5.6 |
| Ammonium flux - ($\mu$mol N m$^{-2}$ d$^{-1}$) (SE) | 1399 (122.4) | 629 (88.8) | 600 (76.8) | 0 (0) |
| Nitrate flux - ($\mu$mol N m$^{-2}$ d$^{-1}$) (SE) | 4.1 (0.05) | -1.44 (1.0) | -7.68 (0.24) | -85.7 (35.0) |

1010

1015

**Table 65.** Burial rates of total and reactive P

[revised manuscript text omitted]

1080    **Figure 98.** Relationship between denitrification and bottom water nitrate concentrations, and upper sediment $C_{org}$

content for the study sites in the Stockholm Archipelago.

[Figure]

[Figure]

**Figure 910.** Surface sedimentary P  pools for the study sites in the Stockholm Archipelago. The red color indicates the enriched surface sediment layer, or "top layer" (Table 65). Dashed lines indicate "background" sedimentary P.

[Figure]

---

## Author Response (AR2)

This revised version of the MS contains, unfortunately, some important conclusions at the system level that are not correct due to ignorance regarding oceanographic conditions at the study sites. Additional revision of the MS is needed as detailed below.

We respond point-by-point to the reviewer's comments below.

Line 21-23. This statement is refuted by your own data. Ingaröfjärden has a higher upward sulphide flux towards the sediment surface than Baggensfjärden (Table 3). Despite this, Ingaröfjärden has five times larger Fe-oxide-bound P pool. The explanation for this is the much better oxygen penetration of the sediment due to higher oxygen concentration in the bottom water in Ingaröfjärden than in Baggensfjärden (Fig. 2a), see also comments for Line 133-143.

Reply: The formation of a larger Fe-oxide-bound P pool in winter is hindered at all sites. Hence, this also holds for Ingaröfjärden, although clearly to a lesser extent. Indeed, where the water column mixing naturally is relatively low, i.e. at the (most) restricted sites such as Baggensfjärden, and where low bottom water oxygen conditions prevail in summer, the hindrance is greatest. To clarify this we have amended this sentence in the abstract (Lines 21-25): "The abundant presence of sulfide in the porewater and its high upward flux towards the sediment surface ($\sim$4 to 8 mmol m$^{-2}$ d$^{-1}$), linked to prior deposition of organic-rich sediments in a low $O_2$ setting ("legacy of hypoxia"), hinders the formation of a larger Fe-oxide-bound P pool in winter. **This is most pronounced at sites where water column mixing is naturally relatively low and where low bottom water $O_2$ concentrations prevail in summer.**"

While indeed there is a larger Fe-oxide-bound P pool at Ingaröfjärden than at the other sites, we again emphasize that this is a temporary sink for P in surface sediments only and is not the permanent burial of P deeper in the sediment. Hence, the differences in bottom water oxygenation between the sites do not enhance the permanent burial of P.

Line 32-34. This should not be in the Abstract because it is not a conclusion based on results achieved in this MS.

Reply: We have rephrased this sentence to clarify the link between our results on sediment removal of nutrients to the expected response to a load reduction (Lines 34-36): "Further nutrient load reductions are expected to contribute to the recovery of the eutrophic Stockholm Archipelago from hypoxia and to continue to support the capacity of the sediments to remove part of the P and N loads".

Line 85-86. In your reply to my comments for Line 85 and Line 475 (in the original MS) you say that Hansen et al. (2019) does not contain new data. This is wrong, these authors included new data showing that the amount of anoxic water and the area of anoxic bottoms were record high in 2018. This is not a sign of recovery in the region which should be mentioned to give a balanced view.

Reply: In our replies to the comments on Lines 85 and 475 (original MS) we wrote that the Hansson et al. (2019) report, which presents results of an oxygen survey in the open Baltic Sea, does not provide new data on **eutrophication**. Eutrophication includes many more aspects of water quality than oxygen alone. In response to the reviewer's comment we have now modified the text to clarify further that we are referring to coastal regions and we have added a statement that this does not yet hold for the Baltic Proper (Lines 87-90): "Active nutrient reductions from the 1980s onward (Gustafsson et al., 2012) are

now leading to the first signs of recovery in the region (Andersen et al., 2017), although not yet in the Baltic Proper (Hansson et al., 2019). A good example of a recovering coastal system within the Baltic Sea is the Stockholm Archipelago (Karlsson et al., 2010),…."

Line 99-101. I cannot see that the objective "to determine the time scales that govern removal and the implications for management strategies" is met.

Reply: We have removed this sentence.

Line 133-143. (Study sites)

The much greater temperature range in the bottom water in Ingaröfjärden (Fig. 2) than in Baggensfjärden proves that the water exchange in summer and autumn is much more frequent in Ingaröfjärden than in Baggensfjärden where winter temperatures prevail all the year around. This explains the much smaller amplitudes of the recorded annual cycles of P and O2 in Ingaröfjärden than in Baggensfjärden (Fig. 3a).The solid black line in Fig. 3a is not explained. I assume that it is the theoretical relationship between O2 and P in a closed water volume where organic matter of Redfield composition is decomposed. Data from Ingaröfjärden have a much smaller slope than this line, which demonstrates the vivid exchange of bottom water in Ingaröfjärden. Data from Slussen are clearly displaced above the solid line which may be explained by large contributions from the submerged outlets, interleaved in the deepwater, from sewage treatment plants. The oceanographic conditions at the study sites are of great importance for the interpretation of the surface layer sediment data and should thus be more fully described in the MS.

Reply: The differences in water exchange for the basins was discussed in section 4.1 in the original manuscript. The difference in amplitude of the changes in oxygen and total P concentrations between sites is likely the combined effect of differences in water column mixing and in recycling of P from the sediment associated with changes in bottom water oxygen. In response to the reviewer's comment, we have rewritten this paragraph to further clarify the characteristics of the basins regarding water column mixing and we have added this latter statement. We note again that our focus in this manuscript is on permanent removal of P through burial in the sediment. The water column data are provided for general context.

The paragraph now reads as follows (Lines 134-153): "For this study, sediments and bottom water from four different locations in the inner and intermediate part of the Stockholm Archipelago (cf. Almroth-Rosell et al., 2016; Fig. 1) were collected. The study sites are located in the basins Strömmen (central Stockholm), Baggensfjärden, Erstaviken and Ingaröfjärden and are characterized by a range of water depths and bottom water redox conditions (Fig. 2; Table 1; Sup. Fig. 1). Strömmen is located most proximate to the outlets of the sewage treatment plants close to the center of Stockholm, which presumably contributes to the relatively high total P concentrations in the bottom water (Fig. 3a). Baggensfjärden is the most restricted basin in this study (i.e. land-locked with narrow and relatively shallow connections to adjacent basins), leading to reduced water column mixing, culminating in low $O_2$ bottom waters annually in summer. Erstaviken and Ingaröfjärden have a more open connection with the Baltic Proper leading to better water column mixing. Ingaröfjärden is the least restricted basin in this study and subsequently the most consistently well-oxygenated basin throughout

the year. Extensive water quality monitoring of the study area (obtained from the SHARK database at http://www.smhi.se/klimatdata/oceanografi/havsmiljodata/marina-miljoovervakningsdata; Swedish Meteorological and Hydrological Institute - SMHI, 2019), shows a clear inverse correlation between bottom water $O_2$ concentrations and P and a positive correlation between bottom water $O_2$ and N/P-ratios (Fig. 3a,b). Bottom water $O_2$ and nutrient concentrations follow a distinct annual pattern, with maximum $O_2$ and minimum nutrient concentrations in winter. After winter, $O_2$ gradually drops and nutrient concentrations gradually increase, reaching minimum and maximum values, respectively, at the end of summer and in autumn, followed by a reset of the system (Fig. 3c, d). The difference in amplitude of the changes in $O_2$ and total P concentrations between sites is likely the combined effect of differences in water column mixing and in recycling of P from the sediment associated with changes in bottom water $O_2$."

The black line depicts the inverse linear relationship between bottom water total P and bottom water $O_2$. We have added this information to the caption of figure 3.

Line 341. Is this true also for Slussen? Please explain the effect of the contribution from the submerged outlets from the sewage treatment plants.

Reply: In this sentence, we are discussing sources of nitrate for denitrification in the sediment, as shown in Table 5 and Fig. 6. We have now clarified that this is the case for all studied sites, including Strömmen (Lines 358-359): "Nitrification is the dominant source of $NO_3^-$ **for denitrification in the sediments at all sites**, accounting for 60-89 % of all $NO_3^-$ supply **for denitrification in the sediments** (Table 5; Fig. 8).

The contribution from sewage treatment plants would be relevant if we were discussing water column processes in this paragraph and in this manuscript. That is not the subject here, however.

Line 389-391. The conclusion here is wrong. Ingaröfjärden has higher H2S flux to the surface sediment than both Baggensfjärden and Erstaviken (Table 3) but this does not hinder the formation of a thick Fe-bound P pool in the surface sediment in Ingaröfjärden. This shows that high oxygen concentration in the bottom water (Fig. 2a) is the key factor for the formation of a thick Fe-bound P pool is the surface sediment.

Reply: Our conclusion is correct. Also at Ingaröfjärden, the Fe-bound P pool would be larger if there was no high upward $H_2S$ flux. Indeed, Ingaröfjärden has the highest $H_2S$ flux but this $H_2S$-bearing sediment is located much deeper in the sediment when compared to the other sites. This lower impact of the upward $H_2S$ flux is indeed the result of the year-round higher bottom water oxygen concentrations at this site, which also allow for the establishment of macrofaunal in the sediment. But note again: the permanent burial of P is not altered because Fe-bound P is not a permanent sink for P in such a eutrophic setting, as is evident from our data. We now more explicitly discuss the contrasts between the sites (Lines 408-414): "As a consequence of the presence of $H_2S$ in the surface sediments and its high upward $H_2S$ flux, in combination with reduced water column mixing and/or seasonally low $O_2$ bottom water conditions, FeS is formed and/or preserved (Fig. 5), and formation of a large(r) pool of Fe-oxides and Fe-bound P pool is hindered at Strömmen, Baggensfjärden and Erstaviken. At Ingaröfjärden, the well-mixed water column and year-round well-oxygenated bottom water allow a deeper $O_2$ penetration (Sup. Fig. 3), preventing the presence of $H_2S$ in the surface sediment

despite its high upward flux (Table 3) and leading to a thicker Fe-oxide bearing layer (Fig. 5) and a larger Fe-bound P pool (Fig. 6)."

Line 400. Sink-switching should be defined at its first occurrence in the text.

Reply: We have added the definition of sink-switching at its first occurrence in the text (Lines 433-434): "… is little to no sink-switching of sediment P (i.e. the transformation of relatively labile P reservoirs such as Fe-oxide bound P and organic P to authigenic P minerals such as for example vivianite)"

Line 467-469. It should be discussed if the high oxygen concentration in the bottom water in Ingaröfjärden may be an important factor.

Reply: Indeed higher bottom water oxygen concentrations play a role in determining the degradation of organic matter. We have amended the text to clarify this further (Lines 492-498): "However, in complex basin systems such as the Stockholm Archipelago, and the Baltic Sea coastal zone in general, differences in ventilation and retention times between basins (implying differences in vertical and lateral exchange of water and $O_2$, and hence, variations in bottom water $O_2$) may mean that $C_{org}$ inputs are more variable than assumed (see section 4.2.1). Available $C_{org}$ in Ingaröfjärden (Table 2) may be less labile than at other sites due to such variations in hydrology and bottom water $O_2$, with the deep (18 mm) $O_2$ penetration indicating a lower organic matter reactivity and sediment respiration compared to the other sites."

Line 514 – 523 The conclusion here is not in accordance with your own data. As pointed out above, the high upward H2S flux (the "legacy" effect) does not hinder the build-up of a large Fe-P pool if the oxygen concentration is high in the bottom water like it is in Ingaröfjärden. This shows that long-term artificial reoxygenation (e.g. Stigebrandt and Gustafsson, 2007) would be an effective measure towards improving the water quality in the Stockholm archipelago. As serious scientists, you should of course report and not withhold this important conclusion.

Reply: Indeed, higher bottom water oxygen concentrations lead to a larger Fe-bound P pool in surface sediments, as is known from the literature and is evident from our dataset. However, as demonstrated in this manuscript (Figures 6 and 10), this will not increase the permanent removal of P, because this Fe-bound P pool in the surface sediment is only a temporary sink for P and sink-switching to other permanent P burial forms is absent. Therefore reoxygenation of bottom waters will only increase the size of this temporary P sink. If nutrient loads remain unchanged, long-term water quality will not improve because this requires an increase in permanent removal of P. This is for example well-known from lake studies on systems where sulfide formation in the porewater is high relative to the Fe supply (e.g. Gächter and Müller, 2003)

[revised manuscript text omitted]

---

## Author Response (AR3)

Dear Prof. Bange,

Thank you for forwarding this review. We respond to the comments by the reviewer below.

Sincerely, on behalf of all authors,

Dr. Niels van Helmond and Dr. Elizabeth Robertson

**Response to reviewer**

The MS is now fine with many useful observations and analyses. However, there is still a serious inconsistency, between two different cases, in the evaluation of the effects of recovery from hypoxia upon P burial. I have asked questions about these evaluations in my previous reviews. The inconsistency should be solved, and the text revised accordingly before publication is recommended.

Reply: As explained below in our detailed responses, there is no inconsistency. We have revised the text to clarify this.

Case I; L 94 – 95 and L 571-572. In the Stockholm Archipelago recovery of hypoxia may be associated with increased P burial (Norkko et al., 2010). Please explain how you define increased P burial in Case I. How is it related to permanent P burial?

Reply: We have revised these sections to explain the results of the Norkko et al. (2012) study in more detail (Lines 90-95): "A good example of a recovering coastal system within the Baltic Sea is the Stockholm Archipelago (Karlsson et al., 2010), where, based on modeling, recovery from hypoxia was suggested to be potentially associated with the build-up of a pool of Fe-oxide bound P in surface sediments driven by increased macrofaunal activity (Norkko et al., 2012). However, this mechanism would not lead to increased permanent P burial and hence, by itself, not lead to long term recovery of the system (> 5-10 years)."

(Lines 566-574): "Continued recovery of the Stockholm Archipelago is also likely to lead to (re-)colonization by bioturbating macrofaunal populations that have been driven out by hypoxic bottom waters (Diaz and Rosenberg, 2008; Voss et al., 2011). This may enhance temporary P burial and denitrification by sediment reworking and oxygenation (e.g. Pelegri and Blackburn, 1995; Laverock et al., 2011; Norkko et al., 2012). While we still lack the predictive capabilities required to allow us to assess how fauna may influence sediment biogeochemistry (Griffiths et al., 2017; Robertson et al., 2019), reductions in nutrient inputs and phytoplankton bloom intensities, and eventual recolonization by fauna at inner archipelago sites will likely sustain active P and N removal processes. Thus, these coastal sediments are likely to continue to contribute to removal of P and N as long as we continue to actively reduce nutrient inputs."

Case II; L 550 – 555. Increases in the bottom water O2 concentration would allow thicker Fe-oxide bearing layers and a larger Fe-bound P pool in the surface sediments, hence a larger surface sedimentary P pool. This process will however be delayed due to prior deposition of organic rich sediments which results in a high upward flux of H2S hindering the formation of Fe-oxides that can bind P. Because of this we expect that artificial reoxygenation of bottom waters (e.g. Stigebrandt and Gustafsson, 2007), if applied in the Stockholm Archipelago, is unlikely to be a long-term effective measure towards improving the water quality since it does not stimulate permanent burial in these sediments and a large impact on the Fe-P pool is hindered by the high upward H2S flux.

Please explain why recovery from hypoxia in Case I, but not recovery from hypoxia by artificial

reoxygenation in Case II, will lead to increased P burial.

Reply: In both cases, permanent burial will not increase and hence, with this measure only, long-term recovery is not achieved. This should now be evident from the revised text (see previous reply).

L 35 – 38. It is postulated that nutrient load reductions support the capacity of the sediments to remove part of the P load. Is this true? The mechanism(s) behind this effect should be explained in the MS.

Reply: We note that the mechanisms for P and N removal are described in detail in line 27 and line 30 of the abstract. We have revised the final sentence of the abstract and removed "support the capacity" since it apparently led to confusion (Lines 34-37): "Further nutrient load reductions are expected to contribute to the recovery of the eutrophic Stockholm Archipelago from hypoxia. Based on the dominant pathways of P and N removal identified in this study, it is expected that the sediments will continue to remove part of the P and N loads."

We have also modified the last sentence of the conclusion section to avoid confusion and to ensure this is in line with the revised abstract (Lines 591-595): "Further reductions in P and N inputs are expected to reduce the frequency of hypoxic events. Our results show that the permanent burial of P is largely independent from bottom water redox conditions. Increased bottom water oxygen is expected to allow benthic denitrification to be sustained. Hence, we expect that the sediments in the Stockholm Archipelago will continue to remove part of the P and N loads upon reduction of such loads."

Finally, the theoretical paper by Stigebrandt and Gustafsson (2007) is not particularly relevant for oxygenation of the Stockholm Archipelago. The authors should instead refer to the relevant inshore oxygenation experiment in the Byfjord described in Stigebrandt et al. (Ambio, 44, 42-54 2015).

Reply: We have replaced the reference as suggested.

[revised manuscript text omitted]